# Seeing is not Believing: Robust Reinforcement Learning against Spurious Correlation

**Wenhao Ding**[1]*     **Laixi Shi**[2]*     **Yuejie Chi**[1]     **Ding Zhao**[1]
[1]Carnegie Mellon University     [2]California Institute of Technology
{wenhaod,laixis,yuejiec}@andrew.cmu.edu, dingzhao@cmu.edu

## Abstract

Robustness has been extensively studied in reinforcement learning (RL) to handle various forms of uncertainty such as random perturbations, rare events, and malicious attacks. In this work, we consider one critical type of robustness against spurious correlation, where different portions of the state do not have correlations induced by unobserved confounders. These spurious correlations are ubiquitous in real-world tasks, for instance, a self-driving car usually observes heavy traffic in the daytime and light traffic at night due to unobservable human activity. A model that learns such useless or even harmful correlation could catastrophically fail when the confounder in the test case deviates from the training one. Although motivated, enabling robustness against spurious correlation poses significant challenges since the uncertainty set, shaped by the unobserved confounder and causal structure, is difficult to characterize and identify. Existing robust algorithms that assume simple and unstructured uncertainty sets are therefore inadequate to address this challenge. To solve this issue, we propose *Robust State-Confounded Markov Decision Processes* (RSC-MDPs) and theoretically demonstrate its superiority in avoiding learning spurious correlations compared with other robust RL counterparts. We also design an empirical algorithm to learn the robust optimal policy for RSC-MDPs, which outperforms all baselines in eight realistic self-driving and manipulation tasks. Please refer to the website for more details.

## 1  Introduction

Reinforcement learning (RL), aiming to learn a policy to maximize cumulative reward through interactions, has been successfully applied to a wide range of tasks such as language generation [1], game playing [2], autonomous driving [3], etc. While standard RL has achieved remarkable success in simulated environments, a growing trend in RL is to address another critical concern – **robustness** – with the hope that the learned policy still performs well when the deployed (test) environment deviates from the nominal one used for training [4]. Robustness is highly desirable since the performance of the learned policy could significantly deteriorate due to the uncertainty and variations of the test environment induced by random perturbation, rare events, or even malicious attacks [5, 6].

Despite various types of uncertainty that have been investigated in RL, this work focuses on the uncertainty of the environment with semantic meanings resulting from some unobserved underlying variables. Such environment uncertainty, denoted as **structured uncertainty**, is motivated by innumerable real-world applications but still receives little attention in sequential decision-making tasks [7]. To specify the phenomenon of structured uncertainty, let us consider a concrete example (illustrated in Figure 1) in a driving scenario, where a shift between training and test environments caused by an unobserved confounder can potentially lead to a severe safety issue. Specifically, the observations *brightness* and *traffic density* do not have cause and effect on each other but are

---

*Equal contribution.

37th Conference on Neural Information Processing Systems (NeurIPS 2023).

controlled by a confounder (i.e. *sun* and *human activity*) that is usually unobserved [2] to the agent. During training, the agent could memorize the **spurious state correlation** between *brightness* and *traffic density*, i.e., the traffic is heavy during the daytime but light at night. However, such correlation could be problematic during testing when the value of the confounder deviates from the training one, e.g., the traffic becomes heavy at night due to special events (*human activity* changes), as shown at the bottom of Figure 1. Consequently, the policy dominated by the spurious correlation in training fails on out-of-distribution samples (observations of heavy traffic at night) in the test scenarios.

The failure of the driving example in Figure 1 is attributed to the widespread and harmful spurious correlation, namely, the learned policy is not robust to the structured uncertainty of the test environment caused by the unobserved confounder. However, ensuring robustness to structured uncertainty is challenging since the targeted uncertain region – the structured uncertainty set of the environment – is carved by the unknown causal effect of the unobserved confounder, and thus hard to characterize. In contrast, prior works concerning robustness in RL [8] usually consider a homogeneous and structure-agnostic uncertainty set around the state [9, 6, 10], action [11, 12], or the training environment [13–15] measured by some heuristic functions [9, 15, 8] to account for unstructured random noise or small perturbations. Consequently, these prior works could not cope with the structured uncertainty since their uncertainty set is different from and cannot tightly cover the desired structured uncertainty set, which could be heterogeneous and allow for potentially large deviations between the training and test environments.

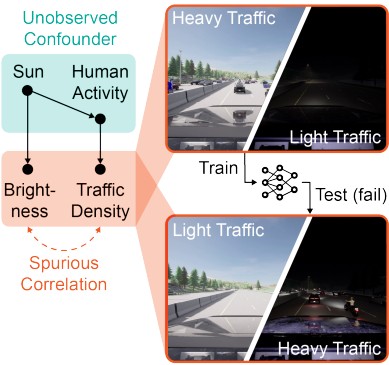

Figure 1: A model trained only with heavy traffic in the daytime learns the spurious correlation between brightness and traffic density and could fail to drive in light traffic in the daytime.

In this work, to address the structured uncertainty, we first propose a general RL formulation called State-confounded Markov decision processes (SC-MDPs), which model the possible causal effect of the unobserved confounder in an RL task from a causal perspective. SC-MDPs better explain the reason for semantic shifts in the state space than traditional MDPs. Then, we formulate the problem of seeking robustness to structured uncertainty as solving Robust SC-MDPs (RSC-MDPs), which optimizes the worst performance when the distribution of the unobserved confounder lies in some uncertainty set. The key contributions of this work are summarized as follows.

- We propose a new type of robustness with respect to structured uncertainty to address spurious correlation in RL and provide a formal mathematical formulation called RSC-MDPs, which are well-motivated by ubiquitous real-world applications.
- We theoretically justify the advantage of the proposed RSC-MDP framework against structured uncertainty over the prior formulation in robust RL without semantic information.
- We implement an empirical algorithm to find the optimal policy of RSC-MDPs and show that it outperforms the baselines on eight real-world tasks in manipulation and self-driving.

## 2  Preliminary and Limitations of Robust RL

In this section, we first introduce the preliminary formulation of standard RL and then discuss a natural type of robustness that is widely considered in the RL literature and most related to this work – robust RL.

**Standard Markov decision processes (MDPs).** An episodic finite-horizon standard MDP is represented by $\mathcal{M} = \{\mathcal{S}, \mathcal{A}, T, r, P\}$, where $\mathcal{S} \subseteq \mathbb{R}^n$ and $\mathcal{A} \subseteq \mathbb{R}^{d_A}$ are the state and action spaces, respectively, with $n/d_A$ being the dimension of state/action. Here, $T$ is the length of the horizon; $P = \{P_t\}_{1 \leq t \leq T}$, where $P_t : \mathcal{S} \times \mathcal{A} \to \Delta(\mathcal{S})$ denotes the probability transition kernel at time step $t$, for all $1 \leq t \leq T$; and $r = \{r_t\}_{1 \leq t \leq T}$ denotes the reward function, where $r_t : \mathcal{S} \times \mathcal{A} \to [0, 1]$ represents the deterministic immediate reward function. A policy (action selection rule) is denoted by $\pi = \{\pi_t\}_{1 \leq t \leq T}$, namely, the policy at time step $t$ is $\pi_t : \mathcal{S} \to \Delta(\mathcal{A})$ based on the current state $s_t$ as $\pi_t(\cdot \,|\, s_t)$. To represent the long-term cumulative

---

[2]sometimes they are observed but ignored given so many variables to be considered in neural networks.

reward, the value function $V_t^{\pi,P} : \mathcal{S} \to \mathbb{R}$ and Q-value function $Q_t^{\pi,P} : \mathcal{S} \times \mathcal{A} \to \mathbb{R}$ associated with policy $\pi$ at step $t$ are defined as $V_t^{\pi,P}(s) = \mathbb{E}_{\pi,P}\left[\sum_{k=t}^{T} r_k(s_k, a_k) \mid s_k = s\right]$ and $Q_t^{\pi,P}(s,a) = \mathbb{E}_{\pi,P}\left[\sum_{k=t}^{T} r_k(s_k, a_k) \mid s_t = s, a_t = a\right]$, where the expectation is taken over the sample trajectory $\{(s_t, a_t)\}_{1 \le t \le T}$ generated following $a_t \sim \pi_t(\cdot \mid s_t)$ and $s_{t+1} \sim P_t(\cdot \mid s_t, a_t)$.

**Robust Markov decision processes (RMDPs).** As a robust variant of standard MDPs motivated by distributionally robust optimization, RMDP is a natural formulation to promote robustness to the uncertainty of the transition probability kernel [13, 15], represented as $\mathcal{M}_{\text{rob}} = \{\mathcal{S}, \mathcal{A}, T, r, \mathcal{U}^\sigma(P^0)\}$. Here, we reuse the definitions of $\mathcal{S}, \mathcal{A}, T, r$ in standard MDPs, and denote $\mathcal{U}^\sigma(P^0)$ as an uncertainty set of probability transition kernels centered around a nominal transition kernel $P^0 = \{P_t^0\}_{1 \le t \le T}$ measured by some 'distance' function $\rho$ with radius $\sigma$. In particular, the uncertainty set obeying the $(s,a)$-rectangularity [16] can be defined over all $(s,a)$ state-action pairs at each time step $t$ as

$$\mathcal{U}^\sigma(P^0) := \otimes\, \mathcal{U}^\sigma(P_{t,s,a}^0), \qquad \mathcal{U}^\sigma(P_{t,s,a}^0) := \left\{ P_{t,s,a} \in \Delta(\mathcal{S}) : \rho\left(P_{t,s,a}, P_{t,s,a}^0\right) \le \sigma \right\}, \quad (1)$$

where $\otimes$ denotes the Cartesian product. Here, $P_{t,s,a} := P_t(\cdot \mid s,a) \in \Delta(\mathcal{S})$ and $P_{t,s,a}^0 := P_t^0(\cdot \mid s,a) \in \Delta(\mathcal{S})$ denote the transition kernel $P_t$ or $P_t^0$ at each state-action pair $(s,a)$ respectively. Consequently, the next state $s_{t+1}$ follows $s_{t+1} \sim P_t(\cdot \mid s_t, a_t)$ for any $P_t \in \mathcal{U}^\sigma(P_{t,s_t,a_t}^0)$, namely, $s_{t+1}$ can be generated from any transition kernel belonging to the uncertainty set $\mathcal{U}^\sigma(P_{t,s_t,a_t}^0)$ rather than a fixed one in standard MDPs. As a result, for any policy $\pi$, the corresponding *robust value function* and *robust Q function* are defined as

$$V_t^{\pi,\sigma}(s) := \inf_{P \in \mathcal{U}^\sigma(P^0)} V_t^{\pi,P}(s), \qquad Q_t^{\pi,\sigma}(s,a) := \inf_{P \in \mathcal{U}^\sigma(P^0)} Q_t^{\pi,P}(s,a), \qquad (2)$$

which characterize the cumulative reward in the worst case when the transition kernel is within the uncertainty set $\mathcal{U}^\sigma(P^0)$. Using samples generated from the nominal transition kernel $P^0$, the goal of RMDPs is to find an optimal robust policy that maximizes $V_1^{\pi,\sigma}$ when $t = 1$, i.e., perform optimally in the worst case when the transition kernel of the test environment lies in a prescribed uncertainty set $\mathcal{U}^\sigma(P^0)$.

**Lack of semantic information in RMDPs.** In spite of the rich literature on robustness in RL, prior works usually hedge against the uncertainty induced by unstructured random noise or small perturbations, specified as a small and homogeneous uncertainty set around the nominal one. For instance, in RMDPs, people usually prescribe the uncertainty set of the transition kernel using a heuristic and simple function $\rho$ with a relatively small $\sigma$. However, the unknown uncertainty in the real world could have a complicated and semantic structure that cannot be well-covered by a homogeneous ball regardless of the choice of the uncertainty radius $\sigma$, leading to either over conservative policy (when $\sigma$ is large) or insufficient robustness (when $\sigma$ is small). Altogether, we obtain the natural motivation of this work: *How to formulate such structured uncertainty and ensure robustness against it?*

## 3 Robust RL against Structured Uncertainty from a Causal Perspective

To describe structured uncertainty, we choose to study MDPs from a causal perspective with a basic concept called the structural causal model (SCM). Armed with the concept, we formulate State-confounded MDPs – a broader set of MDPs in the face of the unobserved confounder in the state space. Next, we provide the main formulation considered in this work – robust state-confounded MDPs, which promote robustness to structured uncertainty.

**Structural causal model.** We denote a structural causal model (SCM) [17] by a tuple $\{X, Y, F, P^x\}$, where $X$ is the set of exogenous (unobserved) variables, $Y$ is the set of endogenous (observed) variables, and $P^x$ is the distribution of all the exogenous variables. Here, $F$ is the set of structural functions capturing the causal relations between $X$ and $Y$ such that for each variable $y_i \in Y$, $f_i \in F$ is defined as $y_i \leftarrow f_i(\mathsf{PA}(y_i), x_i)$, where $x_i \subseteq X$ and $\mathsf{PA}(y_i) \subseteq Y \setminus y_i$ denotes the parents of the node $y_i$. We say that a pair of variables $y_i$ and $y_j$ are confounded by a variable $C$ (confounder) if they are both caused by $C$, i.e., $C \in \mathsf{PA}(y_i)$ and $C \in \mathsf{PA}(y_j)$. When two variables $y_i$ and $y_j$ do not have direct causality, they are still correlated if they are confounded, in which case this correlation is called spurious correlation.

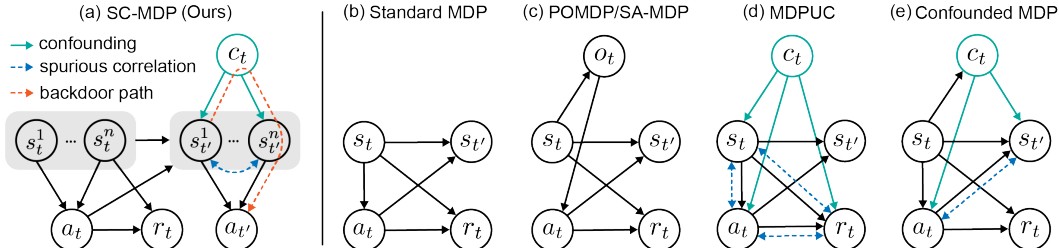

Figure 2: The probabilistic graphs of our formulation (SC-MDP) and other related formulations (specified in Appendix B.1 due to the limited space). $s_t^1$ means the first dimension of $s_t$. $s_{t'}$ is a shorthand for $s_{t+1}$. In SC-MDP, the orange line represents the backdoor path from state $s_{t'}^1$ to action $a_{t'}$ opened by the confounder $c_t$, which makes the learned policy $\pi$ rely on the value of $c_t$.

## 3.1 State-confounded MDPs (SC-MDPs)

We now present state-confounded MDPs (SC-MDPs), whose probabilistic graph is illustrated in Figure 2(a) with a comparison to standard MDPs in Figure 2(b). Besides the components in standard MDPs $\mathcal{M} = \{\mathcal{S}, \mathcal{A}, T, r\}$, we introduce a set of unobserved confounder $C_{\mathsf{s}} = \{c_t\}_{1 \leq t \leq T}$, where $c_t \in \mathcal{C}$ denotes the confounder that is generated from some unknown but fixed distribution $P_t^c$ at time step $t$, i.e., $c_t \sim P_t^c \in \Delta(\mathcal{C})$.

To characterize the causal effect of the confounder $C_{\mathsf{s}}$ on the state dynamic, we resort to an SCM, where $C_{\mathsf{s}}$ is the set of exogenous (unobserved) confounder and endogenous variables include all dimensions of states $\{s_t^i\}_{1 \leq i \leq n, 1 \leq t \leq T}$, and actions $\{a_t\}_{1 \leq t \leq T}$. Specifically, the structural function $F$ is considered as $\{\mathcal{P}_t^i\}_{1 \leq i \leq n, 1 \leq t \leq T}$ – the transition from the current state $s_t$, action $a_t$ and the confounder $c_t$ to each dimension of the next state $s_{t+1}^i$ for all time steps, i.e., $s_{t+1}^i \sim \mathcal{P}_t^i(\cdot \mid s_t, a_t, c_t)$. Notably, the specified SCM does not confound the reward, i.e., $r_t(s_t, a_t)$ does not depend on the confounder $c_t$.

Armed with the above SCM, denoting $P^c := \{P_t^c\}$, we can introduce state-confounded MDPs (SC-MDPs) represented by $\mathcal{M}_{\mathsf{sc}} = \{\mathcal{S}, \mathcal{A}, T, r, \mathcal{C}, \{\mathcal{P}_t^i\}, P^c\}$ (Figure 2(a)). A policy is denoted as $\pi = \{\pi_t\}$, where each $\pi_t$ results in an intervention (possibly stochastic) that sets $a_t \sim \pi_t(\cdot \mid s_t)$ at time step $t$ regardless of the value of the confounder.

**State-confounded value function and optimal policy.** Given $s_t$, the causal effect of $a_t$ on the next state $s_{t+1}$ plays an important role in characterizing value function/Q-function. To ensure the identifiability of the causal effect, the confounder $c_t$ are assumed to obey the backdoor criterion [17, 18], leading to the following *state-confounded value function* (SC-value function) and *state-confounded Q-function* (SC-Q function) [19]:

$$\widetilde{V}_t^{\pi, P^c}(s) = \mathbb{E}_{\pi, P^c}\left[\sum_{k=t}^T r_k(s_k, a_k) \mid s_t = s; c_k \sim P_k^c, s_{k+1}^i \sim \mathcal{P}_k^i(\cdot \mid s_k, a_k, c_k)\right],$$

$$\widetilde{Q}_t^{\pi, P^c}(s, a) = \mathbb{E}_{\pi, P^c}\left[\sum_{k=t}^T r_k(s_k, a_k) \mid s_t = s, a_t = a; c_k \sim P_k^c, s_{k+1}^i \sim \mathcal{P}_k^i(\cdot \mid s_k, a_k, c_k)\right]. \quad (3)$$

***Remark* 1.** Note that the proposed SC-MDPs serve as a general formulation for a broad family of RL problems that include standard MDPs as a special case. Specifically, any standard MDP $\mathcal{M} = \{\mathcal{S}, \mathcal{A}, P, T, r\}$ can be equivalently represented by at least one SC-MDP $\mathcal{M}_{\mathsf{sc}} = \{\mathcal{S}, \mathcal{A}, T, r, \mathcal{C}, \{\mathcal{P}_t^i\}, P^c\}$ as long as $\mathbb{E}_{c_t \sim P_t^c}\left[\mathcal{P}_t^i(\cdot \mid s_t, a_t, c_t)\right] = \left[P(\cdot \mid s_t, a_t)\right]_i$ for all $1 \leq i \leq n, 1 \leq t \leq T$.

## 3.2 Robust state-confounded MDPs (RSC-MDPs)

In this work, we consider robust state-confounded MDPs (RSC-MDPs) – a variant of SC-MDPs promoting the robustness to the uncertainty of the unobserved confounder distribution $P^c$, denoted by $\mathcal{M}_{\mathsf{sc\text{-}rob}} = \{\mathcal{S}, \mathcal{A}, T, r, \mathcal{C}, \{\mathcal{P}_t^i\}, \mathcal{U}^\sigma(P^c)\}$. Here, the perturbed distribution of the unobserved confounder is assumed in an uncertainty set $\mathcal{U}^\sigma(P^c)$ centered around the nominal distribution $P^c$

with radius $\sigma$ measured by some 'distance' function $\rho : \Delta(\mathcal{C}) \times \Delta(\mathcal{C}) \to \mathbb{R}^+$, i.e.,

$$\mathcal{U}^\sigma(P^c) := \otimes\, \mathcal{U}^\sigma(P_t^c), \qquad \mathcal{U}^\sigma(P_t^c) := \{P \in \Delta(\mathcal{C}) : \rho\,(P, P_t^c) \le \sigma\}. \tag{4}$$

Consequently, the corresponding *robust SC-value function* and *robust SC-Q function* are defined as

$$\widetilde{V}_t^{\pi,\sigma}(s) := \inf_{P \in \mathcal{U}^\sigma(P^c)} \widetilde{V}_t^{\pi,P}(s), \quad \widetilde{Q}_t^{\pi,\sigma}(s,a) := \inf_{P \in \mathcal{U}^\sigma(P^c)} \widetilde{Q}_t^{\pi,P}(s,a), \tag{5}$$

representing the worst-case cumulative rewards when the confounder distribution lies in the uncertainty set $\mathcal{U}^\sigma(P^c)$.

Then a natural question is: does there exist an optimal policy that maximizes the robust SC-value function $\widetilde{V}_t^{\pi,\sigma}$ for any RSC-MDP so that we can target to learn? To answer this, we introduce the following theorem that ensures the existence of the optimal policy for all RSC-MDPs. The proof can be found in Appendix C.1.

**Theorem 1** (Existence of an optimal policy). *Let $\Pi$ be the set of all non-stationary and stochastic policies. Consider any RSC-MDP, there exists at least one optimal policy $\pi^{\mathsf{SC},\star} = \{\pi_t^{\mathsf{SC},\star}\}_{1 \le t \le T}$ such that for all $(s,a) \in \mathcal{S} \times \mathcal{A}$ and $1 \le t \le T$, one has*

$$\widetilde{V}_t^{\pi^{\mathsf{SC},\star},\sigma}(s) = \widetilde{V}_t^{\star,\sigma}(s) := \sup_{\pi \in \Pi} \widetilde{V}_t^{\pi,\sigma}(s) \quad and \quad \widetilde{Q}_t^{\pi^{\mathsf{SC},\star},\sigma}(s,a) = \widetilde{Q}_t^{\star,\sigma}(s,a) := \sup_{\pi \in \Pi} \widetilde{Q}_t^{\pi,\sigma}(s,a).$$

In addition, RSC-MDPs also possess benign properties similar to RMDPs such that for any policy $\pi$ and the robust optimal policy $\pi^{\mathsf{SC},\star}$, the corresponding *robust SC Bellman consistency equation* and *robust SC Bellman optimality equation* are also satisfied (specified in Appendix C.3.3).

**Goal.** Based on all the definitions and analysis above, this work aims to find an optimal policy for RSC-MDPs that maximizes the robust SC-value function in (5), yielding optimal performance in the worst case when the unobserved confounder distribution falls into an uncertainty set $\mathcal{U}^\sigma(P^c)$.

## 3.3 Advantages of RSC-MDPs over traditional RMDPs

The most relevant robust RL formulation to ours is RMDPs, which has been introduced in Section 2. Here, we provide a thorough comparison between RMDPs and our RSC-MDPs with theoretical justifications, and leave the comparisons and connections to other related formulations in Figure 2 and Appendix B.1 due to space limits.

To begin, at each time step $t$, RMDPs explicitly introduce uncertainty to the transition probability kernels, while our RSC-MDPs add uncertainty to the transition kernels in a latent (and hence more structured) manner via perturbing the unobserved confounder that partly determines the transition kernels. As an example, imagining the true uncertainty set encountered in the real world is illustrated as the blue region in Figure 3, which could have a complicated structure. Since the uncertainty set in RMDPs is homogeneous (illustrated by the green circles), one often faces the dilemma of being either too conservative (when $\sigma$ is large) or too reckless (when $\sigma$ is small). In contrast, the proposed

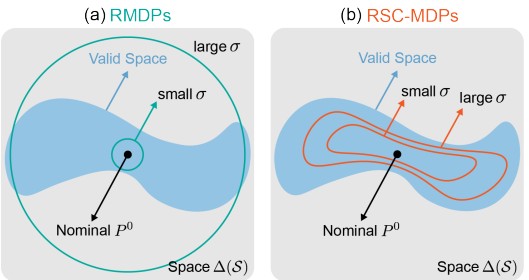

Figure 3: (a) RMDPs add homogeneous noise to states, while (b) RSC-MDPs perturb the confounder to influence states, resulting in a subset of the valid space.

RSC-MDPs – shown in Figure 3(b) – take advantage of the structured uncertainty set (illustrated by the orange region) enabled by the underlying SCM, which can potentially lead to much better estimation of the true uncertainty set. Specifically, the varying unobserved confounder induces diverse perturbation to different portions of the state through the structural causal function, enabling heterogeneous and structural uncertainty sets over the state space.

**Theoretical guarantees of RSC-MDPs: advantages of structured uncertainty.** To theoretically understand the advantages of the proposed robust formulation of RSC-MDPs with comparison to prior works, especially RMDPs, the following theorem verifies that RSC-MDPs enable additional robustness against semantic attack besides small model perturbation or noise considered in RMDPs. The proof is postponed to Appendix C.2.

**Theorem 2.** *Consider any $T \geq 2$. Consider some standard MDPs $\mathcal{M} = \{\mathcal{S}, \mathcal{A}, P^0, T, r\}$, equivalently represented as an SC-MDP $\mathcal{M}_{\text{sc}} = \{\mathcal{S}, \mathcal{A}, T, r, \mathcal{C}, \{\mathcal{P}_t^i\}, P^c\}$ with $\mathcal{C} := \{0, 1\}$, and total variation as the 'distance' function $\rho$ to measure the uncertainty set (the admissible uncertainty level obeys $\sigma \in [0, 1]$). For the corresponding RMDP $\mathcal{M}_{\text{rob}}$ with the uncertainty set $\mathcal{U}^{\sigma_1}(P^0)$, and the proposed RSC-MDP $\mathcal{M}_{\text{sc-rob}} = \{\mathcal{S}, \mathcal{A}, T, r, \mathcal{C}, \{\mathcal{P}_t^i\}, \mathcal{U}^{\sigma_2}(P^c)\}$, the optimal robust policy $\pi_{\text{RMDP}}^{\star,\sigma_1}$ associated with $\mathcal{M}_{\text{rob}}$ and $\pi_{\text{RSC}}^{\star,\sigma_2}$ associated with $\mathcal{M}_{\text{sc-rob}}$ obey: given $\sigma_2 \in \left(\frac{1}{2}, 1\right]$, there exist RSC-MDPs with some initial state distribution $\phi$ such that*

$$\widetilde{V}_1^{\pi_{\text{RSC}}^{\star,\sigma_2}, \sigma_2}(\phi) - \widetilde{V}_1^{\pi_{\text{RMDP}}^{\star,\sigma_1}, \sigma_2}(\phi) \geq \frac{T}{8}, \quad \forall \sigma_1 \in [0, 1]. \tag{6}$$

In words, Theorem 2 reveals a fact about the proposed RSC-MDPs: *RSC-MDPs could succeed in intense semantic attacks while RMDPs fail*. As shown by (6), when fierce semantic shifts appear between the training and test scenarios – perturbing the unobserved confounder in a large uncertainty set $\mathcal{U}^{\sigma_2}(P^c)$, solving RSC-MDPs with $\pi_{\text{RSC}}^{\star,\sigma_2}$ succeeds in testing while $\pi_{\text{RMDP}}^{\star,\sigma_1}$ trained by solving RMDPs can fail catastrophically. The proof is achieved by constructing hard instances of RSC-MDPs that RMDPs could not cope with due to inherent limitations. Moreover, this advantage of RSC-MDPs is consistent with and verified by the empirical performance evaluation in Section 5.3 **R1**.

## 4  An Empirical Algorithm to Solve RSC-MDPs: RSC-SAC

When addressing distributionally robust problems in RMDPs, the worst case is typically defined within a prescribed uncertainty set in a clear and implementation-friendly manner, allowing for iterative or analytical solutions. However, solving RSC-MDPs could be challenging as the structured uncertainty set is induced by the causal effect of perturbing the confounder. The precise characterization of this structured uncertainty set is difficult since neither the unobserved confounder nor the true causal graph of the observable variables is accessible, both of which are necessary for intervention or counterfactual reasoning. Therefore, we choose to approximate the causal effect of perturbing the confounder by learning from the data collected during training.

---

**Algorithm 1:** RSC-SAC Training

**Input:** policy $\pi$, data buffer $\mathcal{D}$, transition model $P_\theta$, ratio of modified data $\beta$

**for** $t \in [1, T]$ **do**
    Sample action $a_t \sim \pi(\cdot | s_t)$
    $(s_{t+1}, r_t) \leftarrow \text{Env}(s_t, a_t)$
    Add buffer $\mathcal{D} = \mathcal{D} \cup \{s_t, a_t, s_{t+1}, r_t\}$
    **for** *sample batch* $\mathcal{B} \in \mathcal{D}$ **do**
        Randomly select $\beta\%$ data in $\mathcal{B}$
        Modify $s_t$ in selected data with (7)
        $(\hat{s}_{t+1}, \hat{r}_t) \sim P_\theta(s_t, a_t, \mathcal{G}_\phi)$
        Replace data with $(s_t, a_t, \hat{s}_{t+1}, \hat{r}_t)$
        $\mathcal{L} = \|s_{t+1} - \hat{s}_{t+1}\|_2^2 + \|r_t - \hat{r}_t\|_2^2$
        Update $\theta$ and $\phi$ with $\mathcal{L} + \lambda \|\boldsymbol{G}\|_p$
        Update $\pi$ with SAC algorithm

---

In this section, we propose an intuitive yet effective empirical approach named RSC-SAC for solving RSC-MDPs, which is outlined in Algorithm 1. We first estimate the effect of perturbing the distribution $P^c$ of the confounder to generate new states (Section 4.1). Then, we learn the structural causal model $\mathcal{P}_t^i$ to predict rewards and the next states given the perturbed states (Section 4.2). By combining these two components, we construct a data generator capable of simulating novel transitions $(s_t, a_t, r_t, s_{t+1})$ from the structured uncertainty set. To learn the optimal policy, we construct the data buffer with a mixture of the original data and the generated data and then use the Soft Actor-Critic (SAC) algorithm [20] to optimize the policy.

### 4.1  Distribution of confounder

As we have no prior knowledge about the confounders, we choose to approximate the effect of perturbing them without explicitly estimating the distribution $P^c$. We first randomly select a dimension $i$ from the state $s_t$ to apply perturbation and then assign the dimension $i$ of $s_t$ with a heuristic rule. We select the value from another sample $s_k$ that has the most different value from $s_t$ in dimension $i$ and the most similar value to $s_t$ in the remaining dimensions. Formally, this process solves the following optimization problem to select sample $k$ from a batch of $K$ samples:

$$s_t^i \leftarrow s_k^i, \ k = \arg\max \frac{\|s_t^i - s_k^i\|_2^2}{\sum_{\neg i} \|s_t^{\neg i} - s_k^{\neg i}\|_2^2}, \ k \in \{1, ..., K\} \tag{7}$$

where $s_t^i$ and $s_t^{\neg i}$ means dimension $i$ of $s_t$ and other dimensions of $s_t$ except for $i$, respectively. Intuitively, permuting the dimension of two samples breaks the spurious correlation and remains the

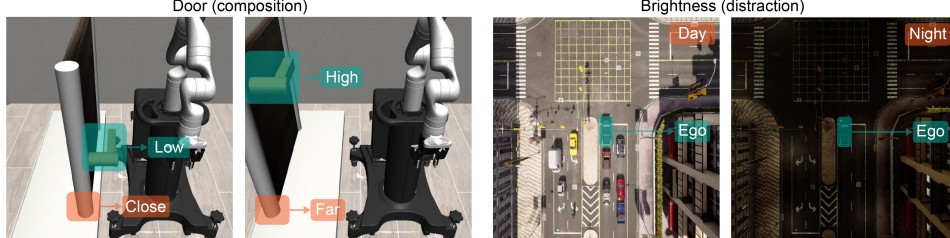

Figure 4: Two tasks used in experiments. *Door* is a composition task implemented in Robosuite with a spurious correlation between the positions of the handle and the door. *Brightness* is a distraction task implemented in Carla with a spurious correlation between the number vehicles and day/night.

most semantic meaning of the state space. However, this permutation sometimes also breaks the true cause and effect between dimensions, leading to a performance drop. The trade-off between robustness and performance [21] is a long-standing dilemma in the robust optimization framework, which we will leave to future work.

## 4.2 Learning of structural causal model

With the perturbed state $s_t$, we then learn an SCM to predict the next state and reward considering the effect of the action on the previous state. This model contains a causal graph to achieve better generalization to unseen state-action pairs. Specifically, we simultaneously learn the model parameter and discover the underlying causal graph in a fully differentiable way with $(\hat{s}_{t+1}, \hat{r}_t) \sim P_\theta(s_t, a_t, \mathcal{G}_\phi)$, where $\theta$ is the parameter of the neural network of the dynamic model and $\phi \in \mathbb{R}^{(n+d_\mathcal{A}) \times (n+1)}$ is the parameter to represent causal graph $\mathcal{G}$ between $\{s_t, a_t\}$ and $\{s_{t+1}, r_t\}$. This graph is represented by a binary adjacency matrix $\boldsymbol{G}$, where $1/0$ means the existence/absence of an edge. $P_\theta$ has an encoder-decoder structure with matrix $\boldsymbol{G}$ as an intermediate linear transformation. The encoder takes state and action in and outputs features $f_e \in \mathbb{R}^{(n+d_\mathcal{A}) \times d_f}$ for each dimension, where $d_f$ is the dimension of the feature. Then, the causal graph is multiplied to generate the feature for the decoder $f_d = f_e^T \boldsymbol{G} \in \mathbb{R}^{d_f \times (n+1)}$. The decoder takes in $f_d$ and outputs the next state and reward. The detailed architecture of this causal transition model can be found in Appendix D.1.

The objective for training this model consists of two parts, one is the supervision signal from collected data $\|s_{t+1} - \hat{s}_{t+1}\|_2^2 + \|r_t - \hat{r}_t\|_2^2$, and the other is a penalty term $\lambda\|\boldsymbol{G}\|_p$ with weight $\lambda$ to encourage the sparsity of the matrix $\boldsymbol{G}$. The penalty is important to break the spurious correlation between dimensions of state since it forces the model to eliminate unnecessary inputs for prediction.

## 5 Experiments and Evaluation

In this section, we first provide a benchmark consisting of eight environments with spurious correlations, which may be of independent interest to robust RL. Then we evaluate the proposed algorithm RSC-SAC with comparisons to prior robust algorithms in RL.

### 5.1 Tasks with spurious correlation

To the best of our knowledge, no existing benchmark addresses the issues of spurious correlation in the state space of RL. To bridge the gap, we design a benchmark consisting of eight novel tasks in self-driving and manipulation domains using the Carla [22] and Robosuite [23] platforms (shown in Figure 4). Tasks are designed to include spurious correlations in terms of human common sense, which is ubiquitous in decision-making applications and could cause safety issues. We categorize the tasks into *distraction correlation* and *composition correlation* according to the type of spurious correlation. We specify these two types of correlation below and leave the full descriptions of the tasks in Appendix D.2.

- **Distraction correlation** is between task-relevant and task-irrelevant portions of the state. The task-irrelevant part could distract the policy model from learning important features and lead to a performance drop. A typical method to avoid distraction is background augmentation [24, 25]. We design four tasks with this category of correlation, i.e., *Lift*, *Wipe*, *Brightness*, and *CarType*.
- **Composition correlation** is between two task-relevant portions of the state. This correlation usually exists in compositional generalization, where states are re-composed to form novel tasks during testing. Typical examples are multi-task RL [26, 27] and hierarchical RL [28, 29]. We design four tasks with this category of correlation, i.e., *Stack*, *Door*, *Behavior*, and *Crossing*.

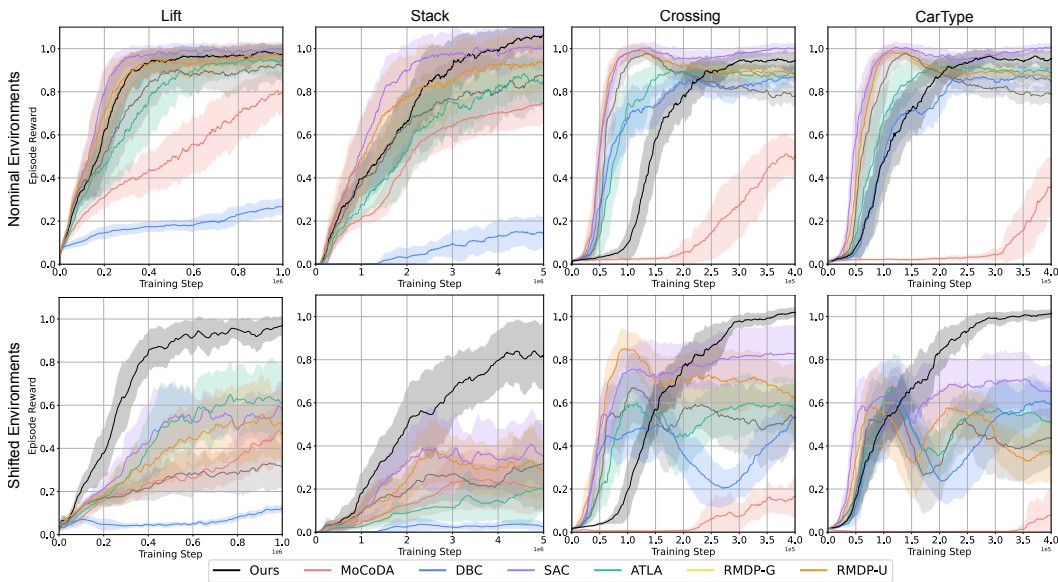

Figure 5: The first row shows the testing reward on the nominal environments, while the second row shows the testing reward on the shifted environments.

Table 1: Testing reward on shifted environments. Bold font means the best reward.

| Method | Brightness | Behavior | Crossing | CarType | Lift | Stack | Wipe | Door |
|---|---|---|---|---|---|---|---|---|
| SAC | $0.56{\pm}0.13$ | $0.13{\pm}0.03$ | $0.81{\pm}0.13$ | $0.63{\pm}0.14$ | $0.58{\pm}0.13$ | $0.26{\pm}0.12$ | $0.16{\pm}0.20$ | $0.08{\pm}0.07$ |
| RMDP-G | $0.55{\pm}0.15$ | $0.16{\pm}0.04$ | $0.47{\pm}0.13$ | $0.53{\pm}0.16$ | $0.31{\pm}0.08$ | $0.33{\pm}0.15$ | $0.06{\pm}0.17$ | $0.07{\pm}0.03$ |
| RMDP-U | $0.54{\pm}0.19$ | $0.13{\pm}0.05$ | $0.60{\pm}0.15$ | $0.39{\pm}0.13$ | $0.51{\pm}0.17$ | $0.23{\pm}0.11$ | $0.06{\pm}0.17$ | $0.10{\pm}0.13$ |
| MoCoDA | $0.50{\pm}0.14$ | $0.16{\pm}0.05$ | $0.22{\pm}0.14$ | $0.23{\pm}0.12$ | $0.46{\pm}0.14$ | $0.29{\pm}0.11$ | $0.01{\pm}0.24$ | $0.09{\pm}0.14$ |
| ATLA | $0.48{\pm}0.11$ | $0.14{\pm}0.03$ | $0.61{\pm}0.14$ | $0.52{\pm}0.14$ | $0.61{\pm}0.18$ | $0.21{\pm}0.12$ | $0.29{\pm}0.18$ | $0.28{\pm}0.19$ |
| DBC | $0.52{\pm}0.18$ | $0.16{\pm}0.03$ | $0.68{\pm}0.12$ | $0.45{\pm}0.10$ | $0.12{\pm}0.02$ | $0.03{\pm}0.02$ | $0.19{\pm}0.35$ | $0.01{\pm}0.01$ |
| Active | $0.47{\pm}0.14$ | $0.14{\pm}0.03$ | $0.83{\pm}0.09$ | $0.77{\pm}0.14$ | $0.35{\pm}0.09$ | $0.24{\pm}0.12$ | $0.17{\pm}0.17$ | $0.05{\pm}0.02$ |
| RSC-SAC | $\mathbf{0.99{\pm}0.11}$ | $\mathbf{1.02{\pm}0.09}$ | $\mathbf{1.04{\pm}0.02}$ | $\mathbf{1.03{\pm}0.02}$ | $\mathbf{0.98{\pm}0.04}$ | $\mathbf{0.77{\pm}0.20}$ | $\mathbf{0.85{\pm}0.12}$ | $\mathbf{0.61{\pm}0.17}$ |

## 5.2 Baselines

Robustness in RL has been explored in terms of diverse uncertainty set over state, action, or transition kernels. Regarding this, we use a non-robust RL and four representative algorithms of robust RL as baselines, all of which are implemented on top of the SAC [20] algorithm. **Non-robust RL (SAC):** This serves as a basic baseline without considering any robustness during training; **Solving robust MDP:** We generate the samples to cover the uncertainty set over the state space by adding perturbation around the nominal states that follows two distribution, i.e., uniform distribution (RMDP-U) and Gaussian distribution (RMDP-G). **Solving SA-MDP:** We compare ATLA [6], a strong algorithm that generates adversarial states using an optimal adversary within the uncertainty set. **Invariant feature learning:** We choose DBC [30] that learns invariant features using the bi-simulation metric [31] and [32] (Active) that actively sample uncertain transitions to reduce causal confusion. **Counterfactual data augmentation:** We select MoCoDA [33], which identifies local causality to switch components and generate counterfactual samples to cover the targeted uncertainty set. We adapt this algorithm using an approximate causal graph rather than the true causal graph.

## 5.3 Results and Analysis

To comprehensively evaluate the performance of the proposed method RSC-SAC, we conduct experiments to answer the following questions: **Q1.** Can RSC-SAC eliminate the harmful effect of spurious correlation in learned policy? **Q2.** Does the robustness of RSC-SAC only come from the sparsity of model? **Q3.** How does RSC-SAC perform in the nominal environments compared to non-robust algorithms? **Q4.** Which module is critical in our empirical algorithm? **Q5.** Is RSC-SAC robust to other types of uncertainty and model perturbation? **Q6.** How does RSC-SAC balance the tradeoff between performance and robustness? We analyze the results and answer these questions in the following.

Table 2: Testing reward on nominal environments. Underline means the reward is over 0.9.

| Method | Brightness | Behavior | Crossing | CarType | Lift | Stack | Wipe | Door |
|---|---|---|---|---|---|---|---|---|
| SAC | 1.00±0.09 | 1.00±0.08 | 1.00±0.02 | 1.00±0.03 | 1.00±0.03 | 1.00±0.09 | 1.00±0.12 | 1.00±0.03 |
| RMDP-G | 1.04±0.09 | 1.00±0.11 | 0.78±0.05 | 0.79±0.05 | 0.92±0.07 | 0.86±0.14 | 0.99±0.13 | 0.99±0.06 |
| RMDP-U | 1.02±0.09 | 1.04±0.07 | 0.90±0.03 | 0.88±0.03 | 0.97±0.05 | 0.92±0.12 | 0.97±0.14 | 0.88±0.31 |
| MoCoDA | 0.65±0.17 | 0.78±0.15 | 0.57±0.07 | 0.55±0.13 | 0.79±0.11 | 0.72±0.08 | 0.69±0.13 | 0.41±0.22 |
| ATLA | 0.99±0.11 | 0.98±0.11 | 0.89±0.05 | 0.88±0.04 | 0.94±0.08 | 0.88±0.10 | 0.96±0.12 | 0.97±0.05 |
| DBC | 0.75±0.12 | 0.78±0.10 | 0.85±0.08 | 0.86±0.06 | 0.27±0.04 | 0.12±0.08 | 0.31±0.21 | 0.01±0.01 |
| Active | 1.02±0.10 | 1.08±0.06 | 1.00±0.02 | 1.00±0.02 | 0.99±0.03 | 0.90±0.12 | 0.93±0.20 | 0.99±0.05 |
| RSC-SAC | 0.92±0.31 | 1.06±0.07 | 0.96±0.03 | 0.96±0.03 | 0.96±0.05 | 1.04±0.08 | 0.92±0.14 | 0.98±0.05 |

**R1.** RSC-SAC **is robust against spurious correlation.** The testing results of our proposed method with comparisons to the baselines are presented in Table 1, where the rewards are normalized by the episode reward of SAC in the nominal environment. The results reveal that RSC-SAC significantly outperforms other baselines in shifted test environments, exhibiting comparable performance to that of vanilla SAC on the nominal environment in 5 out of 8 tasks. An interesting and even surprising finding, as shown in Table 1, is that although RMDP-G, RMDP-U, and ATLA are trained desired to be robust against small perturbations, their performance tends to drop more than non-robust SAC in some tasks. This indicates that using the samples generated from the traditional robust algorithms could harm the policy performance when the test environment is outside of the prescribed uncertainty set considered in the robust algorithms.

**R2. Sparsity of the model is only one reason for the robustness of** RSC-SAC. As existing literature shows [34], sparsity regularization benefits the elimination of spurious correlation and causal confusion. Therefore, we compare our method with a sparse version of SAC (SAC-Sparse): we add an additional penalty $\alpha|W|_1$ during the optimization, where $W$ is the parameter of the first linear layer of the policy and value networks and $\alpha$ is the weight. The results of both *Distraction* and *Composition* are shown in Figure 6. We have two important findings based on the results: (1) The sparsity improves the robustness of SAC in the setting of distraction spurious correlation, which is consistent with the findings in [34]. (2) The sparsity does

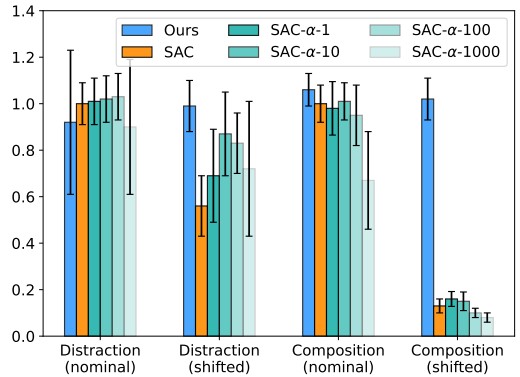

Figure 6: Comparison between SAC-Sparse and our method. $\alpha$ is the weight of regularization.

not help with the composition type of spurious correlation, which indicates that purely using sparsity regularization cannot explain the improvement of our RSC-SAC. In fact, the semantic perturbation in our method plays an important role in augmenting the composition generalization.

**R3.** RSC-SAC **maintains great performance in the nominal environments.** Previous literature [21] finds out that there usually exists a trade-off between the performance in the nominal environment and the robustness against uncertainty. To evaluate the performance of RSC-SAC in the nominal environment, we conduct experiments and summarize results in Table 2, which shows that RSC-SAC still performs well in the training environment. Additionally, the training curves are displayed in Figure 5, showing that RSC-SAC achieves similar rewards compared to non-robust SAC although converges slower than it.

**R4. Both the distribution of confounder and the structural causal model are critical.** To assess the impact of each module in our algorithm, we conduct three additional ablation studies (in Table 3), where we remove the causal graph $\boldsymbol{G}_\phi$, the transition model $P_\theta$, and the distribution of the confounder $P^c$ respectively. The results demonstrate

Table 3: Influence of modules

| Method | Lift | Behavior | Crossing |
|---|---|---|---|
| w/o $\boldsymbol{G}_\phi$ | 0.79±0.15 | 0.51±0.24 | 0.87±0.10 |
| w/o $P_\theta$ | 0.75±0.13 | 0.41±0.28 | 0.89±0.08 |
| w/o $P^c$ | 0.90±0.09 | 0.66±0.21 | 0.96±0.04 |
| Full model | 0.98±0.04 | 1.02±0.09 | 1.04±0.02 |

that the learnable causal graph $\boldsymbol{G}_\phi$ is critical for the performance that enhances the prediction of

the next state and reward, thereby facilitating the generation of high-quality next states with current perturbed states. The transition model without $G_\phi$ may still retain numerous spurious correlations, resulting in a performance drop similar to the one without $P_\theta$, which does not alter the next state and reward. In the third row of Table 3, the performance drop indicates that the confounder $P^c$ also plays a crucial role in preserving semantic meaning and avoiding policy training distractions.

**R5.** RSC-SAC **is also robust to random pertur-
bation.** The final investigation aims to assess the generalizability of our method to cope with random perturbation that is widely considered in robust RL (RMDPs). Towards this, we evaluate the proposed algorithm in the test environments added with random noise under the Gaussian distribution with two varying scales in the *Lift* environment. In Table 4,

Table 4: Random purterbuation

| Method | Lift-0 | Lift-0.01 | Lift-0.1 |
|---|---|---|---|
| SAC | 1.00±0.03 | 0.77±0.13 | 0.46±0.23 |
| RMDP-0.01 | 0.97±0.05 | 0.96±0.06 | 0.51±0.21 |
| RMDP-0.1 | 0.85±0.12 | 0.82±0.09 | 0.39±0.15 |
| RSC-SAC | 0.96±0.05 | 0.94±0.06 | 0.44±0.18 |

*Lift-0* indicates the nominal training environment, while *Lift-0.01* and *Lift-0.1* represent the environments perturbed by the Gaussian noise with standard derivation 0.01 and 0.1, respectively. The results indicate that our RSC-SAC achieves comparable robustness compared to RMDP-0.01 in both large and small perturbation settings and outperforms RMDP methods in the nominal training environment.

**R6.** RSC-SAC **keeps good performance
and robustness for a wide range of** $\beta$**.** As shown in Figure 7, the proposed RSC-SAC performs well in both nominal and shifted settings – keeping good performance in the nominal setting and achieving robustness, for a large range of (20%-70%). When the ratio of perturbed data is very small (1%), RSC-SAC almost achieves the same results as vanilla SAC in nominal settings and there is no robustness in shifted set-

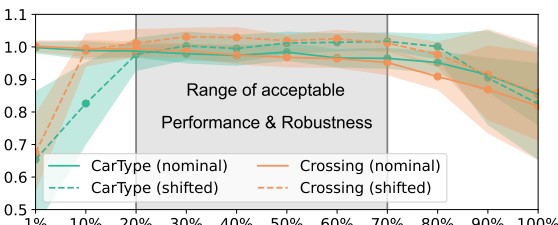

Figure 7: Performance-robustness tradeoff with different augmentation ratio $\beta$.

tings. As it increases (considering more robustness), the performance of RSC-SAC in the nominal setting gradually gets worse, while reversely gets better in the shifted settings (more robust). However, when the ratio is too large (>80%), the performances of RSC-SAC in both settings degrade a lot, since the policy is too conservative so that fails in all environments.

## 6 Conclusion and Limitation

This work focuses on robust reinforcement learning against spurious correlation in state space, which broadly exists in (sequential) decision-making tasks. We propose robust SC-MDPs as a general framework to break spurious correlations by perturbing the value of unobserved confounders. We not only theoretically show the advantages of the framework compared to existing robust works in RL, but also design an empirical algorithm to solve robust SC-MDPs by approximating the causal effect of the confounder perturbation. The experimental results demonstrate that our algorithm is robust to spurious correlation – outperforms the baselines when the value of the confounder in the test environment derivates from the training one. It is important to note that the empirical algorithm we propose is evaluated only for low-dimensional states.However, the entire framework can be extended in the future to accommodate high-dimensional states by leveraging powerful generative models with disentanglement capabilities [35] and state abstraction techniques [36].

## Acknowledgments and Disclosure of Funding

The work of W. Ding is supported by the Qualcomm Innovation Fellowship. The work of L. Shi and Y. Chi is supported in part by the grants ONR N00014-19-1-2404, NSF CCF-2106778, DMS-2134080, and CNS-2148212. L. Shi is also gratefully supported by the Leo Finzi Memorial Fellowship, Wei Shen and Xuehong Zhang Presidential Fellowship, and Liang Ji-Dian Graduate Fellowship at Carnegie Mellon University.

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

# Appendix

## Table of Contents

## A   Broader Impact

Incorporating causality into reinforcement learning methods increases the interpretability of artificial intelligence, which helps humans understand the underlying mechanism of algorithms and check the source of failures. However, the learned causal transition model may contain human-readable private information about the environment, which could raise privacy issues. To mitigate this potential negative societal impact, the causal transition model needs to be encrypted and only accessible to algorithms and trustworthy users.

## B   Other Related Works

In this section, besides the most related formulation, robust RL introduced in Sec 3.3, we also introduce some other related RL problem formulations partially shown in Figure 3. Then, we limit our discussion to mainly two lines of work that are related to ours: (1) promoting robustness in RL; (2) concerning the spurious correlation issues in RL.

### B.1   Related RL formulations

**Robustness to noisy state: POMDPs and SA-MDPs.** State-noisy MDPs refer to the RL problem that the agent can only access and choose the action based on a noisy observation rather than the true state at each step, including two existing types of problems: Partially observable MDPs (POMDPs) and state-adversarial MDPs (SA-MDPs), shown in Figure 3(b). In particular, at each step $t$, in POMDPs, the observation $o_t$ is generated by a fixed probability transition $\mathcal{O}(\cdot \mid s_t)$ (we refer to the case that $o_t$ only depends on the state $s_t$ but not action); for state-adversarial MDPs, the observation is an adversary $\nu(s_t)$ against and thus determined by the conducted policy, leading to the worst performance by perturbing the state in a small set around itself. To defend the state perturbation, both POMDPs, and SA-MDPs are indeed robust to the noisy observation, or called agent-observed state, but not the real state that transitions to the environment and next steps. In contrast, our RSC-MDPs

propose the robustness to the real state shift that will directly transition to the next state in the environment, involving additional challenges induced by the appearance of out-of-distribution states.

**Robustness to unobserved confounder: MDPUC and confounded MDPs.** To address the misleading spurious correlations hidden in components of RL, people formulate RL problems as MDPs with some additional components – unobserved confounders. In particular, the Markov decision process with unobserved confounders (MDPUC) [37] serves as a general framework to concern all types of possible spurious correlations in RL problems – at each step, the state, action, and reward are all possibly influenced by some unobserved confounder, shown in Figure 2(d); confounded MDPs [19] mainly concerns the misleading correlation between the current action and the next state, illustrated in Figure 3(e). The proposed state-confounded MDPs (SC-MDPs) can be seen as a specified type of MDPUC that focuses on breaking the spurious correlation between different parts of the state space itself (different from confounded MDPs which consider the correlation between action and next state), motivated by various real-world applications in self-driving and control tasks. In addition, the proposed formulation is more flexible and can work in both online and offline RL settings.

**Contexual MDPs (CDMPs).** A contextual MDP (CMDP) [38] is basically a set of standard MDPs sharing the same state and action space but specified by different contexts within a context space. In particular, the transition kernel, reward, and action of a CMDP are all determined by a (possibly unknown) fixed context. The proposed robust state-confounded MDPs (RSC-MDPs) are similar to CMDPs if we cast the unobserved confounder as the context in CMDPs, while different in two aspects: (1) In a CMDP, the context is fixed throughout an episode, while the unobserved confounder in RSC-MDPs can vary as $\{c_t\}_{1 \leq t \leq T}$; (2) In the online setting, the goal of CMDP is to beat the optimal policy depending on the context, while RSC-MDPs seek to learn the optimal policy that does not depend on the confounder $\{c_t\}_{1 \leq t \leq T}$.

## B.2 Related literature of robustness in RL

**Robust RL (robust MDPs).** Concerning the robust issues in RL, a large portion of works focus on robust RL with explicit uncertainty of the transition kernel, which is well-posed and a natural way to consider the uncertainty of the environment [13, 39–48]. However, to define the uncertainty set for the environment, most existing works use task structure-agnostic and heuristic 'distance' such as R-contamination, KL divergence, $\chi^2$, and total variation [49, 50, 14, 51, 52, 15, 49, 53–58, 49, 59, 48, 60–64] to measure the shift between the training and test transition kernel, leading to a homogeneous (almost structure-free) uncertainty set around the state space. In contrast, we consider a more general uncertainty set that enables the robustness to a task-dependent heterogeneous uncertainty set shaped by unobserved confounder and causal structure, in order to break the spurious correlation hidden in different parts of the state space.

**Robustness in RL.** Despite the remarkable success that standard RL has achieved, current RL algorithms are still limited since the agent is vulnerable if the deployed environment is subject to uncertainty and even structural changes. To address these challenges, a recent line of RL works begins to concern robustness to the uncertainty or changes over different components of MDPs – state, action, reward, and transition kernel, where a review [8] can be referred to. Besides robust RL framework concerning the shift of the transition kernel and reward, to promote robustness in RL, there exist various works [11, 12] that consider the robustness to action uncertainty, i.e., the deployed action in the environment is distorted by an adversarial agent smoothly or circumstantially; some works [9, 6, 10, 65–67] investigate the robustness to the state uncertainty including but not limited to the introduced POMDPs and SA-MDPs in Appendix B.1, where the agent chooses the action based on observation – the perturbed state determined by some restricted noise or adversarial attack. The proposed RSC-MDPs can be regarded as addressing the state uncertainty since the shift of the unobserved confounder leads to state perturbation. In contrast, RSC-MDPs consider the out-of-distribution of the real state that will directly influence the subsequent transition in the environment, but not the observation in POMDPs and SA-MDPs that will not directly influence the environment.

## B.3 Related literature of spurious correlation in RL

**Confounder in RL.** These works mainly focus on the confounder between action (treatment) and state (effect), which is a long-standing problem that exists in the causal inference area. However,

we find that the confounder may cause problems from another perspective, where the confounder is built upon different dimensions of the state variable. Some people focus on the confounder between action and state, which is common in offline settings since the dataset is fixed and intervention is not allowed. But in the online setting, actions are controlled by an agent and intervention is available to eliminate spurious correlation. [68] reduces the spurious correlation between action and state in the offline setting. [69] deal with environment-irrelevant white noise; possible shift + causal [70]. The confounder problem is usually easy to solve since agents can interact with the environment to do interventions. However, different from most existing settings, we find that even with the capability of intervention, the confounding between dimensions in states cannot be fully eliminated. Then the learned policy is heavily influenced if these confounders change during testing.

**Invariant Feature learning.** The problem of spurious correlation has attracted attention in the supervised learning area for a long time and many solutions are proposed to learn invariant features to eliminate spurious correlations. A general framework to remedy the ignorance of spurious correlation in empirical risk minimization (ERM) is invariant risk minimization (IRM) [71]. Other works tackle this problem with group distributional robustness [72], adversarial robustness [73], and contrastive learning [74]. These methods are also adapted to sequential settings. The idea of increasing the robustness of RL agents by training agents on multiple environments has been shown in previous works [75, 30, 30]. However, a shared assumption among these methods is that multiple environments with different values of confounder are accessible, which is not always true in the real world.

**Counterfactual Data Augmentation in RL.** One way to simulate multiple environments is data augmentation. However, most data augmentation works [24, 76, 25, 77–80] apply image transformation to raw inputs, which requires strong domain knowledge for image manipulation and cannot be applied to other types of inputs. In RL, the dynamic model and reward model follow certain causal structures, which allow counterfactual generation of new transitions based on the collected samples. This line of work, named counterfactual data augmentation, is very close to this work. Deep generative models [81] and adversarial examples [82] are considered for the generation to improve sample efficiency in model-based RL. CoDA [83] and MocoDA [33] leverage the concept of locally factored dynamics to randomly stitch components from different trajectories. However, the assumption of local causality may be limited.

**Domain Randomization.** If we are allowed to control the data generation process, e.g., the underlying mechanism of the simulator, we can apply the golden rule in causality – Randomized Controlled Trial (RCT). The well-known technique, domain randomization [84], exactly follows the idea of RCT, which randomly perturbs the internal state of the experiment in simulators. Later literature follows this direction and develops variants including randomization guided by downstream tasks in the target domain [85, 86], randomization to match real-world distributions [87, 88], and randomization to minimize data divergence [89]. However, it is usually impossible to randomly manipulate internal states in most situations in the real world. In addition, determining which variables to randomize is even harder given so many factors in complex systems.

**Discovering Spurious Correlations** Detecting spurious correlations helps models remove features that are harmful to generalization. Usually, domain knowledge is required to find such correlations [90–92]. However, when prior knowledge is accessible, techniques such as clustering can also be used to reveal spurious attributes [37, 93, 94]. When human inspection is available, recent works [95–97] also use explainability techniques to find spurious correlations. Another area for discovery is concept-level and interactive debugging [98, 99], which leverage concepts or human feedback to perform debugging.

# C  Theoretical Analyses

## C.1  Proof of Theorem 1

In this section, we verify the existence of an optimal policy of the proposed RSC-MDPs, involving additional components — confounder $C_s$ and the infimum optimization problems with comparisons to standard MDPs [100].

To begin with, we recall that the goal is to find a policy $\widetilde{\pi} = \{\widetilde{\pi}_t\}_{1 \leq t \leq T} \in \Pi$ such that for all $(s,a) \in \mathcal{S} \times \mathcal{A}$:

$$\widetilde{V}_t^{\widetilde{\pi},\sigma}(s) = \widetilde{V}_t^{\star,\sigma}(s) := \sup_{\pi \in \Pi} \widetilde{V}_t^{\pi,\sigma}(s) \quad \text{and} \quad \widetilde{Q}_t^{\widetilde{\pi},\sigma}(s,a) = \widetilde{Q}_t^{\star,\sigma}(s,a) := \sup_{\pi \in \Pi} \widetilde{Q}_t^{\pi,\sigma}(s,a), \quad (8)$$

which we called an optimal policy. Towards this, we start from the first claim in (8).

**Step 1: Introducing additional notations.** Before proceeding, we let $\{S_t, A_t, R_t, C_t\}$ denote the random variables — state, action, reward, and confounder, at time step $t$ for all $1 \leq t \leq T$. Then invoking the Markov properties, we know that conditioned on current state $s_t$, the future state, action, and reward are all independent from the previous $s_1, a_1, r_1, c_1, \cdots, s_{t-1}, a_{t-1}, r_{t-1}, c_{t-1}$. In addition, we represent $P_t \in \Delta(\mathcal{C})$ as some distribution of confounder at time step $t$, for all $1 \leq t \leq T$. For convenience, we introduce the following notation that is defined over time step $t \leq k \leq T$:

$$\forall 1 \leq t \leq T: \quad P_{+t} := \otimes_{t \leq k \leq T} P_k \quad \text{and} \quad \mathcal{U}^\sigma(P_{+t}^c) := \otimes_{t \leq k \leq T} \mathcal{U}^\sigma(P_k^c), \quad (9)$$

which represent some collections of variables from time step $t$ to the end of the episode. In addition, recall that the transition kernel from time step $t$ to $t+1$ is denoted as $s_{t+1}^i \sim \mathcal{P}_t^i(\cdot \mid s_t, a_t, c_t)$ for $i \in 1, 2 \cdots, n$. With slight abuse of notation, we denote $s_{t+1} \sim \mathcal{P}_t(\cdot \mid s_t, a_t, c_t)$ and abbreviate $\mathbb{E}_{s_{t+1} \sim \mathcal{P}_t(\cdot \mid s_t, a_t, c_t)}[\cdot]$ as $\mathbb{E}_{s_{t+1}}[\cdot]$ whenever it is clear.

**Step 2: Establishing recursive relationship.** Recall that the nominal distribution of the confounder is $c_t \in P_t^c$ at time step $t$. We choose $\widetilde{\pi} = \{\widetilde{\pi}_t\}$ which obeys: for all $1 \leq t \leq T$,

$$\widetilde{\pi}_t(s) := \arg\max_{\pi_t \in \Delta(\mathcal{A})} \left\{ \mathbb{E}_{\pi_t}[r_t(s,a_t)] + \inf_{P_t \in \mathcal{U}^\sigma(P_t^c)} \mathbb{E}_{\pi_t}\left[ \mathbb{E}_{c_t \sim P_t}\left[ \mathbb{E}_{s_{t+1}}\left[ \widetilde{V}_{t+1}^{\star,\sigma}(s_{t+1}) \right] \right] \right] \right\}. \quad (10)$$

Armed with these definitions and notations, for any $(t,s) \in \{1, 2, \cdots, T\} \times \mathcal{S}$, one has

$$\widetilde{V}_t^{\star,\sigma}(s)$$

$$\overset{(i)}{=} \sup_{\pi \in \Pi} \inf_{P \in \mathcal{U}^\sigma(P^c)} \widetilde{V}_t^{\pi,P}(s) \overset{(ii)}{=} \sup_{\pi \in \Pi} \inf_{P_{+t} \in \mathcal{U}^\sigma(P_{+t}^c)} \mathbb{E}_{\pi, P_{+t}}\left[ \sum_{k=t}^{T} r_k(s_k, a_k) \right]$$

$$\overset{(iii)}{=} \sup_{\pi \in \Pi} \inf_{P_{+t} \in \mathcal{U}^\sigma(P_{+t}^c)} \mathbb{E}_{\pi_t}\Bigg[ r_t(s, a_t)$$

$$+ \mathbb{E}_{c_t \sim P_t}\left[ \mathbb{E}_{s_{t+1}}\left[ \sum_{k=t+1}^{T} r_k(s_k, a_k) \mid \pi, P_{+(t+1)}, (S_t, A_t, R_t, C_t, S_{t+1}) = (s, a_t, r_t, c_t, s_{t+1}) \right] \right] \Bigg]$$

$$= \sup_{\pi \in \Pi} \mathbb{E}_{\pi_t}[r_t(s, a_t)] + \inf_{P_{+t} \in \mathcal{U}^\sigma(P_{+t}^c)} \mathbb{E}_{\pi_t}\Bigg[ \mathbb{E}_{c_t \sim P_t}\Bigg[$$

$$\mathbb{E}_{s_{t+1}}\left[ \sum_{k=t+1}^{T} r_k(s_k, a_k) \mid \pi, P_{+(t+1)}, (S_t, A_t, R_t, C_t, S_{t+1}) = (s, a_t, r_t, c_t, s_{t+1}) \right] \Bigg] \Bigg]$$

where (i) holds by the definitions in (5), (ii) is due to (3) and that $\widetilde{V}_t^{\pi,P}(s)$ only depends on $P_{+t}$ by the Markov property, (iii) follows from expressing the term of interest by moving one step ahead and $\mathbb{E}_{\pi_t}$ is taken with respect to $a_t \sim \pi_t(\cdot \mid S_t = s)$.

To continue, we observe that the $\widetilde{V}_t^{\star,\sigma}(s)$ can be further controlled as follows:

$$\widetilde{V}_t^{\star,\sigma}(s)$$

$$= \sup_{\pi \in \Pi} \mathbb{E}_{\pi_t}[r_t(s, a_t)] + \inf_{P_{+t} \in \mathcal{U}^\sigma(P_{+t}^c)} \mathbb{E}_{\pi_t}\Bigg[ \mathbb{E}_{c_t \sim P_t}\Bigg[$$

$$\mathbb{E}_{s_{t+1}}\left[ \sum_{k=t+1}^{T} r_k(s_k, a_k) \mid \pi, P_{+(t+1)}, (S_t, A_t, R_t, C_t, S_{t+1}) = (s, a_t, r_t, c_t, s_{t+1}) \right] \Bigg] \Bigg]$$

$$\overset{(i)}{=} \sup_{\pi \in \Pi} \mathbb{E}_{\pi_t}[r_t(s, a_t)] + \inf_{P_t \in \mathcal{U}^\sigma(P_t^c)} \mathbb{E}_{\pi_t} \left[ \mathbb{E}_{c_t \sim P_t} \left[ \mathbb{E}_{s_{t+1}} \right. \right.$$

$$\left. \left. \inf_{P_{+(t+1)} \in \mathcal{U}^\sigma\left(P_{+(t+1)}^c\right)} \sum_{k=t+1}^{T} r_k(s_k, a_k) \mid \pi, P_{+(t+1)}, (S_t, A_t, R_t, C_t, S_{t+1}) = (s, a_t, r_t, c_t, s_{t+1}) \right] \right] \right]$$

$$\leq \sup_{\pi \in \Pi} \mathbb{E}_{\pi_t}[r_t(s, a_t)] + \inf_{P_t \in \mathcal{U}^\sigma(P_t^c)} \mathbb{E}_{\pi_t} \left[ \mathbb{E}_{c_t \sim P_t} \mathbb{E}_{s_{t+1}} \left[ \sup_{\pi' \in \Pi} \inf_{P_{+(t+1)} \in \mathcal{U}^\sigma(P_{+(t+1)}^c)} \right. \right.$$

$$\left. \left. \sum_{k=t+1}^{T} r_k(s_k, a_k) \mid \pi', P_{+(t+1)}, (S_t, A_t, R_t, C_t, S_{t+1}) = (s, a_t, r_t, c_t, s_{t+1}) \right] \right]$$

$$\overset{(ii)}{=} \sup_{\pi \in \Pi} \mathbb{E}_{\pi_t}[r_t(s, a_t)]$$

$$+ \inf_{P_t \in \mathcal{U}^\sigma(P_t^c)} \mathbb{E}_{\pi_t} \left[ \mathbb{E}_{c_t \sim P_t} \left[ \mathbb{E}_{s_{t+1}} \left[ \sup_{\pi' \in \Pi} \inf_{P_{+(t+1)} \in \mathcal{U}^\sigma(P_{+(t+1)}^c)} \mathbb{E}_{\pi', P_{+(t+1)}} \left[ \sum_{k=t+1}^{T} r_k(s_k, a_k) \right] \right] \right] \right]$$

$$= \sup_{\pi \in \Pi} \left\{ \mathbb{E}_{\pi_t}[r_t(s, a_t)] + \inf_{P_t \in \mathcal{U}^\sigma(P_t^c)} \mathbb{E}_{\pi_t} \left[ \mathbb{E}_{c_t \sim P_t} \left[ \mathbb{E}_{s_{t+1}} \left[ \widetilde{V}_{t+1}^{\star,\sigma}(s_{t+1}) \right] \right] \right] \right\}$$

$$= \sup_{\pi_t \in \Delta(\mathcal{A})} \left\{ \mathbb{E}_{\pi_t}[r_t(s, a_t)] + \inf_{P_t \in \mathcal{U}^\sigma(P_t^c)} \mathbb{E}_{\pi_t} \left[ \mathbb{E}_{c_t \sim P_t} \left[ \mathbb{E}_{s_{t+1}} \left[ \widetilde{V}_{t+1}^{\star,\sigma}(s_{t+1}) \right] \right] \right] \right\}$$

$$= \inf_{P_t \in \mathcal{U}^\sigma(P_t^c)} \mathbb{E} \left[ r_t(s, a_t) + \mathbb{E}_{c_t \sim P_t} \mathbb{E}_{s_{t+1}} \left[ \left[ \widetilde{V}_{t+1}^{\star,\sigma}(s_{t+1}) \right] \mid a_t = \widetilde{\pi}_t(s) \right] \right], \tag{11}$$

where (i) holds by the operator $\inf_{P_{+(t+1)} \in \mathcal{U}^\sigma\left(P_{+(t+1)}^c\right)}$ is independent from $\pi_t$ conditioned on a fixed distribution of $s_{t+1}$, (ii) arises from the Markov property such that the rewards $\{r_k(s_k, a_k)\}_{t+1 \leq k \leq T}$ conditioned on $(S_t, A_t, R_t, C_t, S_{t+1})$ or $S_{t+1}$ are the same, and the last equality follows from the definition of $\widetilde{\pi}$ in (10).

**Step 3: Completing the proof by applying recursion.**

Applying (11) recursively for $t + 1, \cdots T$, we arrive at

$$\widetilde{V}_t^{\star,\sigma}(s) \leq \inf_{P_t \in \mathcal{U}^\sigma(P_t^c)} \mathbb{E} \left[ r_t(s, a_t) + \mathbb{E}_{c_t \sim P_t} \mathbb{E}_{s_{t+1}} \left[ \left[ \widetilde{V}_{t+1}^{\star,\sigma}(s_{t+1}) \right] \mid a_t = \widetilde{\pi}_t(s) \right] \right]$$

$$\leq \inf_{P_t \in \mathcal{U}^\sigma(P_t^c)} \inf_{P_{t+1} \in \mathcal{U}^\sigma(P_{t+1}^c)} \mathbb{E} \left[ r_t(s, a_t) + \mathbb{E}_{c_t \sim P_t} \left[ \mathbb{E}_{s_{t+1}} \right. \right.$$

$$\left. \left. r_{t+1}(s_{t+1}, a_{t+1}) + \mathbb{E}_{c_{t+1} \sim P_{t+1}} \left[ \mathbb{E}_{s_{t+2}} \left[ \widetilde{V}_{t+2}^{\star,\sigma}(s_{t+2}) \right] \right] \right] \right| (a_t, a_{t+1}) = (\widetilde{\pi}_t(s), \widetilde{\pi}_{t+1}(s_{t+1})) \right]$$

$$\leq \cdots \leq \inf_{P_{+t} \in \mathcal{U}^\sigma(P_{+t}^c)} \mathbb{E}_{\pi, P_{+t}} \left[ \sum_{k=t}^{T} r_k(s_k, a_k) \right] = \widetilde{V}_t^{\widetilde{\pi},\sigma}(s). \tag{12}$$

Observing from (12) that

$$\forall s \in \mathcal{S}: \quad \widetilde{V}_t^{\star,\sigma}(s) \leq \widetilde{V}_t^{\widetilde{\pi},\sigma}(s) \leq \sup_{\pi \in \Pi} \widetilde{V}_t^{\pi,\sigma}(s) = \widetilde{V}_t^{\star,\sigma}(s), \tag{13}$$

which directly verifies the first assertion in (8) $\widetilde{V}_t^{\widetilde{\pi},\sigma}(s) = \widetilde{V}_t^{\star,\sigma}(s)$ for all $s \in \mathcal{S}$. The second assertion in (8) can be achieved analogously. Until now, we verify that there exists at least a policy $\widetilde{\pi}$ that obeys (8), which we refer to as an optimal policy since its value is equal to or larger than any other non-stationary and stochastic policies over all states $s \in \mathcal{S}$.

### C.2 Proof of Theorem 2

We establish the proof by separating it into several key steps.

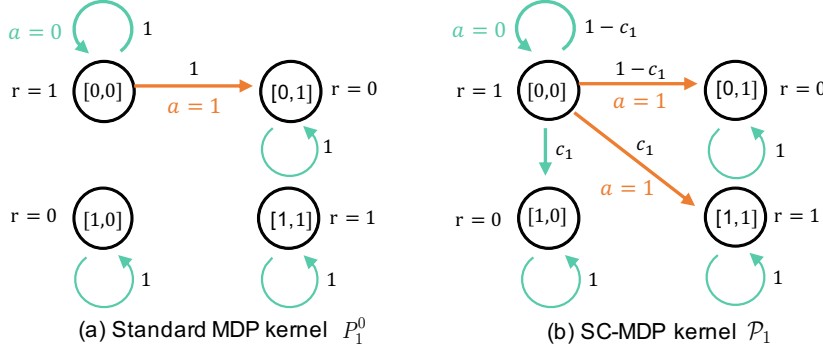

Figure 8: The illustration of the transition kernels of the standard MDP $\mathcal{M}$ and the proposed SC-MDP $\mathcal{M}_{\text{sc}}$ at the first time step $t = 1$, i.e., $P_t^0$ and $\mathcal{P}_1$ respectively.

**Step 1: Constructing a hard instance $\mathcal{M}$ of standard MDP.** In this section, we consider the following standard MDP instance $\mathcal{M} = \{\mathcal{S}, \mathcal{A}, P^0, T, r\}$ where $\mathcal{S} = \{[0,0], [0,1], [1,0], [1,1]\}$ is the state space consisting of four elements in dimension $n = 2$, and $\mathcal{A} = \{0, 1\}$ is the action space with only two options. The transition kernel $P^0 = \{P_t^0\}_{1 \le t \le T}$ at different time steps $1 \le t \le T$ is defined as

$$P_1^0(s' \mid s, a) = \begin{cases} \mathbb{1}(s' = [0,0])\mathbb{1}(a = 0) + \mathbb{1}(s' = [0,1])\mathbb{1}(a = 1) & \text{if } s = [0,0] \\ \mathbb{1}(s' = s) & \text{otherwise} \end{cases}, \qquad (14)$$

which is illustrated in Fig. 8(a), and

$$P_t^0(s' \mid s, a) = \mathbb{1}(s' = s), \quad \forall (t, s, a) \in \{2, 3, \cdots, T\} \times \mathcal{S} \times \mathcal{A}. \qquad (15)$$

Note that this transition kernel $P^0$ ensures that the next state transitioned from the state $[0,0]$ is either $[0,0]$ or $[0,1]$. The reward function is specified as follows: for all time steps $1 \le t \le T$,

$$r_t(s, a) = \begin{cases} 1 & \text{if } s = [0,0] \text{ or } s = [1,1] \\ 0 & \text{otherwise} \end{cases}. \qquad (16)$$

**Step 2: The equivalence between $\mathcal{M}$ and one SC-MDP.** Then, we shall show that the constructed standard MDP $\mathcal{M}$ can be equivalently represented by one SC-MDP $\mathcal{M}_{\text{sc}} = \{\mathcal{S}, \mathcal{A}, T, r, \mathcal{C}, \{\mathcal{P}_t^i\}, P^c\}$ with $\mathcal{C} := \{0, 1\}$. The equivalence is defined as the sequential observations $\{s_t, a_t, r_t\}_{1 \le t \le T}$ induced by any policy and any initial state distribution in two Markov processes are identical. To specify, $\mathcal{S}, \mathcal{A}, T, r$ are kept the same as $\mathcal{M}$. Here, $\{\mathcal{P}_t^i\}$ shall be specified in a while, which determines the transition to each dimension of the next state conditioned on the current state, action, and confounder distribution for all time steps, i.e., $s_{t+1}^i \sim \mathbb{E}_{c_t \sim P_t^c} \left[ \mathcal{P}_t^i(\cdot \mid s_t, a_t, c_t) \right]$ for any $i$-th dimension of the state ($i \in \{1, 2\}$) and all time step $1 \le t \le T$. For convenience, we denote $\mathcal{P}_t := [\mathcal{P}_t^1, \mathcal{P}_t^2] \in \Delta(\mathcal{S})$ as the transition kernel towards the next state, namely, $s_{t+1} \sim \mathbb{E}_{c_t \sim P_t^c} \left[ \mathcal{P}_t(\cdot \mid s_t, a_t, c_t) \right]$.

Then we simply set the nominal distribution of the confounder as follows:

$$P_t^c(c_t) = \mathbb{1}(c_t = 0), \quad \forall 1 \le t \le T, c_t \in \mathcal{C}. \qquad (17)$$

In addition, before introducing the transition kernel $\{\mathcal{P}_t^i\}$ of the SC-MDP $\mathcal{M}_{\text{sc}}$, we introduce an auxiliary transition kernel $P^{\text{sc}} = \{P_t^{\text{sc}}\}$ as follows:

$$P_1^{\text{sc}}(s' \mid s, a) = \begin{cases} \mathbb{1}(s' = [1,0])\mathbb{1}(a = 0) + \mathbb{1}(s' = [1,1])\mathbb{1}(a = 1) & \text{if } (s, a) = ([0,0], 0) \\ \mathbb{1}(s' = s) & \text{otherwise} \end{cases}, \qquad (18)$$

and

$$P_t^{\text{sc}}(s' \mid s, a) = \mathbb{1}(s' = s), \quad \forall (t, s, a) \in \{2, 3, \cdots, T\} \times \mathcal{S} \times \mathcal{A}. \qquad (19)$$

It can be observed that $P^{\text{sc}}$ is similar to $P^0$ except for the transition in the state $[0,0]$.

Armed with this transition kernel $P^{\text{sc}}$, the $\{\mathcal{P}_t^i\}$ of the SC-MDP $\mathcal{M}_{\text{sc}}$ is set to obey

$$\mathcal{P}_1(s' \,|\, s, a, c_1) = \begin{cases} (1 - c_1)P_1^0(s' \,|\, s, a) + c_1 P_1^{\text{sc}}(s' \,|\, s, a) & \text{if } s = [0, 0] \\ \mathbb{1}(s' = s) & \text{otherwise} \end{cases}, \qquad (20)$$

which is illustrated in Fig. 8(b), and

$$\mathcal{P}_t(s' \,|\, s, a, c_t) = \mathbb{1}(s' = s), \quad \forall (t, s, a, c_t) \in \{2, 3, \cdots, T\} \times \mathcal{S} \times \mathcal{A} \times \mathcal{C}. \qquad (21)$$

With them in mind, we are ready to verify that the marginalized transition from the current state and action to the next state in the SC-MDP $\mathcal{M}_{\text{sc}}$ is identical to the one in MDP $\mathcal{M}$: for all $(t, s_t, a_t, s_{t+1}) \in \{1, 2, \cdots, T\} \times \mathcal{S} \times \mathcal{A} \times \mathcal{S}$:

$$\mathbb{P}(s_{t+1} \,|\, s_t, a_t) = \mathbb{E}_{c_t \sim P_t^c}\left[\mathcal{P}_t(s_{t+1} \,|\, s_t, a_t, c_t)\right] = \mathcal{P}_t(s_{t+1} \,|\, s_t, a_t, 0) = P^0(s_{t+1} \,|\, s_t, a_t) \qquad (22)$$

where the second equality holds by the definition of $P^c$ in (17), and the last equality holds by the definitions of $\mathcal{P}$ (see (20) and (21)).

In summary, we verified that the standard MDP $\mathcal{M} = \{\mathcal{S}, \mathcal{A}, P^0, T, r\}$ is equal to the above specified SC-MDP $\mathcal{M}_{\text{sc}}$.

**Step 3: Defining corresponding RMDP and RSC-MDP.** Equipped with the equivalent standard MDP $\mathcal{M}$ and SC-MDP $\mathcal{M}_{\text{sc}}$, we consider the robust variants of them respectively — a RMDP $\mathcal{M}_{\text{rob}} = \{\mathcal{S}, \mathcal{A}, \mathcal{U}^{\sigma_1}(P^0), T, r\}$ with some uncertainty level $\sigma_1$, and the proposed RSC-MDP $\mathcal{M}_{\text{sc-rob}} = \{\mathcal{S}, \mathcal{A}, T, r, \mathcal{C}, \{\mathcal{P}_t^i\}, \mathcal{U}^{\sigma_2}(P^c)\}$ with some uncertainty level $\sigma_2$.

In this section, without loss of generality, we consider total variation as the 'distance' function $\rho$ for the uncertainty sets of both RMDP $\mathcal{M}_{\text{rob}}$ and RSC-MDP $\mathcal{M}_{\text{sc-rob}}$, i.e., for any probability vectors $P', P \in \Delta(\mathcal{C})$ (or $P', P \in \Delta(\mathcal{S})$), $\rho(P', P) := \frac{1}{2}\|P' - P\|_1$. Consequently, for any uncertainty level $\sigma \in [0, 1]$, the uncertainty set $\mathcal{U}^{\sigma_1}(P^0)$ of the RMDP (see (1)) and $\mathcal{U}^{\sigma_2}(P^c)$ of the RSC-MDP $\mathcal{M}_{\text{sc-rob}}$ (see (4)) are defined as follow, respectively:

$$\mathcal{U}^{\sigma}(P^0) := \otimes\, \mathcal{U}^{\sigma}(P_{t,s,a}^0), \qquad \mathcal{U}^{\sigma}(P_{t,s,a}^0) := \left\{ P_{t,s,a} \in \Delta(\mathcal{S}) : \frac{1}{2}\left\|P_{t,s,a} - P_{t,s,a}^0\right\|_1 \le \sigma \right\},$$

$$\mathcal{U}^{\sigma}(P^c) := \otimes\, \mathcal{U}^{\sigma}(P_t^c), \qquad \mathcal{U}^{\sigma}(P_t^c) := \left\{ P \in \Delta(\mathcal{C}) : \frac{1}{2}\|P - P_t^c\|_1 \le \sigma \right\}. \qquad (23)$$

**Step 4: Comparing between the performance of the optimal policy of RMDP $\mathcal{M}_{\text{rob}}$ ($\pi_{\text{RMDP}}^{\star,\sigma_1}$) and that of RSC-MDP $\mathcal{M}_{\text{sc-rob}}$ ($\pi_{\text{RSC}}^{\star,\sigma_2}$).** To continue, we specify the robust optimal policy $\pi_{\text{RMDP}}^{\star,\sigma_1}$ associated with $\mathcal{M}_{\text{rob}}$ and $\pi_{\text{RSC}}^{\star,\sigma_2}$ associated with $\mathcal{M}_{\text{sc-rob}}$ and then compare their performance on RSC-MDP with some initial state distribution.

To begin, we introduce the following lemma about the robust optimal policy $\pi_{\text{RMDP}}^{\star,\sigma_1}$ associated with the RMDP $\mathcal{M}_{\text{rob}}$.

**Lemma 1.** *For any $\sigma_1 \in (0, 1]$, the robust optimal policy of $\mathcal{M}_{\text{rob}}$ obeys*

$$\forall s \in \mathcal{S} : \quad \left[\pi_{\text{RMDP}}^{\star,\sigma_1}\right]_1(0 \,|\, s) = 1. \qquad (24a)$$

In addition, we characterize the robust SC-value functions of the RSC-MDP $\mathcal{M}_{\text{sc-rob}}$ associated with any policy, combined with the optimal policy and its optimal robust SC-value functions, shown in the following lemma.

**Lemma 2.** *Consider any $\sigma_2 \in (\frac{1}{2}, 1]$ and the RSC-MDP $\mathcal{M}_{\text{sc-rob}} = \{\mathcal{S}, \mathcal{A}, T, r, \mathcal{C}, \{\mathcal{P}_t^i\}, \mathcal{U}^{\sigma_2}(P^c)\}$. For any policy $\pi$, the corresponding robust SC-value functions satisfy*

$$\widetilde{V}_1^{\pi,\sigma_2}([0,0]) = 1 + (T - 1) \inf_{P \in \mathcal{U}^{\sigma}(P_1^c)} \mathbb{E}_{c_1 \sim P}\left[\pi_1(0 \,|\, [0,0])(1 - c_1) + \pi_1(1 \,|\, [0,0])c_1\right]. \qquad (25a)$$

*In addition, the optimal robust SC-value function and the robust optimal policy $\pi_{\text{RSC}}^{\star,\sigma_2}$ of the RMDP $\mathcal{M}_{\text{sc-rob}}$ obeys:*

$$\widetilde{V}_1^{\pi_{\text{RSC}}^{\star,\sigma_2},\sigma_2}([0,0]) = \widetilde{V}_1^{\star,\sigma_2}([0,0]) = 1 + \frac{T - 1}{2}. \qquad (26)$$

Armed with above lemmas, applying Lemma 2 with policy $\pi = \pi_{\mathsf{RMDP}}^{\star,\sigma_1}$ obeying $\left[\pi_{\mathsf{RMDP}}^{\star,\sigma_1}\right]_1(0 \,|\, s) = 1$ in Lemma 1, one has

$$\widetilde{V}_1^{\pi_{\mathsf{RMDP}}^{\star,\sigma_1},\sigma_2}([0,0]) = 1 + (T-1) \inf_{P \in \mathcal{U}^\sigma(P_1^c)} \mathbb{E}_{c_1 \sim P}\left[1 - c_1\right]$$

$$\leq 1 + (T-1)\left[\frac{1}{4}\cdot 1 + \frac{3}{4}\cdot 0\right] = 1 + \frac{T-1}{4}, \tag{27}$$

where the inequality holds by the fact that the probability distribution $P$ obeying $P_1(0) = \frac{1}{4}$ and $P_1(1) = \frac{3}{4}$ is inside the uncertainty set $\mathcal{U}^{\sigma_2}(P_1^c)$ (recall that $\sigma_2 \in (\frac{1}{2}, 1]$ and $P_1^c(0) = 1$).

Finally, combining (27) and (26) together, we complete the proof by showing that with the initial state distribution $\phi$ defined as $\phi([0,0]) = 1$, we arrive at

$$\widetilde{V}_1^{\pi_{\mathsf{RSC}}^{\star,\sigma_2},\sigma_2}(\phi) - \widetilde{V}_1^{\pi_{\mathsf{RMDP}}^{\star,\sigma_1},\sigma_2}([0,0]) = \widetilde{V}_1^{\star,\sigma_2}(\phi) - \widetilde{V}_1^{\pi_{\mathsf{RMDP}}^{\star,\sigma_1},\sigma_2}([0,0]) \geq \frac{T-1}{4} \geq \frac{T}{8}, \tag{28}$$

where the last inequality holds by $T \geq 2$.

## C.3 Proof of auxiliary results

### C.3.1 Proof of Lemma 1

**Step 1: specifying the minimum of the robust value functions over states.** For any uncertainty set $\sigma_1 \in (0, 1]$, we first characterize the robust value function of any policy $\pi$ over different states. To start, we denote the minimum of the robust value function over states at each time step $t$ as below:

$$V_{\min,t}^{\pi,\sigma_1} := \min_{s \in \mathcal{S}} V_t^{\pi,\sigma_1}(s) \geq 0, \tag{29}$$

where the last inequality holds that the reward function defined in (16) is always non-negative. Obviously, there exists at least one state $s_{\min,t}^\pi$ that satisfies $V_t^{\pi,\sigma_1}(s_{\min,t}^\pi) = V_{\min,t}^{\pi,\sigma_1}$.

With this in mind, we shall verify that for any policy $\pi$,

$$\forall 1 \leq t \leq T: \quad V_t^{\pi,\sigma_1}([0,1]) = V_t^{\pi,\sigma_1}([1,0]) = 0. \tag{30}$$

To achieve this, we use a recursive argument. First, the base case can be verified since when $t + 1 = T + 1$, the value functions are all zeros at $T + 1$ step, i.e., $V_{T+1}^{\pi,\sigma_1}(s) = 0$ for all $s \in \mathcal{S}$. Then, the goal is to verify the following fact

$$V_t^{\pi,\sigma_1}([0,1]) = V_t^{\pi,\sigma_1}([1,0]) = 0 \tag{31}$$

with the assumption that $V_{t+1}^{\pi,\sigma_1}(s) = 0$ for any state $s = \{[0,1],[1,0]\}$. It is easily observed that for any policy $\pi$, the robust value function when state $s = \{[0,1],[1,0]\}$ at any time step $t$ obeys

$$0 \leq V_t^{\pi,\sigma_1}(s) = \mathbb{E}_{a \sim \pi_t(\cdot \,|\, s)}\left[r_t(s,a) + \inf_{P \in \mathcal{U}^{\sigma_1}(P_{t,s,a}^0)} P V_{t+1}^{\pi,\sigma_1}\right]$$

$$\overset{(i)}{=} 0 + (1 - \sigma_1)V_{t+1}^{\pi,\sigma_1}(s) + \sigma_1 V_{\min,t+1}^{\pi,\sigma_1} \overset{(ii)}{=} 0 + \sigma_1 V_{\min,t+1}^{\pi,\sigma_1}$$

$$\leq 0 + \sigma_1 V_{t+1}^{\pi,\sigma_1}(s) = 0 \tag{32}$$

where (i) holds by $r_t(s,a) = 0$ for all $s = \{[0,1],[1,0]\}$, the fact $P_t^0(s \,|\, s, a) = 1$ for $s \in \mathcal{S}$ (see (14) and (15)), and the definition of the uncertainty set $\mathcal{U}^{\sigma_1}(P^0)$ in (23). Here (ii) follows from the recursive assumption $V_{t+1}^{\pi,\sigma_1}(s) = 0$ for any state $s = \{[0,1],[1,0]\}$, and the last equality holds by $V_{\min,t+1}^{\pi,\sigma_1} \leq V_{t+1}^{\pi,\sigma_1}([0,1])$ (see (29)). Until now, we complete the proof for (31) and then verify (30).

Note that (30) direcly leads to

$$\forall 1 \leq t \leq T: \quad V_{\min,t}^{\pi,\sigma_1} = 0. \tag{33}$$

**Step 2: Considering the robust value function at state $[0,0]$.** Armed with the above facts, we are now ready to derive the robust value function for the state $[0,0]$.

When $2 \le t \le T$, one has

$$V_t^{\pi,\sigma_1}([0,0]) = \mathbb{E}_{a \sim \pi_t(\cdot \,|\, [0,0])} \left[ r_t([0,0],a) + \inf_{P \in \mathcal{U}^{\sigma_1}(P_{t,[0,0],a})} PV_{t+1}^{\pi,\sigma_1} \right]$$

$$\overset{(i)}{=} 1 + \left[ (1-\sigma_1)V_{t+1}^{\pi,\sigma_1}([0,0]) + \sigma_1 V_{\min,t+1}^{\pi,\sigma_1} \right]$$

$$= 1 + (1-\sigma_1)V_{t+1}^{\pi,\sigma_1}([0,0]) \tag{34}$$

where (i) holds by $r_t([0,0],a) = 1$ for all $a \in \{0,1\}$ and the definition of $P^0$ (see (15)), and the last equality arises from (33) .

Applying (34) recursively for $t, t+1, \cdots, T$ yields that

$$V_t^{\pi,\sigma_1}([0,0]) = \sum_{k=t}^{T} (1-\sigma_1)^{k-t} \ge 1. \tag{35}$$

When $t = 1$, the robust value function obeys:

$$V_1^{\pi,\sigma_1}([0,0]) = \mathbb{E}_{a \sim \pi_1(\cdot \,|\, [0,0])} \left[ r_1([0,0],a) + \inf_{P \in \mathcal{U}^{\sigma_1}(P_{1,[0,0],a})} PV_2^{\pi,\sigma_1} \right]$$

$$\overset{(i)}{=} 1 + \pi_1(0 \,|\, [0,0]) \inf_{P \in \mathcal{U}^{\sigma_1}(P_{1,[0,0],0})} PV_2^{\pi,\sigma_1} + \pi_1(1 \,|\, [0,0]) \inf_{P \in \mathcal{U}^{\sigma_1}(P_{1,[0,0],1})} PV_2^{\pi,\sigma_1}$$

$$\overset{(ii)}{=} 1 + \pi_1(0 \,|\, [0,0]) \left[ (1-\sigma_1)V_2^{\pi,\sigma_1}([0,0]) + \sigma_1 V_{\min,2}^{\pi,\sigma_1} \right]$$

$$+ \pi_1(1 \,|\, [0,0]) \left[ (1-\sigma_1)V_2^{\pi,\sigma_1}([0,1]) + \sigma_1 V_{\min,2}^{\pi,\sigma_1} \right]$$

$$= 1 + \pi_1(0 \,|\, [0,0])(1-\sigma_1)V_2^{\pi,\sigma_1}([0,0]) \tag{36}$$

where (i) holds by $r_1([0,0],a) = 1$ for all $a \in \{0,1\}$, (ii) follows from the definition of $P^0$ (see (14)), and the last equality arises from (30) and (33).

**Step 3: the optimal policy $\pi_{\mathsf{RMDP}}^{\star,\sigma_1}$.** Observing that $V_1^{\pi,\sigma_1}([0,0])$ is increasing monotically as $\pi_1(0 \,|\, [0,0])$ is larger, we directly have that $\pi_{\mathsf{RMDP}}^{\star,\sigma_1}(0 \,|\, [0,0]) = 1$.

Considering that the action does not influence the state transition for $t = 2, 3, \cdots, T$ and all other states $s \ne [0,0]$, without loss of generality, we choose the robust optimal policy as

$$\forall s \in \mathcal{S}: \quad \left[ \pi_{\mathsf{RMDP}}^{\star,\sigma_1} \right]_1(0 \,|\, s) = 1. \tag{37}$$

### C.3.2 Proof of Lemma 2

To begin with, for any uncertainty level $\sigma_2 \in (\frac{1}{2}, 1]$ and any policy $\pi = \{\pi_t\}$, we consider the robust SC-value function $\widetilde{V}_t^{\pi,\sigma_2}$ of the RSC-MDP $\mathcal{M}_{\mathsf{sc\text{-}rob}}$.

**Step 1: deriving $\widetilde{V}_t^{\pi,\sigma_2}$ for $2 \le t \le T$.** Towards this, for any $2 \le t \le T$ and $s \in \mathcal{S}$, one has

$$\widetilde{V}_t^{\pi,\sigma_2}(s) = \mathbb{E}_{a \sim \pi_t(s)} \left[ \widetilde{Q}_t^{\pi,\sigma_2}(s,a) \right]$$

$$\overset{(i)}{=} \mathbb{E}_{a \sim \pi_t(s)} \left[ r_t(s,a) + \inf_{P \in \mathcal{U}^{\sigma}(P_t^c)} \mathbb{E}_{c_t \sim P} \left[ \mathcal{P}_{t,s,a,c_t} \widetilde{V}_{t+1}^{\pi,\sigma_2} \right] \right]$$

$$\overset{(ii)}{=} r_t(s,a) + \inf_{P \in \mathcal{U}^{\sigma}(P_t^c)} \mathbb{E}_{c_t \sim P} \left[ \mathcal{P}_{t,s,a,c_t} \widetilde{V}_{t+1}^{\pi,\sigma_2} \right]$$

$$= r_t(s,a) + \widetilde{V}_{t+1}^{\pi,\sigma}(s), \tag{38}$$

where (i) follows from the *state-confounded* Bellman consistency equation in (47), (ii) holds by that the reward function $r_t$ and $\mathcal{P}_t$ are all independent from the action (see (16) and (21)), and the last inequality holds by $\mathcal{P}_t(s' \,|\, s,a,c_t) = \mathbb{1}(s' = s)$ is independent from $c_t$ (see (21)).

Applying the above fact recursively for $t, t+1, \cdots, T$ leads to that for any $s \in \mathcal{S}$,

$$\widetilde{V}_t^{\pi,\sigma_2}(s) = r_t(s,a_t) + \widetilde{V}_{t+1}^{\pi,\sigma}(s) = r_t(s,a) + r_{t+1}(s,a_{t+1}) + \widetilde{V}_{t+2}^{\pi,\sigma}(s)$$

$$= \cdots = r_t(s, a_t) + \sum_{k=t+1}^{T} r_k(s_k, a_k), \tag{39}$$

which directly yields (see reward $r$ in (16))

$$\widetilde{V}_2^{\pi,\sigma_2}([0,0]) = \widetilde{V}_2^{\pi,\sigma_2}([1,1]) = T - 1 \quad \text{and} \quad \widetilde{V}_2^{\pi,\sigma_2}([0,1]) = \widetilde{V}_2^{\pi,\sigma_2}([1,0]) = 0. \tag{40}$$

**Step 2: characterizing $\widetilde{V}_1^{\pi,\sigma_2}([0,0])$ for any policy $\pi$.** In this section, we consider the value of $\widetilde{V}_1^{\pi,\sigma_2}$ on the state $[0,0]$. To proceed, one has

$$\widetilde{V}_1^{\pi,\sigma_2}([0,0]) = \mathbb{E}_{a \sim \pi_1([0,0])} \left[ \widetilde{Q}_1^{\pi,\sigma_2}([0,0], a) \right]$$

$$\stackrel{(i)}{=} \mathbb{E}_{a \sim \pi_1([0,0])} \left[ r_1([0,0], a) + \inf_{P \in \mathcal{U}^\sigma(P_1^c)} \mathbb{E}_{c_1 \sim P} \left[ \mathcal{P}_{1,[0,0],a,c_1} \widetilde{V}_2^{\pi,\sigma_2} \right] \right]$$

$$\stackrel{(ii)}{=} 1 + \inf_{P \in \mathcal{U}^\sigma(P_1^c)} \mathbb{E}_{c_1 \sim P} \left[ \left( \pi_1(0 \mid [0,0]) \mathcal{P}_{1,[0,0],0,c_1} + \pi_t(1 \mid [0,0]) \mathcal{P}_{1,[0,0],1,c_1} \right) \widetilde{V}_2^{\pi,\sigma} \right]$$

$$\stackrel{(iii)}{=} 1 + \inf_{P \in \mathcal{U}^\sigma(P_1^c)} \mathbb{E}_{c_1 \sim P} \left[ \pi_1(0 \mid [0,0]) \left( (1 - c_1) P_{1,[0,0],0}^0 + c_1 P_{1,[0,0],0}^{\mathrm{sc}} \right) \widetilde{V}_2^{\pi,\sigma} \right.$$

$$\left. + \pi_1(1 \mid [0,0]) \left( (1 - c_1) P_{1,[0,0],1}^0 + c_1 P_{1,[0,0],1}^{\mathrm{sc}} \right) \widetilde{V}_2^{\pi,\sigma} \right]$$

$$\stackrel{(iv)}{=} 1 + \inf_{P \in \mathcal{U}^\sigma(P_1^c)} \mathbb{E}_{c_1 \sim P} \left[ \pi_1(0 \mid [0,0]) \left( (1 - c_1) \widetilde{V}_2^{\pi,\sigma}([0,0]) + c_1 \widetilde{V}_2^{\pi,\sigma}([1,0]) \right) \right.$$

$$\left. + \pi_1(1 \mid [0,0]) \left( (1 - c_1) \widetilde{V}_2^{\pi,\sigma}([0,1]) + c_1 \widetilde{V}_2^{\pi,\sigma}([1,1]) \right) \right]$$

$$= 1 + (T - 1) \inf_{P \in \mathcal{U}^\sigma(P_1^c)} \mathbb{E}_{c_1 \sim P} \left[ \pi_1(0 \mid [0,0])(1 - c_1) + \pi_1(1 \mid [0,0]) c_1 \right]$$

$$= 1 + (T - 1)\pi_1(0 \mid [0,0]) + (T - 1) \inf_{P \in \mathcal{U}^\sigma(P_1^c)} \mathbb{E}_{c_1 \sim P} \left[ c_1 \left( 1 - 2\pi_1(0 \mid [0,0]) \right) \right], \tag{41}$$

where (i) holds by *robust state-confounded* Bellman consistency equation in (47), (ii) follows from $r_1([0,0], a) = 1$ for all $a \in \{0,1\}$ which is independent from $c_t$. (iii) arises from the definition of $\mathcal{P}$ in (20), (iv) can be verified by plugging in the definitions from (14) and (18), and the penultimate equality holds by (40).

**Step 3: characterizing the optimal robust SC-value functions.** Before proceeding, we recall the fact that $\mathcal{U}^\sigma(P_1^c) = \left\{ P \in \Delta(\mathcal{C}) : \frac{1}{2} \|P - P_1^c\|_1 \leq \sigma_2 \right\}$.

Observing from (41) that for any fixed $\pi_1(0 \mid [0,0])$, $c_1 \left( 1 - 2\pi_1(0 \mid [0,0]) \right)$ is monotonously increasing with $c_1$ when $1 - 2\pi_1(0 \mid [0,0]) \geq 0$ and decreasing with $c_1$ otherwise, it is easily verified that the maximum of the following function

$$f\left( \pi_1(0 \mid [0,0]) \right) := (T - 1) \inf_{P \in \mathcal{U}^\sigma(P_1^c)} \mathbb{E}_{c_1 \sim P} \left[ c_1 \left( 1 - 2\pi_1(0 \mid [0,0]) \right) \right] \tag{42}$$

obeys

$$\max f\left( \pi_1(0 \mid [0,0]) \right) = \begin{cases} 0 & \text{if } \pi_1(0 \mid [0,0]) \geq \frac{1}{2} \\ (T - 1)\sigma_2 \left( 1 - 2\pi_1(0 \mid [0,0]) \right) & \text{otherwise} \end{cases}. \tag{43}$$

Then, note that the value of $\widetilde{V}_1^{\pi,\sigma_2}([0,0])$ only depends on $\pi_1(\cdot \mid [0,0])$ which can be represent by $\pi_1(0 \mid [0,0])$. Plugging in (43) into (41) arrives at when $\pi_1(0 \mid [0,0]) \geq \frac{1}{2}$,

$$\max_{\pi_1(0 \mid [0,0]) \geq \frac{1}{2}} \widetilde{V}_1^{\pi,\sigma_2}([0,0])$$

$$= \max_{\pi_1(0\,|\,[0,0])\geq\frac{1}{2}} 1 + (T-1)\pi_1(0\,|\,[0,0]) + (T-1)\sigma_2\big(1 - 2\pi_1(0\,|\,[0,0])\big)$$

$$= 1 + (T-1)\sigma_2 + (T-1)\max_{\pi_1(0\,|\,[0,0])\geq\frac{1}{2}}(1 - 2\sigma_2)\pi_1(0\,|\,[0,0])$$

$$= 1 + (T-1)\sigma_2 + \frac{(T-1)(1-2\sigma_2)}{2} = 1 + \frac{T-1}{2}, \tag{44}$$

where the penultimate equality holds by $\sigma_2 > \frac{1}{2}$ and letting $\pi_1(0\,|\,[0,0]) = \frac{1}{2}$. Similarly, when $\pi_1(0\,|\,[0,0]) < \frac{1}{2}$,

$$\max_{\pi_1(0\,|\,[0,0])<\frac{1}{2}} \widetilde{V}_1^{\pi,\sigma_2}([0,0]) = \max_{\pi_1(0\,|\,[0,0])<\frac{1}{2}} 1 + (T-1)\pi_1(0\,|\,[0,0]) < 1 + \frac{T-1}{2}. \tag{45}$$

Consequently, combining (44) and (45), we conclude that

$$\widetilde{V}_1^{\pi_{\mathsf{RSC}}^{\star,\sigma_2},\sigma_2}([0,0]) = \widetilde{V}_1^{\star,\sigma_2}([0,0]) = \max_{\pi} \widetilde{V}_1^{\pi,\sigma_2}([0,0]) = 1 + \frac{T-1}{2}. \tag{46}$$

### C.3.3  Auxiliary results of RSC-MDPs

It is easily verified that for any RSC-MDP $\mathcal{M}_{\mathsf{sc\text{-}rob}} = \big\{\mathcal{S}, \mathcal{A}, T, r, \mathcal{C}, \{\mathcal{P}_t^i\}, \mathcal{U}^{\sigma_2}(P^c)\big\}$, any policy $\pi$ and optimal policy $\pi^\star$ satisfy the corresponding *robust state-confounded* Bellman consistency equation and Bellman optimality equation shown below, respectively:

$$\widetilde{Q}_t^{\pi,\sigma}(s,a) = r_t(s,a) + \inf_{P\in\mathcal{U}^\sigma(P_t^c)} \mathbb{E}_{c_t\sim P}\left[\mathcal{P}_{t,s,a,c_t}\widetilde{V}_{t+1}^{\pi,\sigma}\right],$$

$$\widetilde{Q}_t^{\star,\sigma}(s,a) = r_t(s,a) + \inf_{P\in\mathcal{U}^\sigma(P_t^c)} \mathbb{E}_{c_t\sim P}\left[\mathcal{P}_{t,s,a,c_t}\widetilde{V}_{t+1}^{\star,\sigma}\right], \tag{47}$$

where $\mathcal{P}_{t,s,a,c_t} \in \mathbb{R}^{1\times S}$ such that $\mathcal{P}_{t,s,a,c_t}(s') := \mathcal{P}_t(s'\,|\,s,a,c_t)$ for $s' \in \mathcal{S}$, and $\widetilde{V}_t^{\star,\sigma}(s) = \sup_{\pi_t\in\Delta(\mathcal{A})}\left\{\mathbb{E}_{\pi_t}[r_t(s,a_t)] + \inf_{P_t\in\mathcal{U}^\sigma(P_t^c)}\mathbb{E}_{\pi_t}\left[\mathbb{E}_{c_t\sim P_t}\left[\mathcal{P}_{t,s,a,c_t}\widetilde{V}_{t+1}^{\star,\sigma}(s_{t+1})\right]\right]\right\}$.

## D  Experiment Details

### D.1  Architecture of the structural causal model

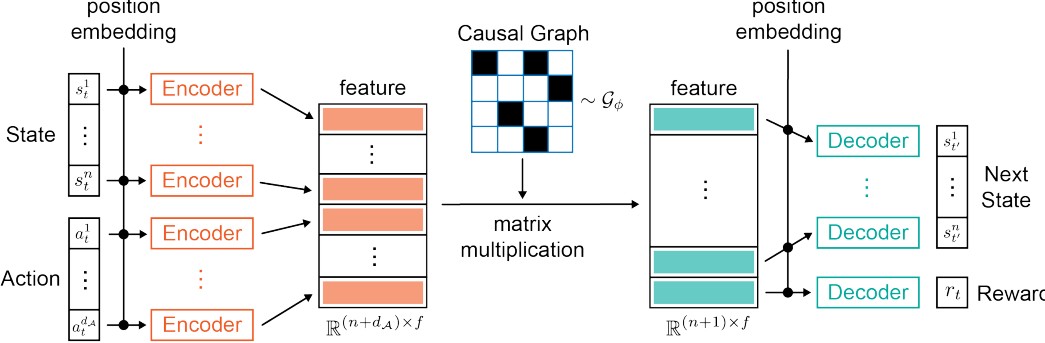

Figure 9: Model architecture of the structural causal model. Encoder, Decoder, position embedding, and Causal Graph are learnable during the training stage.

We plot the architecture of the structural causal model we used in our method in Figure 9. In normal neural networks, the input is treated as a whole to pass through linear layers or convolution layers. However, this structure blends all information in the input, making the causal graph useless to separate cause and effect. Thus, in our model, we design an encoder that is shared across all dimensions of the input. Since different dimensions could have exactly the same values, we add a learnable position

embedding to the input of the encoder. In summary, the input dimension of the encoder is $1 + d_{pos}$, where $d_{pos}$ is the dimension of the position embedding.

After the encoder, we obtain a set of independent features for each dimension of the input. We now multiply the features with a learnable binary causal graph $G$. The element $(i, j)$ of the graph is sampled from a Gumbel-Softmax distribution with parameter $\phi_{i,j}$ to ensure the loss function is differentiable w.r.t $\phi$.

The multiplication of the causal graph and the input feature creates a linear combination of the input feature with respect to the causal graph. The obtained features are then passed through a decoder to predict the next state and reward. Again, the decoder is shared across all dimensions to avoid information leaking between dimensions. Position embedding is included in the input to the decoder and the output dimension of the decoder is $1$.

## D.2 Details of Tasks

We design four self-driving tasks in the Carla simulator [22] and four manipulation tasks in the Robosuite platform [23]. All of these realistic tasks contain strong spurious correlations that are explicit to humans. We provide detailed descriptions of all these environments in the following.

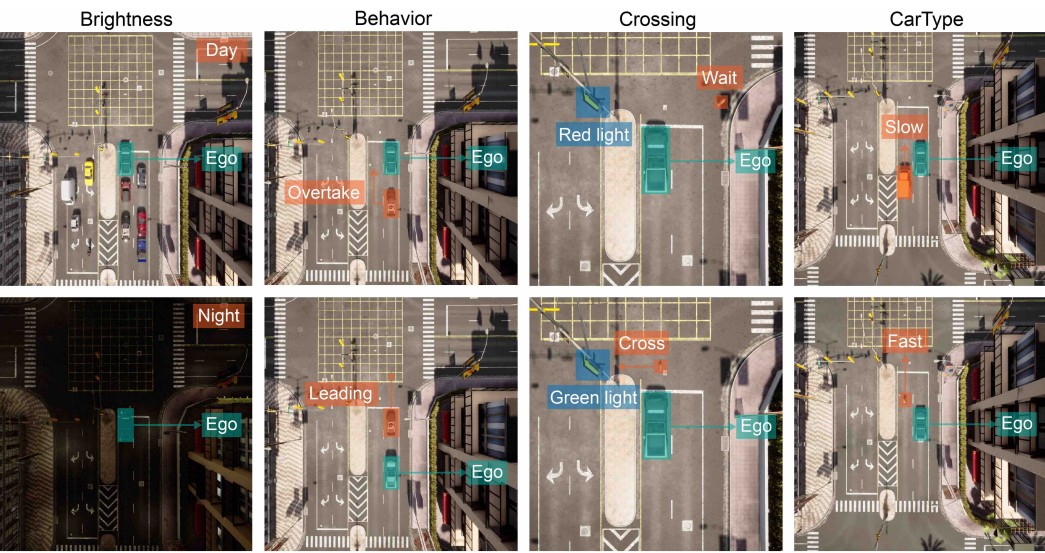

Figure 10: Illustration of tasks in the Carla simulator.

**Brightness.** The nominal environments are shown in the 1[th] column of Figure 10, where the brightness and the traffic density are correlated. When the ego vehicle drives in the daytime, there are many surrounding vehicles (first row). When the ego vehicle drives in the evening, there is no surrounding vehicle (second row). The shifted environment swaps the brightness and traffic density in the nominal environment, i.e., many surrounding vehicles in the evening and no surrounding vehicles in the daytime.

**Behavior.** The nominal environments are shown in the 2[nd] column of Figure 10, where the other vehicle has aggressive driving behavior. When the ego vehicle is in front of the other vehicle, the other vehicle always accelerates and overtakes the ego vehicle in the left lane. When the ego vehicle is behind the other vehicle, the other vehicle will always accelerate. In the shifted environment, the behavior of the other vehicle is conservative, i.e., the other vehicle always decelerates to block the ego vehicle.

**Crossing.** The nominal environments are shown in the 3[rd] column of Figure 10, where the pedestrian follows the traffic rule and only crosses the road when the traffic light is green. In the shifted environment, the pedestrian disobeys the traffic rules and crosses the road when the traffic light is red.

**CarType.** The nominal environments are shown in the 4[th] column of Figure 10, where the type of vehicle and the speed of the vehicle are correlated. When the vehicle is a truck, the speed is low and

when the vehicle is a motorcycle, the speed is high. In the shifted environment, the truck drives very fast and the motorcycle drives very slow.

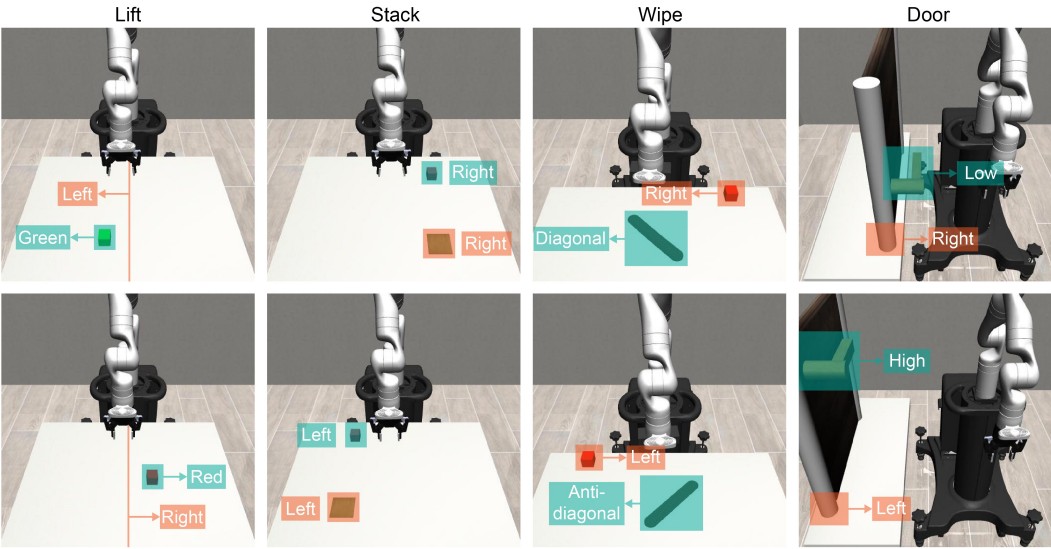

Figure 11: Illustration of tasks in the Robosuite simulator.

**Lift.** The nominal environments are shown in the $1^{th}$ column of Figure 11, where the position of the cube and the color of the cube are correlated. When the cube is in the left part of the table, the color of the cube is green, when the cube is in the right part of the table, the color of the cube is red. The shifted environment swaps the color and position of the cube in the nominal environment, i.e., the cube is green when it is in the right part and the cube is red when it is in the left part.

**Stack.** The nominal environments are shown in the $2^{nd}$ column of Figure 11, where the position of the red cube and green plate are correlated. When the cube is in the left part of the table, the plate is also in the left part; when the cube is in the right part of the table, the plate is also in the right part. In the shifted environment, the relative position of the cube and the plate changes, i.e., When the cube is in the left part of the table, the plate is in the right part; when the cube is in the right part of the table, the plate is in the left part.

**Wipe.** The nominal environments are shown in the $3^{rd}$ column of Figure 11, where the shape of the dirty region is correlated to the position of the cube. When the dirty region is diagonal, the cube is on the right-hand side of the robot arm. When the dirty region is anti-diagonal, the cube is on the left-hand side of the robot arm. In the shifted environment, the correlation changes, i.e., when the dirty region is diagonal, the cube is on the left-hand side of the robot arm; when the dirty region is anti-diagonal, the cube is on the right-hand side of the robot arm.

**Door.** The nominal environments are shown in the $4^{th}$ column of Figure 11, where the height of the handle and the position of the door are correlated. When the door is closed to the robot arm, the handle is in a low position. When the door is far from the robot arm, the handle is in a high position. In the shifted environment, the correlation changes, i.e., when the door is closed to the robot arm, the handle is in a high position; when the door is far from the robot arm, the handle is in a low position.

### D.3 Example of Generated Data by Perturbations

We show an example of generated trajectories in the Lift task to demonstrate the reason why our method obtains robustness against spurious correlation. In Figure 12 (a), we show the collected trajectories from the data buffer. Since the green block is always generated on the left side of the table, the trajectories of the green block mainly appear on the left side of the table. In Figure 12 (b), we generate new trajectories from a trained transition model and we observe that the distribution of trajectories follows the collected data. In Figure 12 (c), we directly perturbed the dimensions of the state and used the same transition model to generate new trajectories. We find that the generated trajectories blend the color but fail to maintain the spatial distribution of the original data. In Figure 12

(d), we use the causal-based transition model to generate new trajectories and we find that the results not only follow the spatial distribution but also blend the color.

The results shown in Figure 12 illustrate that the data generated by our method eliminates the spurious correlation between the color and position of the block, therefore, enabling the policy model to generalize to the shifted environment.

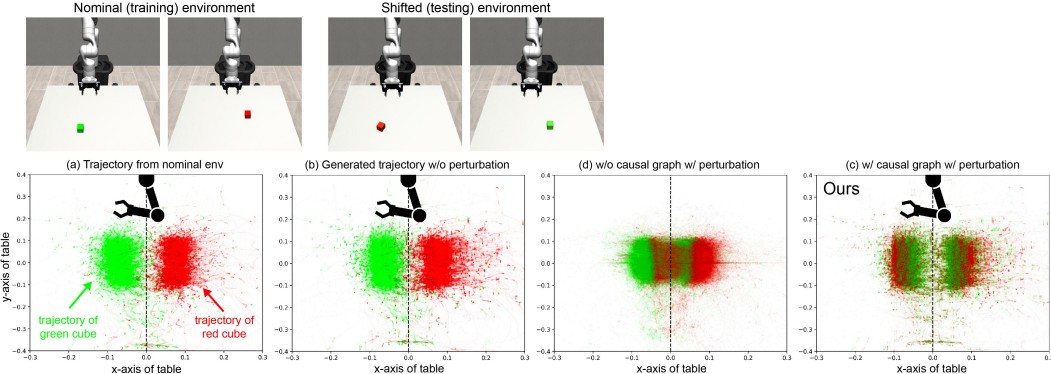

Figure 12: The generated transition data from different perturbation methods. (a) Trajectories collected from the policy interact with the nominal environment. (b) Generated trajectories without any perturbation. (c) Generated trajectories with perturbation but without the causal graph. (d) Generated trajectories with perturbation and with the causal graph.

## D.4 Computation Resources

Our algorithm is implemented on top of the Tianshou [101] package. All of our experiments are conducted on a machine with an Intel i9-9900K CPU@3.60GHz (16 core) CPU, an NVIDIA GeForce GTX 1080Ti GPU, and 64GB memory.

## D.5 Hyperparameters

We summarize all hyper-parameters used in the Carla experiments (Table 5) and Robosuite experiments (Table 6).

## D.6 Discovered Causal Graph in SCM

To show the performance of our learned SCM, we plot the estimated causal graphs of all experiments in Figure 13, Figure 14, Figure 15, Figure 16, and Figure 17.

Table 5: Hyper-parameters in Carla experiments

| Parameters | Notation | Environment | | | |
|---|---|---|---|---|---|
| | | Brightness | Behavior | Crossing | CarType |
| Horizon steps | $T$ | 100 | 100 | 100 | 100 |
| State dimension | $n$ | 24 | 12 | 12 | 12 |
| Action dimension | $d_{\mathcal{A}}$ | 2 | 2 | 2 | 2 |
| Max training steps | | $1\times10^5$ | $1\times10^5$ | $5\times10^5$ | $5\times10^5$ |
| Weight of $\|\boldsymbol{G}\|_p$ | $\lambda$ | 0.1 | - | - | - |
| norm of $\|\boldsymbol{G}\|_p$ | $p$ | 0.1 | - | - | - |
| Actor learning rate | | $3 \times 10^{-4}$ | - | - | - |
| Critic learning rate | | $1 \times 10^{-3}$ | - | - | - |
| Batch size | | 256 | - | - | - |
| Discount factor | $\gamma$ in SAC | 0.99 | - | - | - |
| Soft update weight | $\tau$ in SAC | 0.005 | - | - | - |
| Weight of entropy | $\alpha$ in SAC | 0.1 | - | - | - |
| Hidden layers | | [256, 256, 256] | - | - | - |
| Returns estimation step | | 4 | - | - | - |
| Buffer size | | $1 \times 10^5$ | - | - | - |
| Steps per update | | 10 | - | - | - |

Table 6: Hyper-parameters in Robosuite experiments

| Parameters | Notation | Environment | | | |
|---|---|---|---|---|---|
| | | Lift | Stack | Door | Wipe |
| Horizon steps | $T$ | 300 | 300 | 300 | 500 |
| Control frequency (Hz) | | 20 | 20 | 20 | 20 |
| State dimension | $n$ | 50 | 110 | 22 | 30 |
| Action dimension | $d_{\mathcal{A}}$ | 4 | 4 | 8 | 7 |
| Controller type | | OSC position | OSC position | Joint velocity | Joint velocity |
| Max training steps | | $1\times10^6$ | $5\times10^6$ | $1\times10^6$ | $1\times10^6$ |
| Weight of $\|\boldsymbol{G}\|_p$ | $\lambda$ | 0.01 | - | - | - |
| norm of $\|\boldsymbol{G}\|_p$ | $p$ | 0.1 | - | - | - |
| Actor learning rate | | $3 \times 10^{-4}$ | - | - | - |
| Critic learning rate | | $1 \times 10^{-3}$ | - | - | - |
| Batch size | | 128 | - | - | - |
| Discount factor | $\gamma$ in SAC | 0.99 | - | - | - |
| Soft update weight | $\tau$ in SAC | 0.005 | - | - | - |
| alpha learning rate | $lr_\alpha$ in SAC | $3 \times 10^{-4}$ | - | - | - |
| Hidden layers | | [256, 256, 256] | - | - | - |
| Returns estimation step | | 4 | - | - | - |
| Buffer size | | $1 \times 10^6$ | - | - | - |
| Steps per update | | 10 | - | - | - |

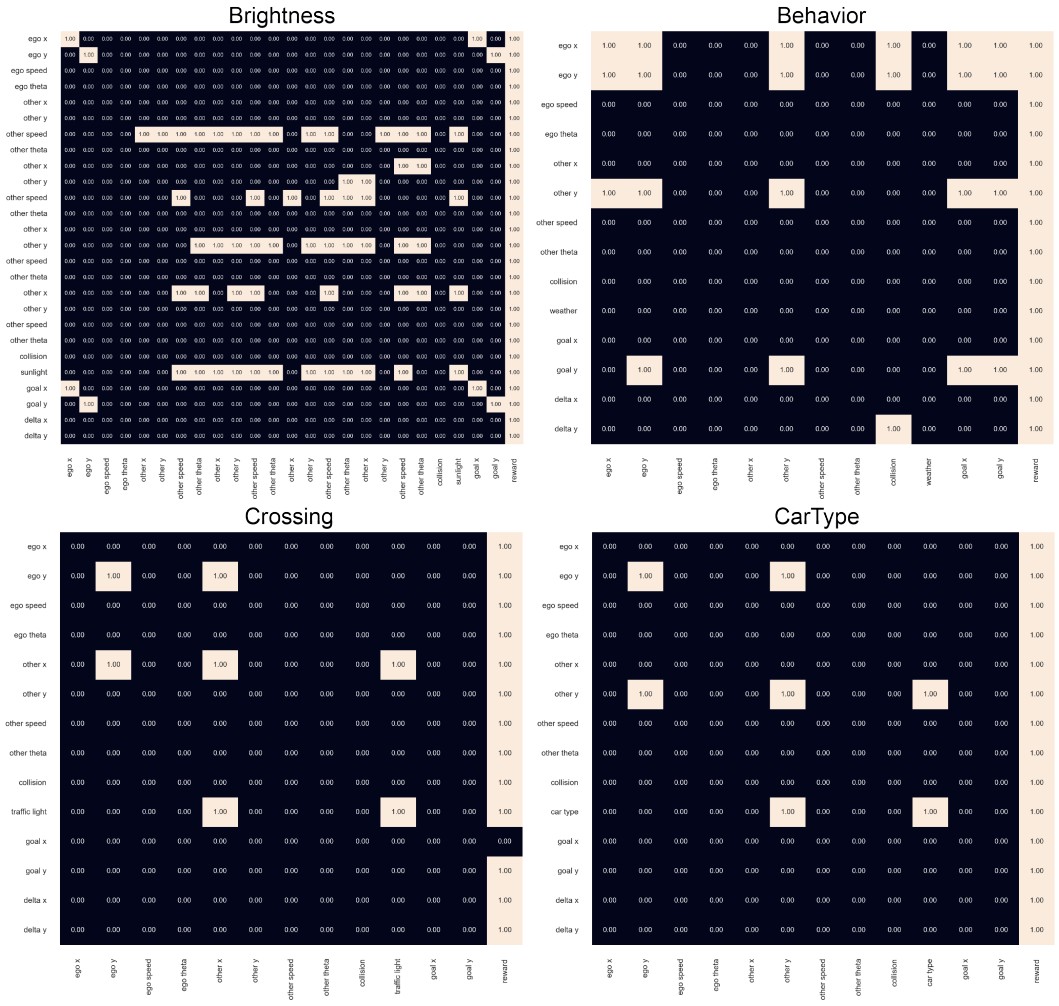

Figure 13: Estimated Causal Graphs of four tasks in Carla.

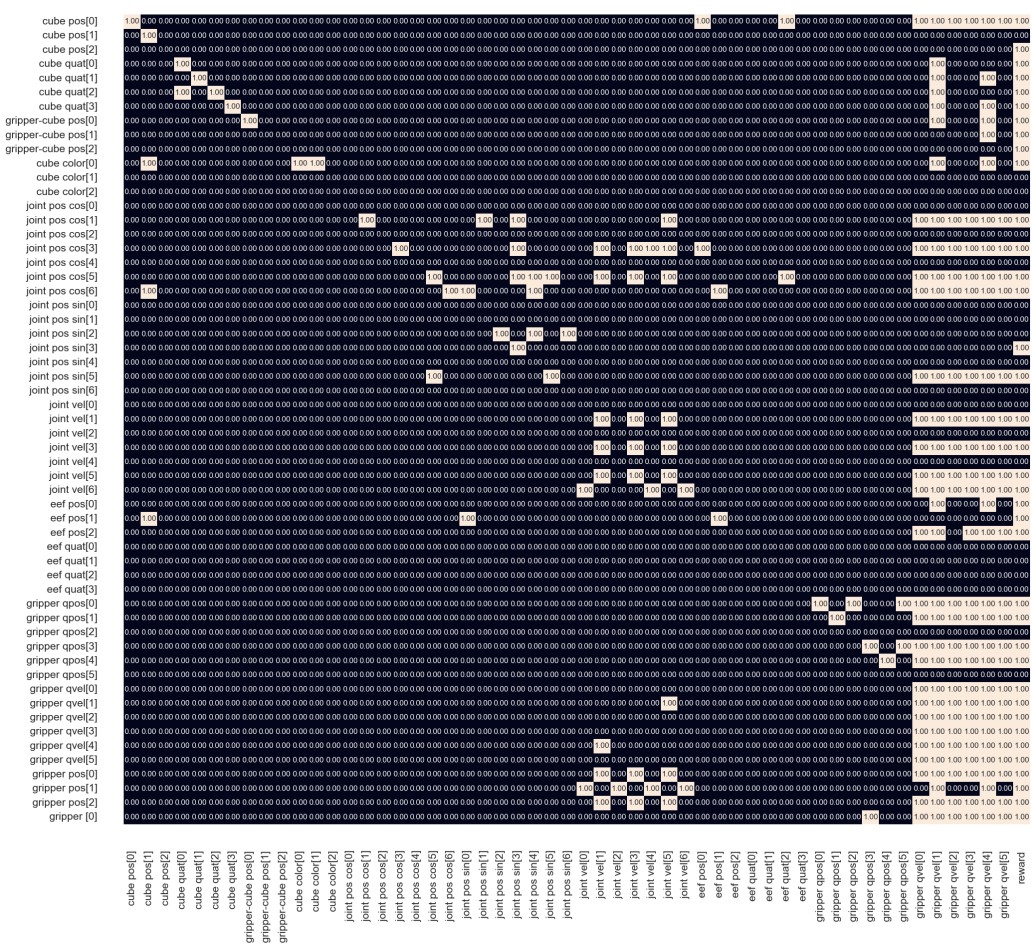

Figure 14: Estimated Causal Graphs of the Lift task in Robosuite.

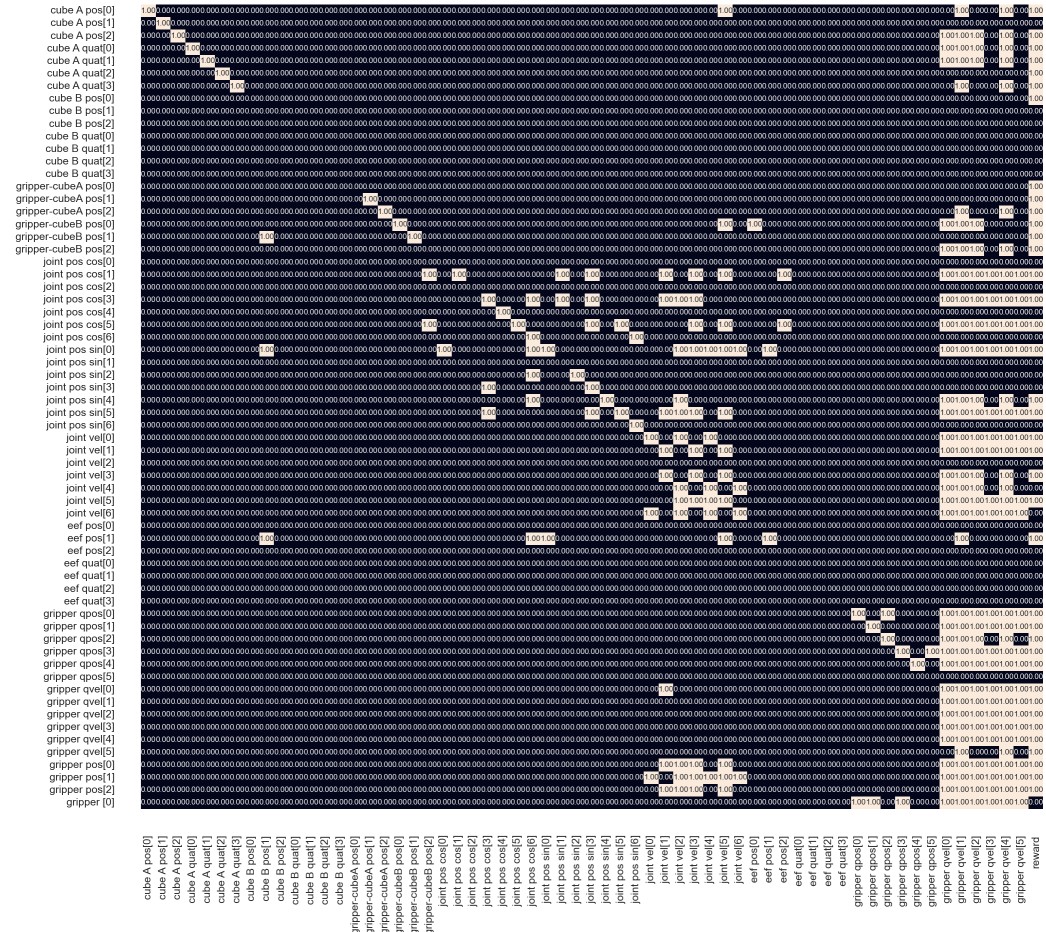

Figure 15: Estimated Causal Graphs of the Stack task in Robosuite.

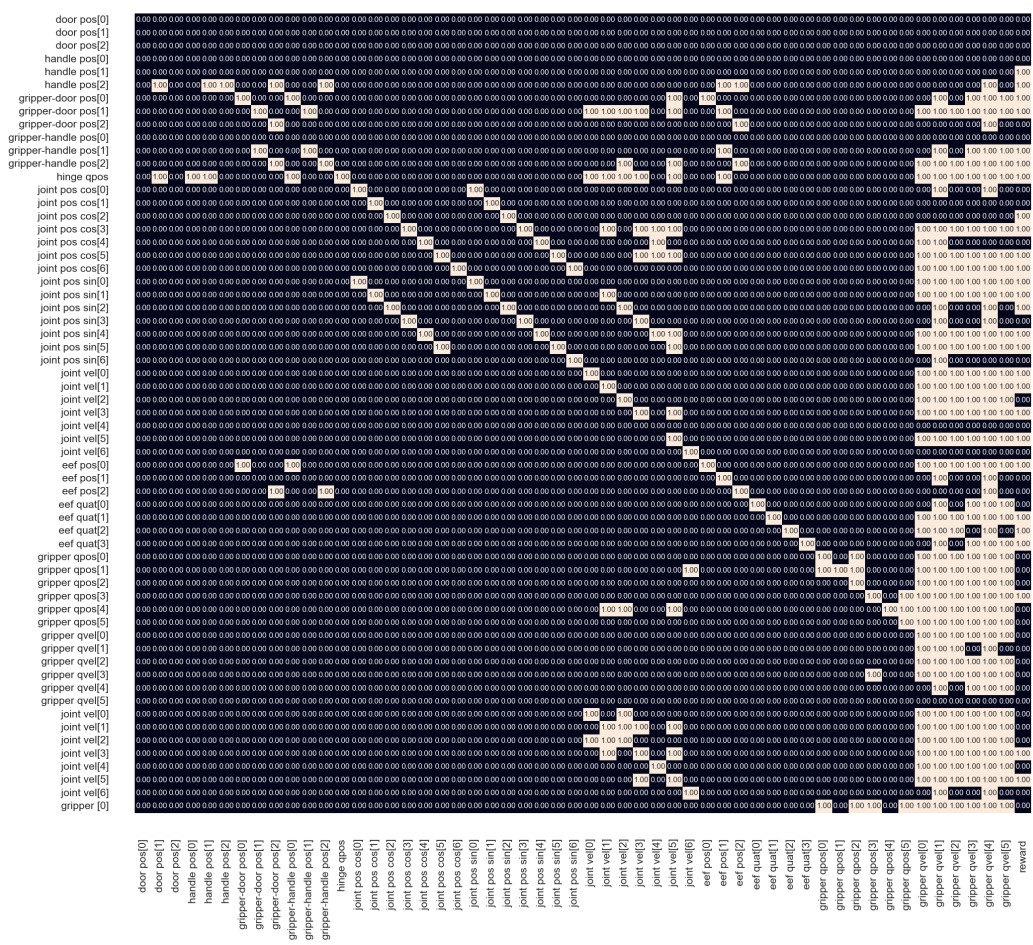

Figure 16: Estimated Causal Graphs of the Door task in Robosuite.

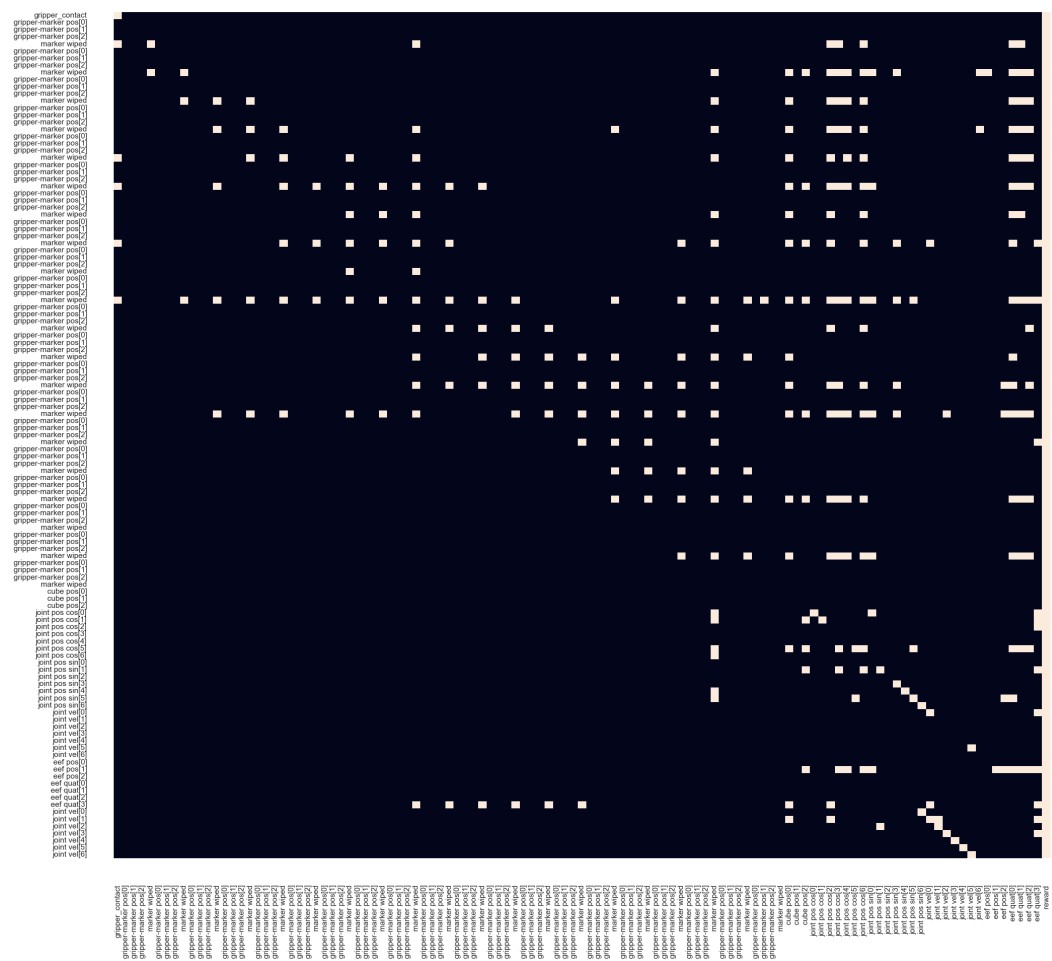

Figure 17: Estimated Causal Graphs of the Wipe task in Robosuite.

