The proof follows the pipeline of proving the existence of the optimal policy for standard MDPs but tailored for RSC-MDPs since the additional components confounder $C_s$ and the infimum operator. To begin with, recall that the goal is to find a policy $\widetilde{\pi} = \{\widetilde{\pi}_t\}_{1 \leq t \leq T}$ such that:

$$\widetilde{V}_t^{\widetilde{\pi},\sigma}(s) = \widetilde{V}_t^{\star,\sigma}(s) \coloneqq \sup_{\pi \in \Pi} \widetilde{V}_t^{\pi,\sigma}(s) \quad \text{and} \quad \widetilde{Q}_t^{\widetilde{\pi},\sigma}(s,a) = \widetilde{Q}_t^{\star,\sigma}(s,a) \coloneqq \sup_{\pi \in \Pi} \widetilde{Q}_t^{\pi,\sigma}(s,a). \tag{8}$$

Towards this, we start from the first claim in equation 8. Before proceeding, we let $\{S_t, A_t, R_t, C_t\}$ denote the random variables at time step $t$ for all $1 \leq t \leq T$. Then due to the Markov properties, we know that conditioned on current state $s_t$, the future state, action, and reward are all independent from the previous $s_1, a_1, r_1, c_1, \cdots, s_{t-1}, a_{t-1}, r_{t-1}, c_{t-1}$. For convenience, we introduce the following notation:

$$\forall 1 \leq t \leq T: \quad P_{+t} := \{P_k\}_{t \leq k \leq T} \quad \text{and} \quad \mathcal{U}^\sigma(P_{+t}^c) := \{\mathcal{U}^\sigma(P_k^c)\}_{t \leq k \leq T} \tag{9}$$

to represent the collection of variables from time step $t$ to the end of the episode, and choose $\widetilde{\pi}$ to obey

$$\forall 1 \leq t \leq T: \quad \pi_t(s) := \mathrm{argmax}_{a \in \mathcal{A}} \mathbb{E}\left[r_t(s,a) + \inf_{P_t \in \mathcal{U}^\sigma(P_{t,s,a}^c)} \mathbb{E}_{c_t \sim P_t}\left[\widetilde{V}_{t+1}^{\star,\sigma}(s_{t+1})\right]\right] \tag{10}$$

With the above preparation in mind, for any $(t,s) \in \{1, 2, \cdots, T\} \times \mathcal{S}$, one has

$$\widetilde{V}_t^{\star,\sigma}(s)$$

$$\overset{(i)}{=} \sup_{\pi \in \Pi} \inf_{P_{+t} \in \mathcal{U}^\sigma(P_{+t}^c)} \widetilde{V}_t^{\pi,P}(s) \overset{(ii)}{=} \sup_{\pi \in \Pi} \inf_{P_{+t} \in \mathcal{U}^\sigma(P_{+t}^c)} \mathbb{E}_{\pi, P_{+t}}\left[\sum_{k=t}^T r_k(s_k, a_k)\right]$$

$$\overset{(iii)}{=} \sup_{\pi \in \Pi} \inf_{P_{+t} \in \mathcal{U}^\sigma(P_{+t}^c)} \mathbb{E}_{\pi_t}\left[r_t(s, a_t)\right.$$

$$\left. + \mathbb{E}_{c_t \sim P_t}\mathbb{E}\left[\sum_{k=t+1}^T r_k(s_k, a_k) \,|\, \pi, P_{+(t+1)}, (S_t, A_t, R_t, C_t, S_{t+1}) = (s, a_t, r_t, c_t, s_{t+1})\right]\right]$$

$$= \sup_{\pi \in \Pi} \mathbb{E}_{\pi_t}\left[\inf_{P_{+t} \in \mathcal{U}^\sigma(P_{+t}^c)} r_t(s, a_t) + \inf_{P_{+t} \in \mathcal{U}^\sigma(P_{+t}^c)} \mathbb{E}_{c_t \sim P_t}\right.$$

$$\left. \mathbb{E}\left[\sum_{k=t+1}^T r_k(s_k, a_k) \,|\, \pi, P_{+(t+1)}, (S_t, A_t, R_t, C_t, S_{t+1}) = (s, a_t, r_t, c_t, s_{t+1})\right]\right]$$

where (i) and (ii) holds by the definitions in equation 5 and equation 3 respectively, and (iii) follows from expressing the term of interest by moving one step ahead and $\mathbb{E}_{\pi_t}$ is taken with respect to $a_t \sim \pi_t(\cdot \,|\, S_1 = s_1, A_1 = a_1, \cdots, S_t = s)$, and the last equality arises from we can exchange the operators $\mathbb{E}_{\pi_t}$ and $\inf_{P \in \mathcal{U}^\sigma(P^c)}$ since they are independent.

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

To ensure the marginalized transition probability from any state-action pair $(s_t, a_t)$ to the next state $s_{t+1}$ in $\mathcal{M}_{\mathsf{sc}}$ aligns with the one in the MDP $\mathcal{M}$, we set

$$P_t^c(c) = \mathbb{1}(c = 0), \quad \forall 1 \leq t \leq T. \tag{17}$$

In addition, before introducing the transition kernel $\{\mathcal{P}_t^i\}$ of the SC-MDP $\mathcal{M}_{\mathsf{sc}}$, we introduce an auxiliary transition kernel $P^{\mathsf{sc}} = \{P_t^{\mathsf{sc}}\}$ as follows:

$$P_1^{\mathsf{sc}}(s' \mid s, a) = \begin{cases} \mathbb{1}(s' = [1, 0])\mathbb{1}(a = 0) + \mathbb{1}(s' = [1, 1])\mathbb{1}(a = 1) & \text{if}(s, a) = ([0, 0], 0) \\ \mathbb{1}(s' = s) & \text{otherwise} \end{cases}, \tag{18}$$

and

$$P_t^{\mathsf{sc}}(s' \mid s, a) = \mathbb{1}(s' = s), \quad \forall(t, s, a) \in \{2, 3, \cdots, T\} \times \mathcal{S} \times \mathcal{A}. \tag{19}$$

It can be observed that $P^{\mathsf{sc}}$ is similar to $P^0$ except for the transition in the state $[0, 0]$.

Armed with this transition kernel $P^{\mathsf{sc}}$, the $\{\mathcal{P}_t^i\}$ of the SC-MDP $\mathcal{M}_{\mathsf{sc}}$ is set to obey

$$\mathcal{P}_1(s' \mid s, a, c) = \begin{cases} (1 - c)P_1^0(s' \mid s, a) + cP_1^{\mathsf{sc}}(s' \mid s, a) & \text{if}(s, a) = ([0, 0], a) \\ \mathbb{1}(s' = s) & \text{otherwise} \end{cases}, \tag{20}$$

and

$$\mathcal{P}_t(s' \mid s, a, c) = \mathbb{1}(s' = s), \quad \forall(t, s, a, c) \in \{2, 3, \cdots, T\} \times \mathcal{S} \times \mathcal{A} \times \mathcal{C}. \tag{21}$$

With the above preparation, we are ready to verify that the marginalized transition from the current state and action to the next state in the SC-MDP $\mathcal{M}_{\mathsf{sc}}$ is identical to the one in MDP $\mathcal{M}$: for all $(t, s_t, a_t, s_{t+1}) \in \{1, 2, \cdots, T\} \times \mathcal{S} \times \mathcal{A} \times \mathcal{S}$:

$$\mathbb{P}(s_{t+1} \mid s_t, a_t) = \mathbb{E}_{c_t \sim P_t^c}[\mathcal{P}_t(s_{t+1} \mid s_t, a_t, c_t)] = \mathcal{P}_t(s_{t+1} \mid s_t, a_t, 0) = P^0(s_{t+1} \mid s_t, a_t) \tag{22}$$

where the second equality holds by the definition of $P^c$ in equation 17, and the last equality holds by the definitions of $P^0$ (see equation 14 and equation 15) and $\mathcal{P}$ (see equation 20 and equation 21).

In summary, we verified that the standard MDP $\mathcal{M} = \{\mathcal{S}, \mathcal{A}, P^0, T, r\}$ is equal to the above specified SC-MDP $\mathcal{M}_{\mathsf{sc}}$.

**Defining the corresponding RMDP and RSC-MDP.** Equipped with the equivalent MDP $\mathcal{M}$ and SC-MDP $\mathcal{M}_{\mathsf{sc}}$, people could consider the robust variants of them respectively — a RMDP $\mathcal{M}_{\mathsf{rob}} = \{\mathcal{S}, \mathcal{A}, \mathcal{U}^{\sigma_1}(P^0), T, r\}$ with the uncertainty level $\sigma_1$, and the proposed RSC-MDP $\mathcal{M}_{\mathsf{sc\text{-}rob}} = \{\mathcal{S}, \mathcal{A}, T, r, \mathcal{C}, \{\mathcal{P}_t^i\}, \mathcal{U}^{\sigma_2}(P^c)\}$ with the uncertainty level $\sigma_2$.

In this section, without loss of generality, we consider total deviation as the 'distance' function $\rho$ for the uncertainty sets of both RMDP $\mathcal{M}_{\mathsf{rob}}$ and RSC-MDP $\mathcal{M}_{\mathsf{sc\text{-}rob}}$, i.e., for any probability vectors $P', P \in \Delta(\mathcal{C})$ (or $P', P \in \Delta(\mathcal{S})$), $\rho(P', P) := \frac{1}{2}\|P' - P\|_1$. Consequently, for any uncertainty set $\sigma \in [0, 1]$, the uncertainty set $\mathcal{U}^{\sigma_1}(P^0)$ of the RMDP (see equation 1) and $\mathcal{U}^{\sigma_2}(P^c)$ of the RSC-MDP $\mathcal{M}_{\mathsf{sc\text{-}rob}}$ (see equation 4) are defined as follows:

$$\mathcal{U}^{\sigma}(P^0) := \otimes\, \mathcal{U}^{\sigma}(P_{t,s,a}^0), \qquad \mathcal{U}^{\sigma}(P_{t,s,a}^0) := \left\{ P_{t,s,a} \in \Delta(\mathcal{S}) : \frac{1}{2}\|P_{t,s,a} - P_{t,s,a}^0\|_1 \leq \sigma \right\},$$

$$\mathcal{U}^{\sigma}(P^c) := \otimes\, \mathcal{U}^{\sigma}(P_t^c), \qquad \mathcal{U}^{\sigma}(P_t^c) := \left\{ P \in \Delta(\mathcal{C}) : \frac{1}{2}\|P - P_t^c\|_1 \leq \sigma \right\}. \tag{23}$$

To continue, the proof is established by specifying the robust optimal policy $\pi_{\mathsf{RMDP}}^{\star, \sigma_1}$ associated with $\mathcal{M}_{\mathsf{rob}}$ and $\pi_{\mathsf{RSC}}^{\star, \sigma_2}$ associated with $\mathcal{M}_{\mathsf{sc\text{-}rob}}$ and then compare their performance on RSC-MDP with some initial state distribution.

**The performance comparisons between $\pi_{\mathsf{RMDP}}^{\star, \sigma_1}$ of RMDP $\mathcal{M}_{\mathsf{rob}}$ and $\pi_{\mathsf{RSC}}^{\star, \sigma_2}$ of RSC-MDP $\mathcal{M}_{\mathsf{sc\text{-}rob}}$.**

To begin, we introduce the following lemma which specifies the robust optimal policy $\pi_{\mathsf{RMDP}}^{\star, \sigma_1}$ associated with the RMDP $\mathcal{M}_{\mathsf{rob}}$.

**Lemma 1.** *For any $\sigma_1 \in (0, 1]$, the robust optimal policy and its corresponding robust SC-value functions satisfy*

$$\pi_{\mathsf{RMDP}}^{\star,\sigma_1}(0 \,|\, s) = 1, \qquad \text{for } s \in \mathcal{S}. \tag{24a}$$

In addition, we characterize the robust SC-value functions of the RSC-MDP $\mathcal{M}_{\text{sc-rob}}$ associated with any policy, combined with the robust optimal policy $\pi_{\mathsf{RSC}}^{\star,\sigma_2}$ of $\mathcal{M}_{\text{sc-rob}}$ — the optimal robust SC-value functions, shown in the following lemma.

**Lemma 2.** *Consider any $\sigma_2 \in (\frac{3}{4}, 1]$ and the RSC-MDP $\mathcal{M}_{\text{sc-rob}} = \big\{\mathcal{S}, \mathcal{A}, T, r, \mathcal{C}, \{\mathcal{P}_t^i\}, \mathcal{U}^{\sigma_2}(P^c)\big\}$. For any policy $\pi$, the corresponding robust SC-value functions satisfy*

$$\widetilde{V}_1^{\pi,\sigma_2}([0,0]) = 1 + (T-1)\inf_{P \in \mathcal{U}^\sigma(P_1^c)} \mathbb{E}_{c_1 \sim P}\left[\pi_1(0 \,|\, [0,0])(1 - c_1) + \pi_1(1 \,|\, [0,0])c_1\right]. \tag{25a}$$

*In addition, the optimal robust SC-value function and the robust optimal policy $\pi_{\mathsf{RSC}}^{\star,\sigma_2}$ of the RMDP $\mathcal{M}_{\text{sc-rob}}$ obeys:*

$$\widetilde{V}_1^{\pi_{\mathsf{RSC}}^{\star,\sigma_2},\sigma_2}([0,0]) = \widetilde{V}_1^{\star,\sigma_2}([0,0]) = 1 + \frac{T-1}{2}. \tag{26}$$

Applying Lemma 2 with policy $\pi = \pi_{\mathsf{RMDP}}^{\star,\sigma_1}$ in Lemma 1, one has

$$\widetilde{V}_1^{\pi_{\mathsf{RMDP}}^{\star,\sigma_1},\sigma_2}([0,0]) = 1 + (T-1)\inf_{P \in \mathcal{U}_2^\sigma(P_1^c)} \mathbb{E}_{c_1 \sim P}\left[1 - c_1\right] \leq 1 + \frac{T-1}{4}, \tag{27}$$

where the last inequality holds by the probability distribution $P$ obeying $P_1(0) = \frac{1}{4}$ and $P_1(1) = \frac{3}{4}$ is inside the uncertainty set $\mathcal{U}_2^\sigma(P_1^c)$.

Finally, putting equation 27 and equation 26 together, we complete the proof by showing that with the initial state distribution $\phi$ define as $\rho(s_1 = [0,0]) = 1$, we arrive at

$$\widetilde{V}_1^{\pi_{\mathsf{RSC}}^{\star,\sigma_2},\sigma_2}(\phi) - \widetilde{V}_1^{\pi_{\mathsf{RMDP}}^{\star,\sigma_1},\sigma_2}(\phi) = \widetilde{V}_1^{\star,\sigma_2}(\phi) - \widetilde{V}_1^{\pi_{\mathsf{RMDP}}^{\star,\sigma_1},\sigma_2}(\phi) \geq \frac{T-1}{4} \approx \frac{T}{4}. \tag{28}$$

### C.2.1 Proof of Lemma 1

**Specifying the minimum of the robust value functions in different states.** For any uncertainty set $\sigma_1 \in (0, 1]$, we first characterize the robust value function of any policy $\pi$ over different states. To start, we denote the minimum of the robust value function over states at each time step $t$ as below:

$$V_{\min,t}^{\pi,\sigma_1} := \min_{s \in \mathcal{S}} V_t^{\pi,\sigma_1}(s) \geq 0, \tag{29}$$

where the last inequality holds by that the reward function defined in equation 16 is always non-negative. Obviously, there exists at least one state $s_{\min,t}^\pi$ that satisfies $V_t^{\pi,\sigma_1}(s_{\min,t}^\pi) = V_{\min,t}^{\pi,\sigma_1}$.

With this in mind, we shall verify that for any policy $\pi$,

$$\forall 1 \leq t \leq T: \quad V_t^{\pi,\sigma_1}([0,1]) = V_t^{\pi,\sigma_1}([1,0]) = 0. \tag{30}$$

To achieve this, we will use a recursive argument. First, the base case can be verified since when $t + 1 = T + 1$, the value functions are all zeros at $T + 1$ step, i.e., $V_{t+1}^{\pi,\sigma_1}(s) = V_{T+1}^{\pi,\sigma_1}(s) = 0$ for all $s \in \mathcal{S}$. Then, the goal is to verify the following fact

$$V_t^{\pi,\sigma_1}([0,1]) = V_t^{\pi,\sigma_1}([1,0]) = 0 \tag{31}$$

with the assumption that $V_{t+1}^{\pi,\sigma_1}(s) = 0$ for any state $s = \{[0,1],[1,0]\}$. It is easily observed that for any policy $\pi$, the robust value function when state $s = \{[0,1],[1,0]\}$ at any time step $t$ obeys

$$0 \leq V_t^{\pi,\sigma_1}(s) = \mathbb{E}_{a \sim \pi_t(\cdot \,|\, s)}\left[r_t(s,a) + \inf_{P \in \mathcal{U}^{\sigma_1}(P_{t,s,a}^0)} P V_{t+1}^{\pi,\sigma_1}\right]$$
$$\overset{(i)}{=} 0 + (1 - \sigma_1)V_{t+1}^{\pi,\sigma_1}(s) + \sigma_1 V_{\min,t+1}^{\pi,\sigma_1} \overset{(ii)}{=} 0 + \sigma_1 V_{\min,t+1}^{\pi,\sigma_1}$$

$$\leq 0 + \sigma_1 V_{t+1}^{\pi,\sigma_1}(s) = 0 \tag{32}$$

where (i) holds by $r_t(s,a) = 0$ for all $s = \{[0,1],[1,0]\}$, the fact $P_t^0(s\,|\,s,a) = 1$ (see equation 14 and equation 15), and the definition of the uncertainty set $\mathcal{U}^{\sigma_1}(P^0)$ in equation 23, (ii) follows from the recursive assumption $V_{t+1}^{\pi,\sigma_1}(s) = 0$ for any state $s = \{[0,1],[1,0]\}$, and the last equality holds by $V_{\min,t+1}^{\pi,\sigma_1} \leq V_{t+1}^{\pi,\sigma_1}(s)$ (see equation 29). Until now, we complete the proof for equation 31 and then verify equation 30.

Note that equation 30 direcly leads to

$$\forall 1 \leq t \leq T: \quad V_{\min,t}^{\pi,\sigma_1} = 0. \tag{33}$$

**Considering the robust value function at state** $[0,0]$. Armed with above facts, we are now ready to derive the robust value function for the state $[0,0]$.

When $2 \leq t \leq T$, one has

$$V_t^{\pi,\sigma_1}([0,0]) = \mathbb{E}_{a\sim\pi_t(\cdot\,|\,[0,0])} \left[ r_t([0,0],a) + \inf_{P\in\mathcal{U}^{\sigma_1}(P_{t,[0,0],a})} PV_{t+1}^{\pi,\sigma_1} \right]$$

$$\overset{(i)}{=} 1 + \left[ (1-\sigma_1)V_{t+1}^{\pi,\sigma_1}([0,0]) + \sigma_1 V_{\min,t+1}^{\pi,\sigma_1} \right]$$

$$= 1 + (1-\sigma_1)V_{t+1}^{\pi,\sigma_1}([0,0]) \tag{34}$$

where (i) holds by $r_t([0,0],a) = 1$ for all $a \in \{0,1\}$ and the definition of $P^0$ (see equation 14 and equation 15), and the last equality arises from equation 33.

Applying equation 34 recursively for $t, t+1, \cdots, T$ yields that

$$V_t^{\pi,\sigma_1}([0,0]) = \sum_{k=t}^{T}(1-\sigma_1)^{k-t} \geq 1. \tag{35}$$

At the first step, the robust value function obeys:

$$V_1^{\pi,\sigma_1}([0,0]) = \mathbb{E}_{a\sim\pi_1(\cdot\,|\,[0,0])} \left[ r_t([0,0],a) + \inf_{P\in\mathcal{U}^{\sigma_1}(P_{1,[0,0],a})} PV_2^{\pi,\sigma_1} \right]$$

$$\overset{(i)}{=} 1 + \pi_1(0\,|\,[0,0]) \inf_{P\in\mathcal{U}^{\sigma_1}(P_{1,[0,0],0})} PV_2^{\pi,\sigma_1} + \pi_1(1\,|\,[0,0]) \inf_{P\in\mathcal{U}^{\sigma_1}(P_{1,[0,0],1})} PV_2^{\pi,\sigma_1}$$

$$\overset{(ii)}{=} 1 + \pi_1(0\,|\,[0,0]) \left[ (1-\sigma_1)V_2^{\pi,\sigma_1}([0,0]) + \sigma_1 V_{\min,2}^{\pi,\sigma_1} \right]$$

$$+ \pi_1(1\,|\,[0,0]) \left[ (1-\sigma_1)V_2^{\pi,\sigma_1}([0,1]) + \sigma_1 V_{\min,2}^{\pi,\sigma_1} \right]$$

$$= 1 + \pi_1(0\,|\,[0,0])(1-\sigma_1)V_2^{\pi,\sigma_1}([0,0]) \tag{36}$$

where (i) holds by $r_t([0,0],a) = 1$ for all $a \in \{0,1\}$, (ii) follows from the definition of $P^0$ (see equation 14 and equation 15), and the last equality arises from equation 30 and equation 33.

**The optimal policy** $\pi_{\mathsf{RMDP}}^{\star,\sigma_1}$. Observing that the positive value of $V_2^{\pi,\sigma_1}([0,0])$ verified in equation 35, as $V_1^{\pi,\sigma_1}([0,0])$ is increasing monotically as $\pi_1(0\,|\,[0,0])$ is larger, we directly have that $\pi_{\mathsf{RMDP}}^{\star,\sigma_1}(0\,|\,[0,0]) = 1$.

Considering that the action does not influence the state transition for all other states $s \neq [0,0]$, without loss of generality, we choose the robust optimal policy to obey

$$\forall s \in \mathcal{S}: \quad \pi_{\mathsf{RMDP}}^{\star,\sigma_1}(0\,|\,s) = 1. \tag{37}$$

### C.2.2 Proof of Lemma 2

To begin with, for any uncertainty level $\sigma_2 \in (\frac{1}{2},1]$ and any policy $\pi = \{\pi_t\}$, we consider the robust SC-value function $\widetilde{V}_1^{\pi,\sigma_2}$ of the RSC-MDP $\mathcal{M}_{\mathsf{sc\text{-}rob}}$.

**Deriving** $\widetilde{V}_t^{\pi,\sigma_2}$ **for** $2 \leq t \leq T$. Towards this, for any $2 \leq t \leq T$ and $s \in \mathcal{S}$, one has

$$\widetilde{V}_t^{\pi,\sigma_2}(s) \overset{(i)}{=} \inf_{P\in\mathcal{U}^{\sigma}(P^c)} \widetilde{V}_t^{\pi,P}(s) = \inf_{P\in\mathcal{U}^{\sigma}(P_t^c)} \mathbb{E}_{a\sim\pi_t(s)} \left[ \widetilde{Q}_t^{\pi,P}(s,a) \right]$$

$$\overset{\text{(ii)}}{=} \inf_{P\in\mathcal{U}^\sigma(P_t^c)} \mathbb{E}_{a\sim\pi_t(s)} \left[ r_t(s,a) + \mathbb{E}_{c_t\sim P} \left[ \mathcal{P}_{t,s,a,c_t} \widetilde{V}_{t+1}^{\pi,\sigma} \right] \right]$$

$$\overset{\text{(iii)}}{=} r_t(s,a) + \inf_{P\in\mathcal{U}^\sigma(P_t^c)} \mathbb{E}_{c_t\sim P} \left[ \mathcal{P}_{t,s,a,c_t} \widetilde{V}_{t+1}^{\pi,\sigma} \right]$$

$$= r_t(s,a) + \widetilde{V}_{t+1}^{\pi,\sigma}(s), \tag{38}$$

where (i) holds by the definition in equation 5, (ii) follows from the *state-confounded* Bellman consistency equation in equation 47, (iii) follows from that the reward function $r$ and $\mathcal{P}_t$ are all independent from the action (see equation 16, equation 17 and equation 21), and the last inequality holds by $\mathcal{P}_t(s'\,|\,s,a,c) = \mathbb{1}(s'=s)$ is independent from $c_t$ (see equation 21).

Applying the above fact recursively for $t, t+1, \cdots, T$ leads to that for any $s \in \mathcal{S}$,

$$\widetilde{V}_t^{\pi,\sigma_2}(s) = r_t(s,a_t) + \widetilde{V}_{t+1}^{\pi,\sigma}(s) = r_t(s,a) + r_{t+1}(s,a_{t+1}) + \widetilde{V}_{t+2}^{\pi,\sigma}(s)$$

$$= \cdots = r_t(s,a_t) + \sum_{k=t+1}^{T} r_k(s_k,a_k), \tag{39}$$

which directly yields

$$\widetilde{V}_2^{\pi,\sigma_2}([0,0]) = \widetilde{V}_2^{\pi,\sigma_2}([1,1]) = T-1 \quad \text{and} \quad \widetilde{V}_2^{\pi,\sigma_2}([0,1]) = \widetilde{V}_2^{\pi,\sigma_2}([1,0]) = 0. \tag{40}$$

**Characterizing $\widetilde{V}_1^{\pi,\sigma_2}([0,0])$ for any policy $\pi$.** In this section, we are especially interested in the value of $\widetilde{V}_1^{\pi,\sigma_2}$ on the state $[0,0]$. To proceed, one has

$$\widetilde{V}_1^{\pi,\sigma_2}([0,0]) \overset{\text{(i)}}{=} \inf_{P\in\mathcal{U}^\sigma(P^c)} \widetilde{V}_1^{\pi,P}([0,0]) = \inf_{P\in\mathcal{U}^\sigma(P_1^c)} \mathbb{E}_{a\sim\pi_1([0,0])} \left[ \widetilde{Q}_1^{\pi,P}([0,0],a) \right]$$

$$\overset{\text{(ii)}}{=} \inf_{P\in\mathcal{U}^\sigma(P_1^c)} \mathbb{E}_{a\sim\pi_t([0,0])} \left[ r_1([0,0],a) + \mathbb{E}_{c_t\sim P} \left[ \mathcal{P}_{1,[0,0],a,c_t} \widetilde{V}_2^{\pi,\sigma} \right] \right]$$

$$\overset{\text{(iii)}}{=} 1 + \inf_{P\in\mathcal{U}^\sigma(P_1^c)} \mathbb{E}_{c_1\sim P} \left[ \left( \pi_1(0\,|\,[0,0])\mathcal{P}_{1,[0,0],0,c_1} + \pi_t(1\,|\,[0,0])\mathcal{P}_{1,[0,0],1,c_1} \right) \widetilde{V}_2^{\pi,\sigma} \right]$$

$$\overset{\text{(iv)}}{=} 1 + \inf_{P\in\mathcal{U}^\sigma(P_1^c)} \mathbb{E}_{c_1\sim P} \left[ \pi_1(0\,|\,[0,0]) \left( (1-c_1)P_{1,[0,0],0}^0 + c_1 P_{1,[0,0],0}^{\mathrm{sc}} \right) \widetilde{V}_2^{\pi,\sigma} \right.$$

$$\left. + \pi_1(1\,|\,[0,0]) \left( (1-c_1)P_{1,[0,0],1}^0 + c_1 P_{1,[0,0],1}^{\mathrm{sc}} \right) \widetilde{V}_2^{\pi,\sigma} \right]$$

$$\overset{\text{(v)}}{=} 1 + \inf_{P\in\mathcal{U}^\sigma(P_1^c)} \mathbb{E}_{c_1\sim P} \left[ \pi_1(0\,|\,[0,0]) \left( (1-c_1)\widetilde{V}_2^{\pi,\sigma}([0,0]) + c_1 \widetilde{V}_2^{\pi,\sigma}([1,0]) \right) \right.$$

$$\left. + \pi_1(1\,|\,[0,0]) \left( (1-c_1)\widetilde{V}_2^{\pi,\sigma}([0,1]) + c_1 \widetilde{V}_2^{\pi,\sigma}([1,1]) \right) \right]$$

$$= 1 + (T-1) \inf_{P\in\mathcal{U}^\sigma(P_1^c)} \mathbb{E}_{c_1\sim P} \left[ \pi_1(0\,|\,[0,0])(1-c_1) + \pi_1(1\,|\,[0,0])c_1 \right]$$

$$= 1 + (T-1)\pi_1(0\,|\,[0,0]) + (T-1) \inf_{P\in\mathcal{U}^\sigma(P_1^c)} \mathbb{E}_{c_1\sim P} \left[ c_1\big(1 - 2\pi_1(0\,|\,[0,0])\big) \right], \tag{41}$$

where (i) holds by the definition in equation 5, (ii) follows from the *state-confounded* Bellman consistency equation in equation 47, (iii) follows from $r_1([0,0],a) = 1$ for all $a \in \{0,1\}$ which is independent from $c_t$. (iv) arises from the definition of $\mathcal{P}$ in equation 20, (v) can be verified by plugging in the definitions from equation 14 and equation 18, and the penultimate equality holds by equation 40.

**Characterizing the optimal robust SC-value functions.**

To further consider equation 41, we recall the fact that $\mathcal{U}^\sigma(P_1^c) = \left\{ P \in \Delta(\mathcal{C}) : \frac{1}{2}\|P - P_1^c\|_1 \le \sigma_2 \right\}$.

Observing from equation 41 that for any fixed $\pi_1(0\,|\,[0,0])$, $c_1\big(1 - 2\pi_1(0\,|\,[0,0])\big)$ is monotonously increasing with $c_1$ when $1 - 2\pi_1(0\,|\,[0,0]) >= 0$ and decreasing with $c_1$ otherwise, it is easily verified that the solution of

$$f(\pi_1(0\,|\,[0,0])) := (T-1) \inf_{P \in \mathcal{U}^\sigma(P_1^c)} \mathbb{E}_{c_1 \sim P}\big[c_1\big(1 - 2\pi_1(0\,|\,[0,0])\big)\big] \tag{42}$$

satisfies

$$f(\pi_1(0\,|\,[0,0])) = \begin{cases} 0 & \text{if } \pi_1(0\,|\,[0,0]) \geq \frac{1}{2} \\ (T-1)\sigma_2\big(1 - 2\pi_1(0\,|\,[0,0])\big) & \text{otherwise} \end{cases}. \tag{43}$$

And note that the value of $\widetilde{V}_1^{\pi,\sigma_2}([0,0])$ only depends on $\pi_1(\cdot\,|\,[0,0])$ which can be represent by $\pi_1(0\,|\,[0,0])$. Plugging in equation 43 into equation 41, we have that when $\pi_1(0\,|\,[0,0]) \geq \frac{1}{2}$,

$$\begin{aligned}
\max_{\pi} \widetilde{V}_1^{\pi,\sigma_2}([0,0]) &= \max_{\pi_1(0\,|\,[0,0]) \geq \frac{1}{2}} 1 + (T-1)\pi_1(0\,|\,[0,0]) + (T-1)\sigma_2\big(1 - 2\pi_1(0\,|\,[0,0])\big) \\
&= 1 + (T-1)\sigma_2 + (T-1) \max_{\pi_1(0\,|\,[0,0]) \geq \frac{1}{2}} (1 - 2\sigma_2)\pi_1(0\,|\,[0,0]) \\
&= 1 + \frac{T-1}{2},
\end{aligned} \tag{44}$$

where the last equality holds by $\sigma_2 > \frac{1}{2}$ and letting $\pi_1(0\,|\,[0,0]) = \frac{1}{2}$. Similarly, when $\pi_1(0\,|\,[0,0]) < \frac{1}{2}$,

$$\max_{\pi} \widetilde{V}_1^{\pi,\sigma_2}([0,0]) = \max_{\pi_1(0\,|\,[0,0]) < \frac{1}{2}} 1 + (T-1)\pi_1(0\,|\,[0,0]) < 1 + \frac{T-1}{2}. \tag{45}$$

Consequently, we complete the proof by concluding that

$$\widetilde{V}_1^{\pi_{\mathsf{RSC}}^{\star,\sigma_2},\sigma_2}([0,0]) = \widetilde{V}_1^{\star,\sigma_2}([0,0]) = \max_{\pi} \widetilde{V}_1^{\pi,\sigma_2}([0,0]) = 1 + \frac{T-1}{2}. \tag{46}$$

### C.3 Auxiliary results of SC-MDPs and RSC-MDPs

**Facts about SC-MDPs.** For any state-confounded MDPs (SC-MDPs) $\mathcal{M}_{\mathsf{SC}} = \big\{\mathcal{S}, \mathcal{A}, T, r,$ $\mathcal{C}, \{\mathcal{P}_t^i\}, P^c\big\}$, denoting the optimal policy as $\pi^\star$ and the corresponding optimal SC-value function as $\widetilde{V}$, any policy $\pi$ satisfies the corresponding *state-confounded* Bellman consistency equation as below: