# OpenReview forum: "Seeing is not Believing: Robust Reinforcement Learning against Spurious Correlation"
_NeurIPS.cc/2023/Conference — NeurIPS 2023 poster_

### Official Review · Reviewer_27Kb · 2023-07-07

**Soundness:** 3 good
**Presentation:** 2 fair
**Contribution:** 2 fair
**Rating:** 6
**Confidence:** 3

**Summary:**

This work aims to make RL robust to distribution shifts by limiting their reliance on spurious correlations between state features.  This is achieved through a new RL algorithm designed in a variant of Robust MDPs extended to include a more structured uncertainty representation.

**Strengths:**

* This paper addresses a serious underlying flaw in the standard approaches for Robust MDPs, in that the uncertainty set is usually artificial an no alined to the ways we need our policies to be robust.

**Weaknesses:**

The language is often confusing, for instance the term "spurious correlations" is often used to refer specifically to spurious correlations between state variables.  This is very confusing as it overloads the term, and  makes it hard to refer to other spurious correlations (such as a spurious correlation between the agents actions and the reward or another agents actions).  Better to say "Spurious state correlations".

On a similar note "semantic uncertainty" is a confusing term that wasn't explained until deep into the paper.  It sounds like uncertainty **about**  semantics (which is how it has been used in the literature), but it is meant to be uncertainty that is not just a norm-ball perturbation over the transition function.  A better term would be "structured" uncertainty.

* Other  more minor confusing wordings make the paper more difficult to read than it needs to be:
	* I don't know what it means for portions of the state to "not have causality"
	* I don't know what it means for an uncertainty set to be "shaped by the unobserved confounder and sequential structure of RL"
	* I don't know what it means for some approach to be "superior in breaking spurious correlations"
	* There is a typo in : "Despite various types of uncertainty have been investigated in RL"


In addition it's unclear how sensitive the approach is to the degree of robustness.  It's a famous issue of RMDPs that the degree of robustness has to be carefully selected, too large and you have  an overly conservative policy, and too small you get no robustness.  Giving that part of the argument for this approach is alleviating that problem (as shown in Figure 3), a sensitivity analysis here would greatly improve the paper, as a main draw away from RMDPs is that they are very sensitive to the robustness parameter.

**Questions:**

How sensitive is the approach to changes in B% and K?

**Limitations:**

It is important to discuss somewhere how hyper-parameters (such as k and B%) were chosen, and could be chosen in a new domain.

---

> ### Author Rebuttal · Authors · 2023-08-09
>
> We gratefully thank the reviewer for valuable suggestions and the recognition of our proposed problem setting. In what follows, we provide our response to the reviewer's comments.
>
> ### **Q1. Change the term "spurious correlation" to "spurious state correlation".**
> Thanks for raising this point. We agree that "spurious correlation" is generally used between different variables of interest and may lead to confusion without further specification. So we shall explicitly introduce that this work focuses on the spurious correlation between state and change "spurious correlation" to "spurious state correlation" when it is better.
>
>
> ### **Q2: Change the term "semantic uncertainty" to "structured uncertainty".**
> Thanks for the valuable suggestion. We agree that the terminology "structured uncertainty" is better to describe our setting, where the uncertainty set is not a norm ball using some divergence function but a possible heterogeneous ball with a causal structure.
>
>
> ### **Q3: Study of the uncertainty level/degree sensitivity ($\beta\%$ and K).**
> Thanks for raising this question. Different from the traditional RMDPs that use a radius parameter $\sigma$ to control the size of the uncertainty ball, our RSC-MDP characterizes the uncertainty set by constructing samples inside it by perturbing the confounder over state space. To control the degree of the perturbation and hence implicitly control the uncertainty size, we introduce two hyper-parameters $\beta \%$ and $K$. We agree with the reviewer that the sensitivity analysis of these two parameters is critical so we conduct additional experiments and show the results in **General Response (2)** (two tables). The results provide three important findings:
> * **Our RSC-SAC is not sensitive to $\beta$.** Shown in the first table, the proposed RSC-SAC can perform well in both nominal and shifted settings --- keeping good performance in the nominal setting and achieving robustness, for a large range of $\beta\%$ (10%-80%). It verifies that RSC-SAC is not sensitive to hyperparameter choices.
> * **Our RSC-SAC is not sensitive to $K$.** We evaluate the proposed RSC-SAC using different $K = [32,64,\cdots, 1024]$ and achieve similar results, shown in the second table. It shows that RSC-SAC is not sensitive to the size $K$ of candidate samples for permutation.
> * **Performance-robustness tradeoff.** From the first table, when the ratio of perturbed data $\beta\%$ is very small (1%), RSC-SAC almost achieves the same results as vanilla SAC in nominal settings and there is no robustness in shifted settings. As $\beta\%$ increases (considering more robustness), the performance of RSC-SAC in the nominal setting gradually gets worse, while reversely gets better in the shifted settings (more robust). However, when the ratio is too large (>80%), the performances of RSC-SAC in both settings degrade a lot, since the policy is too conservative so that fails in all environments.
>
> ### **Q4: Other comments about writing.**
> Thanks for the careful reading and valuable suggestions about the writing. We address them below:
> * "...different portions of the state that do not have causality..." -> "different portions of the state that don't have correlations induced by unobserved confounder."
> * "an uncertainty set shaped by the unobserved confounder and sequential structure of RL." -> "an uncertainty set determined by some causal structure and unobserved confounder"
> * "an approach that is superiority in breaking spurious correlations..." -> "an approach that can achieve robust performance by avoiding learning useless spurious correlations"
> * "Despite various types of uncertainty have been investigated in RL,..." -> "Despite various types of uncertainty that have been investigated in RL,..."

---

> > ### Comment · Reviewer_27Kb · 2023-08-18
> > **Response to Rebuttal**
> >
> > Thank you for the clarifications.  Your comments both here and to the other reviews largely address my concerns, and I will be increasing my score accordingly.

---

> ### Author Response · Authors · 2023-08-14
> **Thanks for your insightful suggestions!**
>
> Dear reviewer,
>
> Thank you once again for investing your valuable time in providing feedback on our paper. Your insightful suggestions have led to significant improvements in our work, and we look forward to possibly receiving more feedback from you. Since the discussion period between the author and reviewer is rapidly approaching its end, we kindly request you to review our responses to ensure that we have addressed all of your concerns. Also, we remain eager to engage in further discussion about any additional questions you may have.
>
> Best,
>
> Authors

---

### Official Review · Reviewer_EdL3 · 2023-07-08

**Soundness:** 2 fair
**Presentation:** 2 fair
**Contribution:** 2 fair
**Rating:** 6
**Confidence:** 3

**Summary:**

The paper proposes a state-confounded (SC-)  and a robust-state confounded (RSC-) MDP formulation to account for setups where a confounder satisfying the backdoor criterion confounds the states. The RSC-MDP setup assumes that the confounder lies in an uncertainty set which is part of the MDP parameters, and the aim is to learn a value function that is the lower bound of the different value functions arising out of different instances of the RSC-MDP at different values of the confounder within its uncertainty set. They propose an empirical algorithm  to approximate the effect of perturbing the confounder by perturbing single dimensions of the states and setting them to values present in ‘otherwise nearby’ states (defined by distance between the two states in dimensions other than the dimension being edited). The paper claims that this indirectly estimates the effects of perturbing the confounder. They learn a  graph-based model that is supposed to be a causal model, to generate new transitions (given a state and action s_t, a_t, generate s_t+1, r_t) based on the perturbed states and (original) action pairs. They augment SAC by randomly replacing part of the sampled batches with the generated transitions and learn as usual, and show that this recipe is robust to the spurious correlation in the nominal envs they define. All experiments are in the state space (non-vision inputs).


**Strengths:**

1. The paper studies an important problem and proposes a set of  benchmarks with handcrafted spurious correlations in a CARLA environment and in a robosuite environment which could be useful for more future works to study spurious correlations in RL.
2. The RSC-MDP formulation could also be useful, although I’m not sure how much the notation and proofs scale to an arbitrary number of confounding variables.


**Weaknesses:**

1. The paper structure is somewhat confusing to me: The MDP formulations and the empirical algorithm don’t seem to have much of a connection and seems like both were developed independently - please correct me if I’m missing something here. It would be great to further clarify exactly how the algorithm is helping to solve the robust SC-MDP.

2. The notion of semantic uncertainty is not very precise or clear (I don’t think there’s any references either), and could be removed in my opinion since the RSC-MDP formulation can simply use the “uncertainty set” phrasing.

3. The appendix suggests that the training environments have perfect spurious correlations: the correlation isn’t broken in even a few instances of the training environments, so for eg. there would never be a (non-generated) transition in the SAC buffer that would have the brightness value low in its state space value, in the case that the training envs are following the correlation regime where brightness and traffic are both high during the day. It’s unclear to me then, why the causal model would ever generate a state value that doesn’t also have the brightness value set to high (same logic for the perturbation procedure). Am I missing something else here? Or are the training envs set up such that an agent will train on two kinds of envs at once - night time with less traffic and day time with more traffic, and so it s possible to see samples with the brightness value set to both low and high across otherwise nearby states?
It would be great to clarify what exactly is the train and test env distribution and what is expected of the graphical causal model - is it to generate counterfactual transitions (counterfactual to what?) and conclusively show that it is actually doing that by visualizing or plotting some property of the samples it is generating. It's possible that I have misinterpreted something here, and I'm willing to raise my score if the authors' response clarifies some of these questions.

4. I would also suggest mentioning relevant related work explicitly aimed at resolving causal confusion in online [1] and offline RL [2] (and comparing to the closest version if applicable).

5. It would be great to include the training curves for the results in Table 1 as well, since that gives insight into whether other methods simply converge later to a similar highest reward, or whether they are entirely limited in their ability to converge to a similarly high reward as the best performing method.

[1] Resolving Causal Confusion in Reinforcement Learning via Robust Exploration . Clare Lyle, Amy Zhang, Minqi Jiang, Joelle Pineau, Yarin Gal. ICLR Self-Supervised RL Workshop 2021.

[2] Can Active Sampling Reduce Causal Confusion in Offline Reinforcement Learning? Gunshi Gupta, Tim G. J. Rudner, Rowan Thomas McAllister, Adrien Gaidon, Yarin Gal, NeurIPS Offline RL Workshop 2022, CleaR 2023


**Questions:**

1. How is the experiment to answer R3 designed? It's unclear how to test the w/o P^{c} case since the confounder isn't known anyway?

2. As stated previously, it would help to get more insights into how novel the generated transitions from the causal model really are.

3. Have you observed or quantified the robustness-performance tradeoff mentioned in line 263, which will lead to a performance drop in some envs? I expect the perturbation procedure of states to only work for specific kinds of envs, so it would be good to show the failure cases as well.



**Limitations:**

Given how heuristic the technique of generating perturbed states is (swapping single dimensions between otherwise nearby states), the paper should properly discuss the many challenges of scaling this approach to a high-dimensional state space like that in visual domains.

---

> ### Author Rebuttal · Authors · 2023-08-09
>
> We gratefully thank the reviewer for recognizing our contributions to problem formulation and the creation of a useful benchmark! We provide our response below:
>
> ### **Q1: The connection between the proposed problem formulation (robust SC-MDP) and the empirical algorithm.**
> The empirical algorithm implicitly constructs and manipulates the unknown uncertainty set proposed in the robust SC-MDP formulation. To solve the RSC-MDP problem, we need to optimize over an unknown transition kernel uncertainty set -- structured ball around some nominal confounder value. Due to lacking the information on both the structure and nominal value, the empirical algorithm approximates the uncertainty set by constructing samples within it -- perturbing the states to mimic different confounder values around the nominal value.
>
> ### **Q2: The notion of semantic uncertainty.**
> Thanks for the valuable suggestion. We agree that the term "semantic uncertainty" may cause confusion without further explanation and references. We replace it with "structured uncertainty" since the uncertainty set is determined by some underlying task-specific causal structure.
>
> ### **Q3: What exactly is the train and test env distribution? Show that our approach is actually generating desired samples by visualizing.**
> We use Brightness to illustrate the train and test envs. We assume the latent confounder has 4 values: $z=0$ (generate day-heavy samples), $z=1$ (generate night-light samples), $z=2$ (generate day-light samples), and $z=3$ (generate night-heavy samples). The training environment is generated with $z=0,1$ and the testing environment is generated with $z=2,3$. Therefore, the shift between training and testing comes from different compositions of brightness and traffic. **Since we cannot explicitly set $z=2,3$ in training, our perturbation method simulates the effect of setting $z=2,3$ by perturbing the training data (with $z=0,1$).**
>
> **We visualize the generated trajectories of Lift in General Response (3) (includes figures and detailed explanation) with comparisons to original trajectories in training env, showing that our perturbation method can generate counterfactual examples to the unobserved confounder.**
>
> ### **Q4: Mention the related works [1][2] explicitly.**
> Thanks for providing important related works [1][2]. We add the following discussion to our related work:
> > [1] designs an exploration algorithm to conduct intervention and improve state-action coverage to avoid biased data collection. However, our work deal with a more general setting, where the testing environment could vary from the training one, leaving the exploration strategy inapplicable.
>
> > [2] proposes an uncertainty-based acquisition function to resample from the data buffer, learning more from the samples that do not have spurious correlations. This method does not explicitly handle spuriousness and could fail when spurious correlations exist in most samples.
>
> **We add [2] as a baseline and show the results in General Response (1)**.
>
>     [1] Resolving Causal Confusion in Reinforcement Learning via Robust Exploration. Lyle et.al. ICLR Workshop 2021
>     [2] Can Active Sampling Reduce Causal Confusion in Offline Reinforcement Learning? Gupta et.al. CleaR 2023
>
> ### **Q5: The robustness-performance tradeoff.**
> The robustness-performance tradeoff is mainly determined by $\beta\%$, the ratio of the perturbed data.
> To investigate more on this tradeoff, we add an ablation study on $\beta\%$ and show the results in **General Response (2)**. Two important messages:
> * **The existence of the tradeoff.** When the ratio of perturbed data $\beta\%$ is very small (1%), RSC-SAC achieves similar results as vanilla SAC in nominal settings and shows no robustness in shifted settings. As $\beta\%$ increases (considering more robustness), the performance of RSC-SAC in the nominal setting gets worse, while reversely the performance gets better in the shifted settings (more robust). However, when the ratio is too large (>80%), the performances of RSC-SAC in both settings degrade a lot, since the policy is too conservative.
> * **Our RSC-SAC maintains both robustness and good performance.** Although the tradeoff exists, the proposed RSC-SAC can perform well in both nominal and shifted settings for a large range of $\beta\%$ (10%-80%) -- keeping good performance in the nominal setting and achieving robustness.
>
> ### **Q6: How are the experiments designed to answer R3? How to test the w/o $P^{c}$ case since the confounder is unknown?**
> * **How to design 3 ablation studies in R3.** All 3 ablation methods are modified on RSC-SAC. For "w/o $\textbf{G}\_\phi$", we use a full graph to replace the learnable causal graph during training. For "w/o $P_{\theta}$", we replace the entire causal model with a fully-connected NN. For "w/o $P^c$", we don't do the perturbation of state introduced in Section 4.1.
> * **The experiments of w/o $P^{c}$.** We denote the unknown confounder distribution as $P^{c}$. "w/o $P^{c}$" means that we don't permute the dimensions of states in Equation 6 in the data generation process. So the generated data will still be in the same distribution as the nominal training environment. We change it to "w/o $P^{c}$ perturbation" in Table 4 to avoid confusion.
>
> ### **Q7: Discussion about scaling our approach to high-dimensional state spaces such as visual domains.**
> We add more discussion in the conclusion section. Please check **General Response (4)**.
>
> ### **Q8: Adding the training curves for the results in Table 1.**
> The training curve is shown in the bottom row of Figure 5, where we display the testing reward on the shifted testing environment with the training step increases. After a long training process, the proposed method RSC-SAC still outperforms all baselines. To verify that all methods are converged (enter a flat area), we also plot the testing reward on the nominal environment during training in the top row of Figure 5.

---

> ### Author Response · Authors · 2023-08-14
> **Thanks for your insightful suggestions!**
>
> Dear reviewer,
>
> Thank you once again for investing your valuable time in providing feedback on our paper. Your insightful suggestions have led to significant improvements in our work, and we look forward to possibly receiving more feedback from you. Since the discussion period between the author and reviewer is rapidly approaching its end, we kindly request you to review our responses to ensure that we have addressed all of your concerns. Also, we remain eager to engage in further discussion about any additional questions you may have.
>
> Best,
>
> Authors

---

> > ### Comment · Reviewer_EdL3 · 2023-08-16
> > **Thanks for your response and new results**
> >
> > Dear Authors,
> >
> > Thankyou for adding the dynamics visualization as well as another baseline comparing to the proposed method. I think the comparison helps to highlight how the proposed method tackles a problem that is not addressed by the baseline - that of compositional generalization (please correct me if I'm conflating different things).
> >
> > I had the following question before I can update my score:
> > Is the following correct?
> > To generate a training example for the causal graph model the following procedure will be followed: We will take a transition (presumably from a daylight driving scenario) with day-light=high, and traffic=heavy, (and the other dimensions being set to some values), and assuming that we decide to perturb the day-light dimension of the current state, we look for states in the buffer where the day-light value is very different, however the other values are similar (therefore likely traffic=heavy, and day-light=low, and other dimensions being similarly valued).
> > Now the s_{t+1} that is used to supervise the causal model output is still the same as that in the original transition (presumably day-light=high and traffic=heavy).
> >
> > I'm confused why the causal model should be expected to predict that a day-light heavy state transitions to a state that is day-light low -- unless the causal graph completely ignored the day-light dimension entirely. Getting the model to ignore the day-light dimension entirely is what I assume is being enforced by the sparsity loss - which leads me to the hypothesis that essentially what the causal graph is doing is helping to enforce the principle of "looking or depending on as few input dimensions as possible". This is important to note, as there is prior work [1] testing this idea for imitation learning by using some sort of dropping out of the input representation of a policy, (they don't make the assumption of operating in the state space). This also suggests that a similar trick might work if for example you removed the causal graph and perturbation mechanism entirely and trained a SAC policy with a input dropout mechanism on the state space inputs since it might enforce the same invariance to irrelevant dimensions. I think comparing the proposed method to this simple baseline will greatly help to refute this claim, as well as further explain what benefit the causal model is really bringing.
> >
> > I do think the proposed method has potential merit, but given the complexity and added assumptions of the additional components in the proposed method, I would like to be sure by dissecting rigorously where the gains are coming from.
> > Thanks for your responses so far, and looking forward to engaging further.
> >
> > [1] Object-Aware Regularization for Addressing Causal Confusion in Imitation Learning
> > Jongjin Park, Younggyo Seo, Chang Liu, Li Zhao, Tao Qin, Jinwoo Shin, Tie-Yan Liu

---

> > > ### Author Response · Authors · 2023-08-17
> > > **Response to follow-up questions**
> > >
> > > Thank you for engaging in the discussion and providing insightful feedback! We provide new experiments as well as analyses to answer your questions.
> > >
> > > ### **Q1: The proposed algorithm (RSC-SAC) tackles compositional generalization**
> > >
> > > Yes, we totally agree that there is a strong connection between compositional generalization and breaking spurious correlation. Actually, In section 5.1, we propose two kinds of environment settings of spurious correlation:
> > > 1.  **Distraction correlation**: Between task-relevant and task-irrelevant features. The task-irrelevant feature is a distractor and should be ignored by the policy (random dropping may solve this);
> > > 2.  **Composition correlation**: Between two task-relevant features. This exactly describes the compositional generalization setting, where the testing environments contain new combinations of task-relevant features (random dropping cannot help).
> > >
> > >
> > > ### **Q2: Correctness of the example of daylight driving scenario**
> > >
> > > The reviewer accurately addresses most aspects of the example, with the exception of the last sentence. After having the perturbed $\{s_t, a_t\}$ (e.g., traffic=heavy, day-light=low), we infer $s_{t+1}$ from our causal model with counterfactual generation. We expect a new $s_{t+1}$ by imagining a different value in $s_t$. In experiments, we observe that most $s_{t+1}$ are different from the original one and have traffic=heavy and day-light=low.
> > >
> > >
> > > ### **Q3: Adding experiments -- prior work OREO [1] as a baseline**
> > >
> > > We evaluate the performance of OREO [1] with different ratios of dropping ($\alpha$) on the Brightness (Distraction) and Behavior (Composition) environments. The results are shown below:
> > > |Method|Ours|OREO ($\alpha$=0.1)|OREO ($\alpha$=0.2)|OREO ($\alpha$=0.3)|OREO ($\alpha$=0.4)|OREO ($\alpha$=0.5)|
> > > |:-:|:-:|:-:|:-:|:-:|:-:|:-:|
> > > |Brightness (nominal)|0.92±0.31|**0.973 ± 0.239**| 0.739 ± 0.327| 0.371 ± 0.229| 0.26 ± 0.243| 0.207 ± 0.197|
> > > |Brightness (shifted)|**0.99±0.11**|0.891 ± 0.153| 0.562 ± 0.18| 0.256 ± 0.087| 0.182 ± 0.087| 0.128 ± 0.059|
> > > |Behavior (nominal)|**1.06±0.07**| 1.04 ± 0.104| 0.989 ± 0.122| 0.855 ± 0.259| 0.715 ± 0.282| 0.482 ± 0.224|
> > > |Behavior (shifted)|**1.02±0.09**| 0.517 ± 0.208| 0.541 ± 0.121| 0.553 ± 0.107| 0.509 ± 0.169| 0.366 ± 0.137|
> > >
> > > 1. Our method outperforms OREO in both Brightness (shifted) and Behavior (shifted) environments. We find that OREO indeed achieves robustness (but still worse than ours) in Brightness, which only contains the distraction correlation.
> > > 2. The advantage of our method (swap values of some dimensions of states) over OREO (drop some dimensions of states) could be explained from two aspects:
> > >     * **Our method has compositional generalization, while OREO does not.** Swapping dimensions within the state creates new compositions of features that can address both compositional generalization and distraction issues. However, dropping information (OREO) can only deal with the distraction issue but does not make policy generalizable to unseen feature combinations.
> > >     * **Dropping dimension of states loses information.** [1] focuses on image input which has a spatial structure -- still contains enough information after dropping some dimensions. However, in our state input setting (each dimension has semantic meaning), dropping dimensions of the state could cause severe information loss. The evidence is that increasing the ratio of dropping dramatically degrades performance.
> > >
> > >
> > > [1] Object-Aware Regularization for Addressing Causal Confusion in Imitation Learning Jongjin Park, Younggyo Seo, Chang Liu, Li Zhao, Tao Qin, Jinwoo Shin, Tie-Yan Liu

---

> > > > ### Comment · Reviewer_EdL3 · 2023-08-17
> > > > **Thanks for your response and explanation of the contribution of the causal graph**
> > > >
> > > > Dear Authors,
> > > > Thank you for your response answering my questions and promptly adding the dropout baseline to help with the analyses.
> > > > To confirm that I understood the response correctly:
> > > > 1. In the response to Q2 you say that a new s_t+1 would be expected when the s_t is perturbed, does that mean the causal graph is never trained on the perturbed samples ie. it is trained on the non-perturbed transitions in the dataset, and then used for inference on the perturbed samples? (since we wouldn't have the correct supervision for s_t+1 for them)
> > > > What is the intuition for why the causal model would be correct on inference on the perturbed samples with novel combinations of the features (If not for the sparsity inducing bias)? My follow-up question would be, how much is the performance reliant on the accuracy of the different components of the transition (reward and next state, or even just the combination of the perturbed s_t and original a_t existing in the buffer) - are both reward and next state prediction accurate? If the tasks are sparse reward I suppose most of the transitions just have a zero reward being predicted?
> > > > 2. About the OREO baseline : Thanks for adding this. Is this the exact method from the OREO paper with the VQVAE component (which IMO is not needed since we are not operating in the image space here and already have all the relevant features as a compact input space), or do you mean that you used their dropping out mechanism on your input space and call that baseline OREO in the paper?
> > > >
> > > > In the meanwhile I have raised my score to Weak Accept in light of the new baselines and the added clarifications.
> > > > Thanks!

---

> > > > > ### Author Response · Authors · 2023-08-17
> > > > > **Response to follow-up questions**
> > > > >
> > > > > Thanks for increasing the score! In response to the clarification questions:
> > > > > 1. The causal graph is not trained on perturbed samples due to the absence of the ground truth. The intuition behind the correct prediction is indeed the **sparsity inducing bias**, as pinpointed by the reviewer. Our hypothesis posits that the subsequent state holds greater significance than the reward. This is based on the observation. We hypothesize that the next state is more important than the reward since in most cases the reward is not influenced by the changing of one dimension of the state.
> > > > > 2. In our implementation, we just use the dropping-out mechanism in this baseline without using VQ-VAE. To be more accurate, we will name this method 'state dropout' instead of 'OREO'.

---

> > > > > > ### Comment · Reviewer_EdL3 · 2023-08-17
> > > > > > **Follow-up question**
> > > > > >
> > > > > > I see, in that case I have two questions/points of observation:
> > > > > > 1. I'm confused about why a sparsity-inducing loss helps compositional generalization. To me it sounds like all tasks are essentially still challenging the distractor-avoiding skill of an agent, and the compositional case could probably be reformulated as a distracting case. Could you provide an example of what the spurious correlation is between the inputs (state dimensions) and outputs (action dimensions of a policy) in the compositional data, without the notion of distractors coming in?
> > > > > > 2. I don't think random dropout would be the closest analogy here: apologies if my previous comparison to OREO added confusion into the mix - I wanted to refer to their insight that dropping out meaningful parts of the inputs or representations could help with robustness of a policy. In this case, I think a sparsity loss is what would help to understand if a learned/guided version dropout by itself is largely responsible for the performance improvements. This could be implemented either as a simple matrix multiplication before the first layer of a policy model with trainable matrix weights penalized by L1 loss (similar to what is happening in the causal graph), or even an L1 loss on the first or second layer MLP weights of the policy, to force it to attend to as few input dimensions as possible (again similar to what the causal graph is doing).
> > > > > >
> > > > > > Lastly, thank you for engaging in the discussion and helping to better understand the workings of your method.

---

> > > > > > > ### Author Response · Authors · 2023-08-18
> > > > > > >
> > > > > > > 1. **The compositional generalization mainly comes from two designs**: (1) (the major one) We swap the dimensions within the state, which creates new compositions of useful features that are unseen in the original training data. (2) We infer a reasonable $s_{t+1}$ according to the modified $s_t$, in which the sparse causal model could provide an accurate prediction.
> > > > > > >
> > > > > > > 2. **One example of the compositional setting** is the Door environment (Figure 8 in Appendix), where there is a spurious correlation between **the distance between the robot and the door** and **the position of the handle**. Both of these features are important for the robot to open the door. Specifically, the training env has [DoorDist=far, HandlePos=high] and [DoorDist=near, HandlePos=low]. After training policy on such a dataset, we observe that the robot always tries to reach the [high] position when the [DoorDist=far] even if it observes [HandlePos=low]. This can be explained as a failure of compositional generalization due to the spurious correlation between state and action.
> > > > > > >
> > > > > > > 3. We apologize for the misunderstanding of the baseline setting mentioned by the reviewer. So we add a new baseline (denoted as Sparse-SAC) that has an **additional L1 loss $\alpha \|W\_{1}\|\_1$** during the optimization, where $W\_{1}$ is the parameters of the first MLP layer of the policy and value networks and $\alpha$ is the weight. The results are shown below.
> > > > > > >
> > > > > > > Type of spurious correlation|Environment|Ours|SAC (non-robust)|Sparse-SAC ($\alpha$=1)|Sparse-SAC ($\alpha$=10)|Sparse-SAC ($\alpha$=100)|Sparse-SAC ($\alpha$=1000)|
> > > > > > > |:-:|:-:|:-:|:-:|:-:|:-:|:-:|:-:|
> > > > > > > |Distration|Brightness(nominal)|0.92±0.31|1.00±0.09|1.01±0.10|1.02±0.10|**1.03±0.10**|0.90±0.29|
> > > > > > > |Distration|Brightness(shifted)|**0.99±0.11**|0.56±0.13|0.69±0.2|0.87±0.18|0.83±0.13|0.72±0.29|
> > > > > > > |Composition|Behavior(nominal)|**1.06±0.07**|1.00±0.08|0.98±0.115|1.01±0.08|0.95±0.13|0.67±0.21|
> > > > > > > |Composition|Behavior(shifted)|**1.02±0.09**|0.13±0.03|0.16±0.032|0.15±0.04|0.10±0.02|0.08±0.02|
> > > > > > >
> > > > > > > Two important findings based on the results:
> > > > > > > * The sparsity regularization can improve the robustness of SAC in the setting of distraction spurious correlation (Brightness). This verifies the hypothesis of the reviewer that sparsity can address the distraction issue.
> > > > > > > * The sparsity regularization does not help with the composition spurious correlation (Behavior), which indicates that purely using sparsity regularization can not improve compositional generalization.
> > > > > > >
> > > > > > > We will add all these findings in the paper to illustrate the advantage of our method. We really appreciate the reviewer for proposing this interesting and effective baseline to improve the quality and presentation of our work, making the contribution more convincing and clearer.

---

> > > > > > > > ### Comment · Reviewer_EdL3 · 2023-08-21
> > > > > > > > **Thanks for adding sparsity loss experiments**
> > > > > > > >
> > > > > > > > Dear Authors,
> > > > > > > >
> > > > > > > > Thanks again for your engagement, and I'm glad that the paper has benefitted from it by including baselines that further help to dissect and highlight the key advantages of the proposed method.
> > > > > > > > I still have some reservations/unanswered questions about why the causal graph model can generalize to give the correct prediction for a perturbed s_t. You also seem to attribute the accuracy of it's predictions to the sparse-ness of the causal graph - which to me suggests that this should then be achievable through the sparsity loss as well, unless this kind of sparsity is adaptive in some way and doesn't always choose to ignore the same input dimension always (thus making it different from the distractor correlation regime).
> > > > > > > > I am inclined to increase my score to reflect a stronger recommendation towards acceptance if the authors can provide an answer to this (or add analysis to demonstrate or refute the above point - however I understand there is a time constraint and more experiments are strictly optional).
> > > > > > > >
> > > > > > > > Thanks!

---

> > > > > > > > > ### Author Response · Authors · 2023-08-21
> > > > > > > > >
> > > > > > > > > Thanks for your question "Why the causal graph model can generalize to give the correct prediction for a perturbed s_t?"
> > > > > > > > >
> > > > > > > > > Unfortunately, we currently **do not** have theoretical proof of why the causal graph can accurately predict $s_{t+1}$, so we would like to empirically illustrate some cases where the prediction of our causal model is more accurate than using a neural network without causal. In general, the advantages are induced by the Independent Causal Mechanism (ICM) [1] that helps address the out-of-distribution issue.
> > > > > > > > >
> > > > > > > > > Assuming there are spurious correlations between two dimensions $s_t^1$ and $s_t^2$ of state $s_t$, we swap the values of two samples for the dimension $s_t^1$.
> > > > > > > > > * If we use a dense neural network (without causal graph) to predict $s_{t+1}$, we may get unpredictable results since the perturbed $s_t$ (a new composition of $s_t^1$ and $s_t^2$) is an out-of-distribution input to the neural network. In our experiments with the dense neural network, the predicted $s_{t+1}$ usually contains invalid values or the same value as the original samples. (We cannot provide the figure since we are not allowed to update the pdf file.)
> > > > > > > > > * If we use a sparse causal graph to predict $s_{t+1}$, we can benefit from the Independent Causal Mechanism (ICM), which learns independent sub-causal graphs between $s_t$ and $s_{t+1}$. Taking one ideal case with ICM as an example, when dimensions 1 and 2 are independent of each other in the causal graph (i.e., the causal graph is $s_t^1 \rightarrow s_{t+1}^1, s_t^2 \rightarrow s_{t+1}^2$), the prediction of $s_{t+1}^1$ is independent of $s_t^2$ and the prediction of $s_{t+1}^2$ is independent of $s_t^1$. Benefited from the decoupling, our causal model can accurately predict $s_{t+1}^1$ (only based on $s_t^1$) and $s_{t+1}^2$ (only based on $s_t^2$) since the dataset contains useful $s_t^1$ and $s_t^2$ separately. Even in non-ideal cases where dependency exists, we can still (empirically) benefit from the sparse connection to avoid $s_{t}$ becoming an out-of-distribution sample to the prediction model to increase the accuracy.
> > > > > > > > >
> > > > > > > > > [1] Parascandolo, Giambattista, Niki Kilbertus, Mateo Rojas-Carulla, and Bernhard Schölkopf. "Learning independent causal mechanisms." In International Conference on Machine Learning, pp. 4036-4044. PMLR, 2018.

---

### Official Review · Reviewer_uQEX · 2023-07-09

**Soundness:** 3 good
**Presentation:** 3 good
**Contribution:** 2 fair
**Rating:** 6
**Confidence:** 3

**Summary:**

This paper aims to address the spurious correlation challenge that arises in RL. Such correlation is typically useless to decision-making but may be learned by agents, which leads to failure in applying to unknown test cases. To this end, the authors proposed a novel RSC-MDP framework that models the spurious correlation challenge with an unobserved confounder. The authors then justify that previous robust MDP methods may fail on the RSC-MDP framework due to inaccurate uncertainty modeling. The authors proposed a heuristic method to handle the RSC-MDP challenge authors then conducted extensive experiments to compare the proposed method and SOTA baselines for robust RL. Experiments justified that the proposed method has better performance for the proposed spurious correlation attacks.

**Strengths:**

1. The paper is very well written, with a clear explanation of motivation and well organized problem setup and experimental design.
2. The authors provide theoretical justification that the proposed robustness is milder and more accurate than the traditional robust MDP.
3. The authors provide novel experimental design to assess the effect of spurious correlation in the state space of RL. The proposed baseline has the potential to be adopted by future study.

**Weaknesses:**

1. In the proposed driving example, the confounder issue spans over multiple steps of interactions, whereas the latent variable c introduced in RSC-MDP only affects one step, and each step is affected independently (i.e., $c_t$ is independently generated for all $t$). In reality, the brightness and traffic density do not normally get perturbed per each step of the transition. Do the proposed RSC-MDPs cover the targeted spurious correlation problem?

2. Seems that the confounding within stat is not affecting the distribution of transition; that said, the observation still follows the distribution of trajectories after marginalizing the confounder $c$. To me, the spurious correlation challenge of RSC-MDPs is more of a robustness challenge than a confounding challenge to me, where the offline trajectory distribution is typically different from the interventional transition dynamics. How does the spurious correlation within each state affect the learning of transition P(s_{t+1} | s_t, a_t), hence affecting regular RL algorithms?

3. Could authors provide more reasoning on the proposed heuristic perturbation in approximating the true underlying perturbation of latent variables? Intuitively it seems to align well with the motivative driving example, but it seems that the perturbation proposed in equation (6) introduces extra correlations within states. In particular, sample $s^i_k$ selected to replace $s^i_t$ is based on a rule that explicitly involves $s^j_t$ for $j\neq i$. Do we worry that such introduced correlation ruins training for similar reasons of the targeted spurious correlation?

Misc:

line 81: discounted infinite-horizon -> finite horizon?

line 220: $\sigma_2 \in (3/4, 1]$ according to the proof.

**Questions:**

See Weaknesses.

**Limitations:**

N.A.

---

> ### Author Rebuttal · Authors · 2023-08-09
>
> We appreciate the reviewer for valuable suggestions and the praise of our systematic formulation and experiments. In what follows, we provide our response to the reviewer's comments.
>
>
> ### **Q1: Does the proposed RSC-MDP formulation cover the targeted spurious correlation problems in the real world and our experiments?**
> Thanks for pointing out this important question! **Our RSC-MDP is a general formulation designed to cover most types of unobserved confounders.**  Note that the proposed RSC-MDP allows but not requires the confounder variable to vary across different time steps, i.e., $c_t$ can have different values for different $t$ but not forced to. So it includes the case that the confounder variable keeps the same in the entire horizon, i.e., $c_1 = c_2 = \cdots c_T$. In addition, RSC-MDP can deal with other scenarios when the confounder variable $c_t$ changes over time, e.g., the car is driving through sudden weather changes.
>
> ### **Q2: How does the spurious correlation within each state affect the learning of transition $P(s_{t+1} |s_t, a_t)$, hence affecting regular RL algorithms?**
> Thanks for raising this insightful question. Recall the transition kernel for the proposed state-confounded MDP (SC-MDP) -- $s_{t+1} \sim \mathcal{P}\_t(\cdot | s_t, a_t, c_{t})$. The spurious correlation -- represented by an unknown confounder distribution $P_t^c$ (i.e., the confounder $c_t \sim P_t^c$) will determine the expected transition kernel $\sum_{c_t} \mathcal{P}\_t(\cdot | s_t, a_t, c_{t})  P_t^c(c_t)$. So the expected transition kernel will change if $P_t^c$ changes. The reviewer is correct that if the distribution $P_t^c$ is fixed, the influence of confounder on the transition kernel will not exist after marginalizing w.r.t. $c_t$. However, in this work, we desire to learn a policy that can address possible perturbation of the confounder distribution $P_t^c$ --- robust SC-MDP. In this case, the standard RL algorithm may learn the transition kernel based on one confounder distribution $P_t^c$ (spurious correlation) and fail catastrophically when $P_t^c$ varies in the testing environment.
>
> ### **Q3: More reasoning on the proposed heuristic perturbation for approximating the true underlying perturbation of the latent variables?**
> Thanks for raising this valuable question. We would like to answer this question from two aspects:
> * **Why our perturbation within the state approximates changing latent variables.** We provide more explanation by showing a concrete example, which visualizes the original state trajectories and the generated trajectories by our perturbation algorithm. Please refer to the figure in **General Response (3)**. In the nominal environment, the green cube is always initialized on the left part of the table and the red cube is initialized on the right part. We assume the latent variable $z$ is discrete and has 4 values: $z=0$ (generate green-left samples), $z=1$ (generate red-right samples), $z=2$ (generate green-right samples), $z=3$ (generate red-left samples). The nominal environment only contains cases with $z=0,1$, which has a strong spurious correlation between color and position (Figure (a)). Without explicitly setting $z=2,3$ (we can't do this during training to unobserved $z$), we directly perturb the states to mimic the effect of controlling $z$, which gives us the samples in Figure (c) that contains the cases with $z=2,3$ (green-right and red-left samples).
> * **Why perturbation within state works and doesn't involve additional spurious correlation.** The correlation between two dimensions in a state could be either causation or spurious correlation. To avoid the neural network overfits to harmful spurious correlation, we randomly perturb a small portion of the data ($\beta\%$) by swapping some dimensions of states to regularize the model. Since this is a random perturbation, it does not introduce additional spurious correlation. As we only perturb a small portion of the data, the model can still learn meaningful features from the remaining large portion of the data.
>
> ### **Q4: Other minor comments.**
> Thanks for improving the writing of our work. We have revised and polished the paper accordingly to the reviewer's comments.

---

> ### Author Response · Authors · 2023-08-14
> **Thanks for your insightful suggestions!**
>
> Dear reviewer,
>
> Thank you once again for investing your valuable time in providing feedback on our paper. Your insightful suggestions have led to significant improvements in our work, and we look forward to possibly receiving more feedback from you. Since the discussion period between the author and reviewer is rapidly approaching its end, we kindly request you to review our responses to ensure that we have addressed all of your concerns. Also, we remain eager to engage in further discussion about any additional questions you may have.
>
> Best,
>
> Authors

---

### Official Review · Reviewer_3Qte · 2023-07-10

**Soundness:** 3 good
**Presentation:** 2 fair
**Contribution:** 3 good
**Rating:** 6
**Confidence:** 4

**Summary:**

The paper studies how to develop more robust reinforcement learning algorithms when spurious correlation exists in the observation space. The paper studies the problem from a causal perspective and present robust state confounded MDP as the problem formulation. The paper proposes an algorithm to solve this problem formulation by learning the structural causal model. The paper demonstrate experimentally that the proposed method is more robust to shift in the environment compared to baselines.


**Strengths:**

The paper is mostly extremely well written and easy to follow. The structure of the paper makes sense, which first discusses the formulation of the problem in the form of robust state confounded MDP and comparison to existing formulation. The algorithm the paper proposes is intuitive and the improvements over the baseline in the experiment section are clear.


**Weaknesses:**

The discussion around related work focuses on a few most relevant papers and does not address the broader literature if I am not mistaken. The problem of spurious correlation is known within the community and various prior works [1, 2]. Please see [2] for example, which also learns the dynamics model to alleviate the issue of spurious correlation. These prior works tackle this problem even though they did not propose the same robust state confounded MDP that the paper proposes. I would have liked to see a more thorough comparison with the broader literature in a related work section.

On page 3, the paper writes “people usually prescribe the uncertainty set of the transition kernel using a  heuristic and simple function rho with a relatively small sigma”. However, because of the lack of a related work session, I do not see evidence for this statement.

[1] https://ben-eysenbach.github.io/rpc/
[2] https://arxiv.org/abs/1909.11373


**Questions:**

Please see my questions about related work

**Limitations:**

Please see my questions about related work.

I would have like to see the limitations that the method currently only work for state-based policies more prominently displayed and discussed.

---

> ### Author Rebuttal · Authors · 2023-08-09
>
> We would like to express our gratitude to the reviewer for their insightful feedback. We are glad to know that the reviewer recognizes the novelty of our contributions, the clarity of our problem formulation, and the empirical algorithm that sufficiently shows the advantages compared to baselines. We provide our response to the questions below.
>
>
> ### **Q1: Comparisons with broader literature such as [1][2].**
> Thanks for the important question and providing the important related works [1][2].
> *  We have included a thorough related work section in Appendix B in the original manuscript with comparisons to literature about 1) other related RL formulations, 2) robustness in RL investigations, and 3) spurious correlation in RL. Within Appendix B.3, we discussed different spurious correlation types that have been considered in the RL community.
> * We will definitely add [2] as a broader reference as the reviewer suggested since it also addresses spurious correlation in RL. We want to note that [2] is kind of far from our topic since 1) [2] sought to solve multi-task RL, while we focus on single-task RL; 2) the spurious correlation that [2] deals with is largely different from the one considered in this work: [2] considers the spurious correlation between the dataset distribution and the dataset identity to design a better task inference module, while the spurious correlation considered in this work is between different states.
> *  After carefully reading [1], we found that it is not quite related to spurious correlation but is implicitly related to robustness in RL. So we will add it to the related section in Appendix B.2. [1] sought to learn a 'simple' policy by minimizing the used information to seek better performance in standard RL, which turns out to have some robustness benefits.
>
>
> ### **Q2: Related works about robustness in RL.**
> For the claim mentioned by the reviewer "people usually prescribe the uncertainty set of the transition kernel using a heuristic and simple function rho with a relatively small sigma" on page 3, we add references [3][4] to the end of the sentence to support. A more detailed review of the existing investigated uncertainty set can be referred to in Appendix B.2, where we thoroughly summarize that most existing works use task structure-agnostic and heuristic 'distance' such as KL divergence and total variation.
>
>
> ### **Q3: More discussion about the potential of the proposed method which currently only work for state-based policies.**
> We add more discussion in the last section:
> > The current method is of great potential to be applied to more complicated problems. In particular, the proposed method requires swapping the dimensions of states to break spurious correlation, where it implicitly assumes that each dimension has semantic meaning. In more complicated problems such as high-dimensional image states, each dimension becomes a pixel without semantic meaning anymore and the dimension size may explode. To extend our method to such cases, we can leverage existing state abstraction techniques [5,6] to project images to a low-dimensional latent space, where each dimension represents a semantic feature. Then, our method can be straightforwardly applied to the low-dimensional latent space.
>
>     [1] Robust predictable control. Eysenbach, Ben, Russ R. Salakhutdinov, and Sergey Levine. Neurips 2021
>     [2] Multi-task batch reinforcement learning with metric learning. Li, Jiachen, Quan Vuong, Shuang Liu, Minghua Liu, Kamil Ciosek, Henrik Christensen, and Hao Su. Neurips 2021
>     [3] Toward theoretical understandings of robust Markov decision processes: Sample complexity and asymptotics. Yang, Wenhao, Liangyu Zhang, and Zhihua Zhang. The Annals of Statistics 50.6 (2022): 3223-3248.
>     [4] Distributionally robust model-based offline reinforcement learning with near-optimal sample complexity. Shi, Laixi, and Yuejie Chi. arXiv preprint arXiv:2208.05767 (2022).
>     [5] Disentangling by factorising. Hyunjik Kim and Andriy Mnih. In International Conference on Machine Learning, pages 2649–2658. PMLR, 2018.
>     [6] A theory of state abstraction for reinforcement learning. David Abel. In Proceedings of the AAAI Conference on Artificial Intelligence, volume 33, pages 9876–9877, 2019.

---

> > ### Comment · Reviewer_3Qte · 2023-08-13
> > **Thank you for updating related works**
> >
> > I am keeping my score as is, since I am not entirely convinced the related work section is well written just yet. For example, on page 16, there is this sentence
> >
> > "The proposed RSC-MDPs can be regarded as addressing the state uncertainty since the shift of the unobserved confounder leads to state perturbation. In contrast, RSC-MDPs consider the out-of-distribution of the real state that will directly influence the subsequent transition in the environment, but not the observation in POMDPs and SA-MDPs that will not directly influence the environment."
> >
> > I am quite confused what the "In contrast" refers to, since the previous sentence also discusses RSD-MDPs.

---

> > > ### Author Response · Authors · 2023-08-13
> > > **Thanks for engaging in discussion**
> > >
> > > Thank you for engaging in discussion with us and pointing out this question. To directly answer the reviewer's question, 'in contrast' refers to other prior works that also address state uncertainty as our RSC-MDPs. As the reviewer suggested, we have revised the related work section. We hope the following sentences make sense to the reviewer.
> > >
> > > > The proposed RSC-MDPs can be regarded as addressing the state uncertainty since the shift of the unobserved confounder leads to state perturbation. **In contrast to prior works which also address state uncertainty**, RSC-MDPs consider **distribution shift** of the real state that will directly influence the subsequent transitions in the environment, ~~but not~~ instead of the observation in POMDPs and SA-MDPs that will not directly influence the environment **but implicitly influences the policy**."

---

### Author Rebuttal · Authors · 2023-08-09

We thank the reviewers for their careful reading of the paper and their insightful and valuable feedback. We provide new experimental results and discussions to answer some common questions raised by reviewers.

### **(1) Add a new baseline [1], which also tackles spurious correlation in RL.**

|Env|Brightness|Behavior|Crossing|CarType|Lift|Stack|Wipe|Door|
|:-:|:-:|:-:|:-:|:-:|:-:|:-:|:-:|:-:|
|[1] (nominal)|**1.07±0.10**|1.00±0.02|**1.08±0.06**|**1.00±0.02**|**0.99±0.03**|0.90±0.12|**0.93±0.20**|**0.99±0.05**|
|Ours(nominal)|0.92±0.31|**1.06±0.07**|0.96±0.03|0.96±0.03|0.96±0.05|**1.04±0.08**|0.92±0.14|0.98±0.05|
|[1] (shifted)|0.47±0.14|0.83±0.09|0.14±0.03|0.77±0.14|0.35±0.09|0.24±0.12|0.17±0.17|0.05±0.02|
|Ours(shifted)|**0.99±0.11**|**1.02±0.09**|**1.04±0.02**|**1.03±0.02**|**0.98±0.04**|**0.77±0.20**|**0.85±0.12**|**0.61±0.17**|

The results indicate that [1] has very limited robustness in the shifted testing environment compared to our method, especially in Crossing, Stack, Wipe, and Door tasks.


    [1] Can Active Sampling Reduce Causal Confusion in Offline Reinforcement Learning? Gunshi Gupta, Tim G. J. Rudner, Rowan Thomas McAllister, Adrien Gaidon, Yarin Gal, NeurIPS Offline RL Workshop 2022, CleaR 2023

### **(2) Add sensitivity analysis of the ratio of perturbed data $\beta\%$ and the number of perturbation candidates $K$.**

|$\beta\%$|1%|10%|20%|30%|40%|50%|60%|70%|80%|90%|100%|
|:-:|:-:|:-:|:-:|:-:|:-:|:-:|:-:|:-:|:-:|:-:|:-:|
|CarType (nominal)|0.998 ±0.018|0.989 ±0.028|0.987 ±0.020|0.979 ±0.030|0.974 ±0.032|0.984 ±0.027|0.966 ±0.029|0.965 ±0.032|0.952 ±0.035|0.914 ±0.053|0.854 ±0.096|
|CarType (shifted)|0.654 ±0.210|0.826 ±0.127|0.977 ±0.051|1.003 ±0.044|0.995 ±0.042|1.012 ±0.035|1.014 ±0.028|1.017 ±0.028|1.001 ±0.039|0.905 ±0.145|0.825 ±0.172|
|Crossing (nominal)|1.002 ±0.018|0.995 ±0.026|0.993 ±0.024|0.988 ±0.022|0.975 ±0.028|0.968 ±0.031|0.964 ±0.029|0.952 ±0.039|0.909 ±0.041|0.869 ±0.135|0.818 ±0.166|
|Crossing (shifted)|0.675 ±0.120|0.990 ±0.051|1.012 ±0.043|1.031 ±0.028|1.029 ±0.032|1.019 ±0.018|1.025 ±0.028|1.012 ±0.027|0.977 ±0.039|0.915 ±0.140|0.859 ±0.147|

|K|32|64|128|256|512|1024|
|:-:|:-:|:-:|:-:|:-:|:-:|:-:|
|CarType (nominal)|0.972±0.022|0.967±0.029|0.978±0.022|0.975±0.023|0.978±0.026|0.967±0.027|
|CarType (shifted)|1.009±0.037|1.009±0.032|1.020±0.031|1.014±0.033|1.005±0.027|1.009±0.034|
|Crossing (nominal)|0.971±0.030|0.974±0.032|0.987±0.023|0.974±0.030|0.983±0.034|0.980±0.024|
|Crossing (shifted)|1.036±0.022|1.040±0.021|1.041±0.021|1.039±0.017|1.050±0.027|1.039±0.019|

The results demonstrate three important messages:
* **Our RSC-SAC is not sensitive to $\beta$.** As shown in the first table, the proposed RSC-SAC performs well in both nominal and shifted settings --- keeping good performance in the nominal setting and achieving robustness, for a large range of $\beta\%$ (10%-80%). It verifies that RSC-SAC is not sensitive to hyperparameter choices.
* **Our RSC-SAC is not sensitive to $K$.** As shown in the second table, we evaluate the proposed RSC-SAC using different $K = [32,64,\cdots, 1024]$ and achieve similar results. It shows that RSC-SAC is not sensitive to the size $K$ of candidate samples for permutation.
* **Performance-robustness tradeoff.** In the first table, when the ratio of perturbed data $\beta\%$ is very small (1%), RSC-SAC almost achieves the same results as vanilla SAC in nominal settings and there is no robustness in shifted settings. As $\beta\%$ increases (considering more robustness), the performance of RSC-SAC in the nominal setting gradually gets worse, while reversely gets better in the shifted settings (more robust). However, when the ratio is too large (>80%), the performances of RSC-SAC in both settings degrade a lot, since the policy is too conservative so that fails in all environments.


### **(3) Add a visualization of the generated trajectory in Lift environment by our perturbation algorithm.**
The figure is in the uploaded **PDF**.

In the nominal (training) environment, the green cube is always initialized on the left part of the table and the red cube is initialized on the right part. In the shifted (testing) environment, the green cube is always initialized on the right part of the table and the red cube is initialized on the left part.

Figure (a) shows the trajectory of the state in the nominal environment. If we don't do any perturbation of the state, the generated trajectories will still have a spurious correlation (Figure (b)). However, with our perturbation within the state, we generate trajectories that break the spurious correlation and blend the color (Figure (c)). In the shifted (testing) environment, we will have the green cube on the right part and the red cube on the left part. With the counterfactual data (to unobserved confounder) generated by our algorithm (Figure (c)), we can prevent the RL model from overfitting to the spurious correlation between the color and the position of the cube.


### **(4) Discussion about the limitation and potential improvement of our method for high-dimensional tasks.**

> The current method is of great potential to be applied to more complicated problems. In particular, the proposed method requires swapping the dimensions of states to break spurious correlation, where it implicitly assumes that each dimension has semantic meaning. In more complicated problems such as high-dimensional image states, each dimension becomes a pixel without semantic meaning anymore and the dimension size may explode. To extend our method to such cases, we can leverage existing state abstraction techniques [5,6] to project images to a low-dimensional latent space, where each dimension represents a semantic feature. Then, our method can be straightforwardly applied to the low-dimensional latent space.

---

### Decision · Program_Chairs · 2023-09-21

**Decision:**

Accept (poster)

**Comment:**

Paper proposes Robust State-Confounded MDPs as a way to mitigate against effects of correlations induced by unobserved confounders between states. Reviewers' main concerns were mainly the many related works not address or included, but the authors have since included many of these included running additional comparisons, which has largely satisfied most of the reviewers' concerns.